# Large-Scale Methods for Distributionally Robust Optimization

**Daniel Levy,**\* **Yair Carmon**\***, John C. Duchi and Aaron Sidford**
Stanford University
{danilevy,jduchi,sidford}@stanford.edu,, ycarmon@cs.tau.ac.il

## Abstract

We propose and analyze algorithms for distributionally robust optimization of convex losses with conditional value at risk (CVaR) and $\chi^2$ divergence uncertainty sets. We prove that our algorithms require a number of gradient evaluations independent of training set size and number of parameters, making them suitable for large-scale applications. For $\chi^2$ uncertainty sets these are the first such guarantees in the literature, and for CVaR our guarantees scale linearly in the uncertainty level rather than quadratically as in previous work. We also provide lower bounds proving the worst-case optimality of our algorithms for CVaR and a penalized version of the $\chi^2$ problem. Our primary technical contributions are novel bounds on the bias of batch robust risk estimation and the variance of a multilevel Monte Carlo gradient estimator due to Blanchet and Glynn [8]. Experiments on MNIST and ImageNet confirm the theoretical scaling of our algorithms, which are 9–36 times more efficient than full-batch methods.

## 1 Introduction

The growing role of machine learning in high-stakes decision-making raises the need to train reliable models that perform robustly across subpopulations and environments [11, 29, 70, 58, 36, 53, 39]. Distributionally robust optimization (DRO) [3, 66] shows promise as a way to address this challenge, with recent interest in both the machine learning community [68, 74, 22, 69, 34, 55] and in operations research [20, 3, 5, 27]. Yet while DRO has had substantial impact in operations research, a lack of scalable optimization methods has hindered its adoption in common machine learning practice.

In contrast to empirical risk minimization (ERM), which minimizes an expected loss $\mathbb{E}_{S\sim P_0} \ell(x; S)$ over $x \in \mathcal{X} \subset \mathbb{R}^d$ with respect to a training distribution $P_0$, DRO minimizes the expected loss with respect to the worst distribution in an uncertainty set $\mathcal{U}(P_0)$, that is, its goal is to solve

$$\underset{x\in\mathcal{X}}{\text{minimize}} \ \mathcal{L}(x; P_0) \coloneqq \sup_{Q\in\mathcal{U}(P_0)} \mathbb{E}_{S\sim Q} \ell(x; S). \tag{1}$$

The literature considers several uncertainty sets [3, 5, 7, 27], and we focus on two particular choices: (a) the set of distributions with bounded likelihood ratio to $P_0$, so that $\mathcal{L}$ becomes the conditional value at risk (CVaR) [59, 67], and (b) the set of distributions with bounded $\chi^2$ divergence to $P_0$ [3, 16]. Some of our results extend to more general $\phi$-divergence (or Rényi divergence) balls [72]. Minimizers of these objectives enjoy favorable statistical properties [22, 34], but finding them is more challenging than standard ERM. More specifically, stochastic gradient methods solve ERM with a number of $\nabla\ell$ computations independent of both $N$, the support size of $P_0$ (i.e., number of data points), and $d$, the dimension of $x$ (i.e., number of parameters). These guarantees do not directly apply to DRO because the supremum over $Q$ in (1) makes cheap sampling-based gradient estimates biased. As a consequence, existing techniques for minimizing the $\chi^2$ objective [2, 20, 3, 5, 47, 22] have $\nabla\ell$

| | CVaR at level $\alpha$ | $\chi^2$ constraint $\rho$ | $\chi^2$ penalty $\lambda$ |
|---|:---:|:---:|:---:|
| Objective $\mathcal{L}(x; P_0) =$ | $\displaystyle\sup_{\|q\|_\infty \le \frac{1}{\alpha N}} q^\top \ell(x)$ | $\displaystyle\sup_{\mathrm{D}_{\chi^2}(q) \le \rho} q^\top \ell(x)$ | $\displaystyle\sup_{q \in \Delta^N} q^\top \ell(x) - \lambda \mathrm{D}_{\chi^2}(q)$ |
| Subgradient method | $N\epsilon^{-2}$ | $N\epsilon^{-2}$ | $N\epsilon^{-2}$ |
| Dual SGM [Appendix A.3] | $\alpha^{-2}\epsilon^{-2}$ | - | $\lambda^{-2}\epsilon^{-2}$ |
| Subsampling [22] | - | $\rho^2 d\epsilon^{-4}$ | - |
| Stoch. primal-dual [17, 47] | $N\epsilon^{-2}$ | $N\rho\epsilon^{-2}$ | - |
| **Ours** | $\alpha^{-1}\epsilon^{-2}$ (Thm. 2) | $\rho\epsilon^{-3}$ (Thm. 4) | $\lambda^{-1}\epsilon^{-2}$ (Thm. 2) |
| Lower Bound | $\alpha^{-1}\epsilon^{-2}$ (Thm. 3) | $\rho\epsilon^{-2}$ [22] | $\lambda^{-1}\epsilon^{-2}$ (Thm. 3) |

**Table 1.** Number of $\nabla\ell$ evaluations to obtain $\mathbb{E}[\mathcal{L}(x; P_0)] - \inf_{x' \in \mathcal{X}} \mathcal{L}(x'; P_0) \le \epsilon$ when $P_0$ is uniform on $N$ training points. For simplicity we omit the Lipschitz constant of $\ell$, the size of the domain $\mathcal{X}$, and logarithmic factors. We define $\ell_i(x) := \ell(x; S_i)$ and $\mathrm{D}_{\chi^2}(q) := \frac{N}{2}\|q - \frac{1}{N}\mathbf{1}\|_2^2$. The suprema are over $q$ in the simplex.

evaluation complexity scaling linearly (or worse) in either $N$ or $d$, which is prohibitive in large-scale applications.

In this paper, we consider the setting in which $\ell$ is a Lipschitz convex loss, a prototype case for stochastic optimization and machine learning [75, 49], and we propose methods for solving the problem (1) with $\nabla\ell$ complexity independent of sample size $N$ and dimension $d$, and with optimal (linear) dependence on the uncertainty set size. In Table 1 we summarize their complexities and compare them to previous work. Each entry of the table shows the number of (sub)gradient evaluations to obtain a point with optimality gap $\epsilon$; for reference, recall that for ERM the stochastic subgradient method requires order $\epsilon^{-2}$ evaluations, independent of $d$ and $N$. We discuss related work further in Section 1.1 after outlining our approach.

We begin our development in Section 3 by considering the surrogate objective $\overline{\mathcal{L}}(x; n) = \mathbb{E}\,\mathcal{L}(x; \widehat{P}_n)$ corresponding to the average empirical robust objective over random batches of size $n$ sampled from $P_0$. In contrast to (1), it is straightforward to obtain unbiased gradient estimates for $\overline{\mathcal{L}}$—using the mini-batch estimator $\nabla\mathcal{L}(x; \widehat{P}_n)$—and to optimize it efficiently with stochastic gradient methods. To obtain guarantees for the true objective $\mathcal{L}$, we establish uniform bounds on the error $|\mathcal{L}(x; P_0) - \overline{\mathcal{L}}(x; n)|$. For CVaR we prove a bound scaling as $1/\sqrt{n}$ and extend it to other uncertainty sets, including $\chi^2$ balls, via the Kusuoka representation [42]. Notably, for the penalty version of the $\chi^2$ objective (Table 1 right column) we prove a stronger bound scaling as $1/n$. This analysis implies that, for large enough batch size $n$, an $\epsilon/2$-minimizer of $\overline{\mathcal{L}}$ is also an $\epsilon$-minimizer of $\mathcal{L}$. Furthermore, for CVaR and $\chi^2$ penalty we show that the variance of the gradient estimator decreases as $1/n$, and we use Nesterov acceleration to decrease the required number of gradient steps.

To obtain stronger guarantees, in Section 4 we present a theoretically more efficient multi-level Monte Carlo (MLMC) [31, 32] gradient estimator which is a slight modification of the general technique of Blanchet and Glynn [8]. The resulting estimator is unbiased for $\nabla\overline{\mathcal{L}}(x; n)$ but requires only a *logarithmic number of samples* in $n$. For CVaR and $\chi^2$ penalty we control the second moment of the gradient estimator, resulting in complexity bounds scaling with $\epsilon^{-2}$. We further prove that these rates are worst-case optimal up to logarithmic factors.

Unfortunately, direct application of the MLMC estimator for the $\chi^2$ uncertainty set (Table 1 center column) demonstrably fails for certain inputs. Instead, in Appendix E we optimize its Lagrange dual—the $\chi^2$ penalty—with respect to $x$ and Lagrange multiplier $\lambda$. Using a doubling scheme on the $\lambda$ domain, we obtain a complexity guarantee scaling as $\epsilon^{-3}$.

Section 5 presents experiments where we use DRO to train linear models for digit classification (on a mixture between MNIST [44] and typed digits [19]), and ImageNet [60]. To the best of our knowledge, the latter is the largest DRO problem solved to date. In both experiments DRO provides generalization improvements over ERM, and we show that our stochastic gradient estimators require far less $\nabla\ell$ computations—between $9\times$ and $40\times$— than full-batch methods. Our experiments also reveal two facts that our theory only hints at. First, using the mini-batch gradient estimator the error floor due to the difference between $\overline{\mathcal{L}}(x; n)$ and $\mathcal{L}(x; P_0)$ becomes negligible even for batch sizes as

small as 10. Second, while the MLMC estimator avoids these error floors altogether, its increased variance makes it practically inferior to the mini-batch estimator with properly tuned batch size and learning rate. Our code implements our gradient estimators in PyTorch [56] and combines them seamlessly with the framework's optimizers; we show an example code snippet in Appendix F.3.

## 1.1 Related work

Distributionally robust optimization grows from the robust optimization literature in operations research [3, 2, 4, 5], and the fundamental uncertainty about the data distribution at test time makes its application to machine learning natural. Experiments in the papers [47, 28, 22, 34, 17, 40] show promising results for CVaR and $\chi^2$-constrained DRO, while other works highlight the importance of incorporating additional constraints into the uncertainty set definition [38, 24, 55, 61]. Below, we review the prior art on solving these DRO problems at scale.

**Full-batch subgradient method.** When $P_0$ has support of size $N$ it is possible to compute a subgradient of the objective $\mathcal{L}(x; P_0)$ by evaluating $\ell(x; s_i)$ and $\nabla \ell(x; s_i)$ for $i = 1, \ldots, N$, computing the $q \in \Delta^N$ attaining the supremum (1), whence $g = \sum_{i=1}^{N} q_i \nabla \ell(x; s_i)$ is a subgradient of $\mathcal{L}$ at $x$. As the Lipschitz constant of $\mathcal{L}$ is at most that of $\ell$, we may use these subgradients in the subgradient method [51] and find an $\epsilon$ approximate solution in order $\epsilon^{-2}$ steps. This requires order $N\epsilon^{-2}$ evaluations of $\nabla \ell$, regardless of the uncertainty set.

**CVaR.** Robust objectives of the form (1) often admit tractable expression in terms of joint minimization over $x$ and the Lagrange multipliers associated with the constrained maximization over $Q$ [e.g., 59, 66]. For CVaR, this dual formulation is an ERM problem in $x$ and $\eta \in \mathbb{R}$, which we can solve in time independent of $N$ using stochastic gradient methods. We refer to this as "dual SGM," providing the associated complexity bounds in Appendix A.3. Fan et al. [28] apply dual SGM for learning linear classifiers, and Curi et al. [17] compare it to their proposed stochastic primal-dual method based on determinantal point processes. While the latter performs better in practice, its worst-case guarantees scale roughly as $N\epsilon^{-2}$, similarly to the full-batch method. Kawaguchi and Lu [40] propose to only use gradients from the highest $k$ losses in every batch, which is essentially identical to our mini-batch estimator for CVaR; they do not, however, relate their algorithm to CVaR optimization. We contribute to this line of work by obtaining tight characterizations of the mini-batch and MLMC gradient estimators, resulting in optimal complexity bounds scaling as $\alpha^{-1}\epsilon^{-2}$.

**DRO with $\chi^2$ divergence.** Similar dual formulations exist for both the constrained and penalized $\chi^2$ objectives, and dual SGM provides similar guarantees to CVaR for the penalized $\chi^2$ objective. For the constrained-$\chi^2$ problem, the additional Lagrange multiplier associated with the constraint induces a "perspective transform" [3, 22], making the method unstable. Indeed, Namkoong and Duchi [47] report that it fails to converge in practice and instead propose a stochastic primal-dual method with convergence rate $(1 + \rho N)\epsilon^{-2}$. Their guarantee is optimal in the weak regularization regime where $\rho \lesssim 1/N$, but is worse than the full-batch method in the setting where $\rho \gtrsim 1$. Hashimoto et al. [34] propose a different scheme alternating between ERM on $x$ and line search over a Lagrange multiplier, but do not provide complexity bounds. Duchi and Namkoong [22] prove that for a sample of size $N' \approx \rho^2 d\epsilon^{-2}$ the empirical objective converges to $\mathcal{L}(x; P_0)$ uniformly in $x \in \mathcal{X}$; substituting $N'$ into the full-batch complexity bound implies a rate of $\rho^2 d\epsilon^{-4}$. This guarantee is independent of $N$, but features an undesirable dependence on $d$. Ghosh et al. [30] use the mini-batch gradient estimator and gradually increase the batch size to $N$ as optimization progresses; they do not provide convergence rate bounds. We establish concrete rates for fixed batch sizes independent of $N$.

**MLMC gradient estimators.** Multi-level Monte Carlo techniques [31, 32] facilitate the estimation of expectations of the form $\mathbb{E} \mathsf{F}(S_1, \ldots, S_n)$, where the $S_i$ are i.i.d. In this work we leverage a variant of a particular MLMC estimator proposed by Blanchet and Glynn [8]. Prior work [6] uses the estimator of [8] in a DRO formulation of semi-supervised learning with Wasserstein uncertainty sets and $\mathsf{F}(\cdot)$ a ratio of expectations, as opposed to a supremum of expectations in our setting.

## 2 Preliminaries

We collect notation, establish a few assumptions, and provide the most important definitions for the remainder of the paper in this section.

**Notation.** We denote the optimization variable by $x \in \mathbb{R}^d$, and use $s$ (or $S$ when it is random) for a data sample in $\mathbb{S}$. We use $z_l^m$ as shorthand for the sequence $z_l, \ldots, z_m$. For fixed $x$ we denote the cdf of $\ell(x, S)$ by $F(t) := \mathbb{P}(\ell(x, S) \leq t)$ and its inverse by $F^{-1}(u) := \inf\{t : F(t) > u\}$, leaving the

dependence on $x$ and $P_0$ implicit. We use $\|\cdot\|$ to denote Euclidean norm, but remark that many of our results carry over to general norms. We let $\Delta^m$ denote the simplex in $m$ dimensions. We write $1_{\{A\}}$ for the indicator of event $A$, i.e., 1 if $A$ holds and 0 otherwise, and write $\mathbb{I}_{\mathcal{C}}$ for the infinite indicator of the set $\mathcal{C}$, $\mathbb{I}_{\mathcal{C}}(x) = 0$ if $x \in \mathcal{C}$ and $\mathbb{I}_{\mathcal{C}}(x) = \infty$ otherwise. The Euclidean projection to a set $\mathcal{C}$ is $\Pi_{\mathcal{C}}$. We use $\nabla$ to denote gradient with respect to $x$, or, for non-differentiable convex functions, an arbitrary subgradient. We denote the positive part of $t \in \mathbb{R}$ by $(t)_+ := \max\{t, 0\}$. Finally, $f \lesssim g$ means that there exists $C \in \mathbb{R}_+$, independent of any problem parameters, such that $f \leq Cg$ holds; we also write $f \asymp g$ if $f \lesssim g \lesssim f$.

**Assumptions.** Throughout, we assume that the domain $\mathcal{X}$ is closed convex and satisfies $\|x-y\| \leq R$ for all $x, y \in \mathcal{X}$. Moreover, we assume the loss function $\ell : \mathcal{X} \times \mathbb{S} \to [0, B]$ is convex and $G$-Lipschitz in $x$, i.e., $0 \leq \ell(x, s) \leq B$ and $|\ell(x; s) - \ell(y; s)| \leq G\|x-y\|$ for $x, y \in \mathcal{X}$ and $s \in \mathbb{S}$.[2] In some cases, we entertain two additional assumptions:

**Assumption A1.** *The gradient $\nabla\ell(x, s)$ is $H$-Lipschitz in $x$.*

**Assumption A2.** *The inverse cdf $F^{-1}$ of $\ell(x; S)$ is $G_{\mathrm{icdf}}$-Lipschitz for each $x \in \mathcal{X}$.*

Most of our bounds do not require Assumptions A1 and A2. Moreover, in Appendix B.2 we argue that these assumptions are frequently not restrictive.

**The distributionally robust objective.** We consider a slight generalization of $\phi$-divergence distributionally robust optimization (DRO). For a convex $\phi : \mathbb{R}_+ \to \mathbb{R} \cup \{+\infty\}$ satisfying $\phi(1) = 0$, the $\phi$-divergence between distributions $P$ and $Q$ absolutely continuous w.r.t. $P$ by $\mathrm{D}_\phi(Q, P) := \int \phi(\frac{\mathrm{d}Q}{\mathrm{d}P}(s))\mathrm{d}P(s)$. Then, for convex $\phi, \psi$ with $\phi(1) = \psi(1) = 0$, a constraint radius $\rho \geq 0$, and penalty $\lambda \geq 0$ *the general form of the objectives we consider* is

$$\mathcal{L}(x; P) := \sup_{Q:\mathrm{D}_\phi(Q,P)\leq\rho} \left\{ \mathbb{E}_Q[\ell(x; S)] - \lambda\mathrm{D}_\psi(Q, P) \right\}. \tag{2}$$

As previewed, we consider the following objectives for general $P_0$ (nonuniform with infinite support):

- $\chi^2$ **constraint.** $\mathcal{L}_{\chi^2}$ corresponds to $\phi(t) = \chi^2(t) := \frac{1}{2}(t-1)^2$ and $\psi = 0$.
- $\chi^2$ **penalty.** $\mathcal{L}_{\chi^2\text{-pen}}$ corresponds to $\phi = 0$ and $\psi(t) = \chi^2(t) = \frac{1}{2}(t-1)^2$.
- **Conditional value at risk $\alpha \in (0, 1]$ (CVaR).** $\mathcal{L}_{\mathrm{CVaR}}$ corresponds to $\phi = 0$ and $\psi = \mathbb{I}_{[0,1/\alpha]}$.

Additionally, define the following smoothed version of the CVaR objective, which we use in Section 3.

- **KL-regularized CVaR.** $\mathcal{L}_{\mathrm{kl\text{-}CVaR}}$ corresponds to $\phi = 0$ and and $\psi(t) = \mathbb{I}_{[0,1/\alpha]}(t) + t \log t - t + 1$.

In Appendix A we present additional standard formulations and useful properties of these objectives.

With mild abuse of notation, for a sample $s_1^n \in \mathbb{S}^n$, we let

$$\mathcal{L}(x; s_1^n) := \mathcal{L}(x; \widehat{P}[s_1^n]) = \sup_{q \in \Delta^n : \sum_{i \leq n} \frac{1}{n}\phi(nq_i) \leq \rho} \left\{ \sum_{i=1}^{n} \left( q_i \ell(x; s_i) - \frac{1}{n}\psi(nq_i) \right) \right\} \tag{3}$$

denote the loss with respect to the empirical distribution on $s_1^n$. Averaging the robust objective over random batches of size $n$, we define the surrogate objective

$$\overline{\mathcal{L}}(x; n) := \mathbb{E}_{S_1^n \sim P_0^n} \mathcal{L}(x; S_1^n). \tag{4}$$

**Complexity metrics.** We measure complexity of our methods by the number of computations of $\nabla\ell(x; s)$ they require to reach a solution with accuracy $\epsilon$. We can bound (up to a constant factor) the runtime of every method we consider by our complexity measure multiplied by $d + \mathsf{T}_{\mathrm{eval}}$, where $\mathsf{T}_{\mathrm{eval}}$ denotes the time to evaluate $\ell(x; s)$ and $\nabla\ell(x; s)$ at a single point $x$ and sample $s$, and is typically $O(d)$. (In the problems we study, solving the problem (4) given $\ell(x; S_1^n)$ takes $O(n \log n)$ time; see Appendix A.2).

# 3 Mini-batch gradient estimators

In this section, we develop and analyze stochastic subgradient methods using the subgradients of the mini-batch loss (3). That is, we estimate $\nabla \mathcal{L}(x; P_0)$ by sampling a mini-batch $S_1, \ldots, S_n \overset{\text{iid}}{\sim} P_0$ and computing

$$\nabla \mathcal{L}(x; S_1^n) = \sum_{i=1}^{n} q_i^\star \nabla \ell(x; S_i),$$

where $q^\star \in \Delta^n$ attains the supremum in Eq. (3). By definition (4) of the surrogate objective $\overline{\mathcal{L}}$, we have that $\mathbb{E} \nabla \mathcal{L}(x; S_1^n) = \nabla \overline{\mathcal{L}}(x; n)$. Therefore, we expect stochastic subgradient methods using $\nabla \mathcal{L}(x; S_1^n)$ to minimize $\overline{\mathcal{L}}$. However, in general, $\overline{\mathcal{L}}(x; n) \neq \mathcal{L}(x; P_0)$ and $\mathbb{E} \nabla \mathcal{L}(x; S_1^n) \neq \nabla \mathcal{L}(x; P_0)$.

To show that the mini-batch gradient estimator is nevertheless effective for minimizing $\mathcal{L}$, we proceed in three steps. First, in Section 3.1 we prove uniform bounds on the bias $\mathcal{L} - \overline{\mathcal{L}}$ that tend to zero with $n$. Second, in Section 3.2 we complement them with $1/n$ variance bounds on $\nabla \mathcal{L}(x; S_1^n)$. Finally, Section 3.3 puts the pieces together: we apply the SGM guarantees to bound the complexity of minimizing $\overline{\mathcal{L}}$ to accuracy $\epsilon/2$, using Nesterov acceleration to exploit our variance bounds, and choose the mini-batch size $n$ large enough to guarantee (via our bias bounds) that the resulting solution is also an $\epsilon$ minimizer of the original objective $\mathcal{L}$.

## 3.1 Bias analysis

**Proposition 1** (Bias of the batch estimator). *For all $x \in \mathcal{X}$ and $n \in \mathbb{N}$ we have*

$$0 \leq \mathcal{L}(x; P_0) - \overline{\mathcal{L}}(x; n) \lesssim \begin{cases} B \min\{1, (\alpha n)^{-1/2}\} & \text{for } \mathcal{L} = \mathcal{L}_{\text{CVaR}} & (5) \\ B\sqrt{(1+\rho)(\log n)/n} & \text{for } \mathcal{L} = \mathcal{L}_{\chi^2} & (6) \\ B^2 (\lambda n)^{-1} & \text{for } \mathcal{L} = \mathcal{L}_{\chi^2\text{-pen}} & (7) \\ G_{\text{icdf}} \, n^{-1} & \text{for any loss (2),} & (8) \end{cases}$$

*where the bound (8) holds under Assumption A1.*

We present the proof in Appendix B.1.1 and make a few remarks before proceeding to discuss the main proof ideas. First, the bounds (5), (6) and (7) are all tight up to constant or logarithmic factors when $\ell(x, S)$ has a Bernoulli distribution, and so are unimprovable without further assumptions (see Proposition 4 in Appendix B.1.2). One such assumption is that $\ell(x; S)$ has $G_{\text{icdf}}$-Lipschitz inverse-cdf, and it allows us to obtain a general $1/n$ bias bound (8) independent of the uncertainty set size. As we discuss in Appendix B.2.2, this assumption has natural relaxations for uniform distributions with finite supports and, for CVaR at level $\alpha$, we only need the inverse cdf $F^{-1}(\beta)$ to be Lipschitz around $\beta = \alpha$, a common assumption in the risk estimation literature [71].

*Proof sketch.* To show that $\mathcal{L}(x; P_0) \geq \overline{\mathcal{L}}(x; n)$ for every loss of the form (2), we use Lagrange duality to write

$$\mathcal{L}(x; P_0) = \inf_{\eta, \nu} \mathbb{E}_{S_1^n \sim P_0^n} \frac{1}{n} \sum_{i=1}^{n} \Upsilon(x; \eta, \nu; S_i) \text{ and } \overline{\mathcal{L}}(x; n) = \mathbb{E}_{S_1^n \sim P_0^n} \inf_{\eta, \nu} \frac{1}{n} \sum_{i=1}^{n} \Upsilon(x; \eta, \nu; S_i),$$

for some $\Upsilon : \mathcal{X} \times \mathbb{R} \times \mathbb{R}_+ \times \mathbb{S} \to \mathbb{R}$. This exposes the fundamental source of the mini-batch estimator bias: when infimum and expectation do not commute (as is the case in general), exchanging them strictly decreases the result.

Our upper bound analysis begins with CVaR, where $\mathcal{L}_{\text{CVaR}} = \frac{1}{\alpha} \int 1_{\{\beta \geq 1-\alpha\}} F^{-1}(\beta) \mathrm{d}\beta$ and $\overline{\mathcal{L}}_{\text{CVaR}} = \frac{1}{\alpha} \int \mathcal{I}_\alpha(\beta) F^{-1}(\beta) \mathrm{d}\beta$, with $F^{-1}$ the inverse cdf of $\ell(x, S)$ and $\mathcal{I}_\alpha$ a "soft step function" that we write in closed form as a sum of Beta densities. To obtain the bound (5) we express $\int (1_{\{\beta \geq 1-\alpha\}} - \mathcal{I}_\alpha(\beta))_+ \mathrm{d}\beta$ as a sum of binomial tail probabilities and apply Chernoff bounds. For CVaR only, the improved bound (8) follows from arguing that replacing $F^{-1}(\beta)$ with $G_{\text{icdf}} \cdot \beta$ overestimates the bias, and showing that $\int (1_{\{\beta \geq 1-\alpha\}} - \mathcal{I}_\alpha(\beta)) \beta \mathrm{d}\beta \leq (n+1)^{-1}$ for any $\alpha$.

To transfer the CVaR bounds to other objectives we express the objective (2) as a weighted CVaR average over different $\alpha$ values, essentially using the Kusuoka representation of coherent risk measures [42]. Given any bias bound $\text{bb}(\alpha)$ for CVaR at level $\alpha$, this expression implies the bound

$\mathcal{L} - \overline{\mathcal{L}} \leq \sup_{w \in \mathcal{W}(\mathcal{L})} \int \mathrm{bb}(\alpha) \mathrm{d}w(\alpha)$, where $\mathcal{W}(\mathcal{L})$ is a set of probability measures. Substituting $\mathrm{bb}(\alpha) = 1/\sqrt{n\alpha}$ and using the Cauchy-Schwartz inequality gives the bound (6), while substituting $\mathrm{bb}(\alpha) = G_{\mathrm{icdf}}/n$ shows this bound in fact holds for any $\mathcal{L}$, as we claim in (8).

Showing the bound (7) requires a fairly different argument. Our proof uses the dual representation of $\mathcal{L}_{\chi^2\text{-pen}}$ as a minimum of an expected risk over a Lagrange multiplier $\eta$ imposing the constraint that $q$ in (3) sums to 1 (or that $Q$ in (2) integrates to 1). Using convexity with respect to $\eta$ we relate the value of the risk at $\eta_n$ (the minimizer for sample $S_1^n$) to $\eta^\star$ (the population minimizer), which on expectation are $\overline{\mathcal{L}}_{\chi^2\text{-pen}}$ and $\mathcal{L}_{\chi^2\text{-pen}}$, respectively. We then apply Cauchy-Schwartz and bound the variance of $\eta_n$ with the Efron-Stein inequality [26] to obtain a $1/n$ bias bound. $\qquad \square$

### 3.2 Variance analysis

With the bias bounds in Proposition 4 established, we analyze the variance of the stochastic gradient estimators $\nabla \mathcal{L}(x; S_1^n)$. More specifically, we prove that the variance of the mini-batch gradient estimator decreases as $1/n$ for penalty-type robust objectives (with $\phi = 0$) for which the maximizing $Q$ has bounded $\chi^2$ divergence from $P_0$, which we call "$\chi^2$-bounded objectives" (see Appendix A.4). Noting that $\mathcal{L}_{\mathrm{kl\text{-}CVaR}}$ (with $\mathcal{L}_{\mathrm{CVaR}}$ as a special case) and $\mathcal{L}_{\chi^2\text{-pen}}$ are $\chi^2$-bounded yields the following.

**Proposition 2** (Variance of the batch estimator). *For all $n \in \mathbb{N}$, $x \in \mathcal{X}$, and $S_1^n \sim P_0^n$,*

$$\mathrm{Var}\Big[\nabla \mathcal{L}_{\mathrm{kl\text{-}CVaR}}(x; S_1^n)\Big] \lesssim \frac{G^2}{\alpha n} \;\; \textit{and} \;\; \mathrm{Var}\Big[\nabla \mathcal{L}_{\chi^2\text{-pen}}(x; S_1^n)\Big] \lesssim \frac{G^2(1 + B/\lambda)}{n}.$$

(Note that the variance bound on $\mathcal{L}_{\mathrm{kl\text{-}CVaR}}$ is independent of $\lambda$ and therefore holds also for $\mathcal{L}_{\mathrm{CVaR}}$ where $\lambda = 0$). We prove Proposition 2 in Appendix B.3 and provide a proof sketch below.[3] Unfortunately, the bounds do not extend to the $\chi^2$ constrained formulation: in Appendix B.3 (Proposition 5) we prove that for any $n$ there exist $\ell$, $P_0$, and $x$ such that $\mathrm{Var}[\nabla \mathcal{L}_{\chi^2}(x; P_0)] \gtrsim \rho$. Whether Proposition 2 holds when adding a $\chi^2$ penalty to the $\chi^2$ constraint remains an open question.

*Proof sketch.* The Efron-Stein inequality [26] is $\mathrm{Var}[\nabla \mathcal{L}(x; S_1^n)] \leq \frac{n}{2} \mathbb{E} \|\nabla \mathcal{L}(x; S_1^n) - \nabla \mathcal{L}(x; \tilde{S}_1^n)\|^2$, where $S_1^n$ and $\tilde{S}_1^n$ are identical except in a random entry $I \in [n]$ for which $\tilde{S}_I$ is an i.i.d. copy of $S_I$. We bound $\|\nabla \mathcal{L}(x; S_1^n) - \nabla \mathcal{L}(x; \tilde{S}_1^n)\| \leq Gq_I + G\|q - \tilde{q}\|_1$ with the triangle inequality, where $q$ and $\tilde{q}$ attain the maximum in (3) for $S$ and $\tilde{S}$, respectively. The crux of our proof is the equality $\|q - \tilde{q}\|_1 = 2|q_I - \tilde{q}_I|$, which holds since increasing one coordinate of $\ell(x; S_1), \ldots, \ell(x; S_n)$ must decrease all other coordinates in $q$. Noting that $\mathbb{E} (q_I - \tilde{q}_I)^2 \leq 4 \mathbb{E}(q_I - 1/n)^2 = \frac{8}{n^2} \mathbb{E} \mathrm{D}_{\chi^2}(q, \frac{1}{n}\mathbf{1})$, the results follow by observing that $\mathrm{D}_{\chi^2}(q, \frac{1}{n}\mathbf{1})$ is bounded by $1/\alpha$ and $B/\lambda$ for $\mathcal{L}_{\mathrm{kl\text{-}CVaR}}$ and $\mathcal{L}_{\chi^2\text{-pen}}$, respectively. $\qquad \square$

### 3.3 Complexity guarantees

With the bias and variance guarantees established, we now provide bounds on the complexity of minimizing $\mathcal{L}(x; P_0)$ to arbitrary accuracy $\epsilon$ using standard gradient methods with the gradient estimator $\tilde{g}(x) = \nabla \mathcal{L}(x; S_1^n)$. (Recall from Section 2 that we measure complexity by the number of individual first order evaluations $(\ell(x; s), \nabla \ell(x; s))$.) Writing $\Pi_{\mathcal{X}}$ for the Euclidean projection onto $\mathcal{X}$, the stochastic gradient method (SGM) with fixed step-size $\eta$ and $x_0 \in \mathcal{X}$ iterates

$$x_{t+1} = \Pi_{\mathcal{X}}(x_t - \eta \tilde{g}(x_t)), \;\; \text{and} \;\; \bar{x}_t = \frac{1}{t} \sum_{\tau \leq t} x_\tau. \tag{9}$$

We also consider Nesterov's accelerated gradient method [50, 43]. For $x_0 = y_0 = z_0 \in \mathcal{X}$, a fixed step-size $\eta > 0$ and a sequence $\{\theta_t\}$, we iterate

$$z_{t+1} = \Pi_{\mathcal{X}}(z_t - \tfrac{\eta}{\theta_t}\tilde{g}(x_t)), \; y_{t+1} = \theta_t z_{t+1} + (1-\theta_t)y_t, \; \text{and} \; x_{t+1} = \theta_{t+1}z_{t+1} + (1-\theta_t)y_{t+1}. \tag{10}$$

In Appendix B.4, we state the rates of convergence of the iterations (9) and (10) following the analysis in [43], with a small variation where the stochastic gradient estimates are unbiased for a uniform approximation of the true objective with additive error $\delta$. Since our gradient estimator has norm bounded by $G$, SGM allows us to find an $\epsilon$-minimizer of $\overline{\mathcal{L}}$ in $T \asymp (GR)^2/\epsilon^2$ steps. Therefore,

choosing $n$ large enough in accordance to Proposition 1 guarantees that we find an $\epsilon$-minimizer of $\mathcal{L}$. The accelerated scheme (10) admits convergence guarantees that scale with the gradient estimator variance instead of its second moment, allowing us to leverage Proposition 2 to reduce $T$ to the order of $1/\epsilon$. The accelerated guarantees require the loss $\mathcal{L}$ to have order $1/\epsilon$-Lipschitz gradients—fortunately, this holds for $\mathcal{L}_{\chi^2\text{-pen}}$ and $\mathcal{L}_{\text{kl-CVaR}}$.

**Claim 1.** *Let Assumption A1 hold. For all $P$, $\nabla\mathcal{L}_{\text{kl-CVaR}}(x;P)$ and $\nabla\mathcal{L}_{\chi^2\text{-pen}}(x;P)$ are $(\frac{G^2}{\lambda}+H)$-Lipschitz in $x$, and $0 \le \mathcal{L}_{\text{CVaR}}(x;P) - \mathcal{L}_{\text{kl-CVaR}}(x;P) \le \lambda\log(1/\alpha)$ for all $x$.*

See proof in Appendix A.1.6. Thus, to minimize $\mathcal{L}_{\text{CVaR}}$ we instead minimize $\mathcal{L}_{\text{kl-CVaR}}$ and choose $\lambda \asymp \epsilon/\log(1/\alpha)$ to satisfy the smoothness requirement while incurring order $\epsilon$ approximation error. For $\mathcal{L}_{\chi^2\text{-pen}}$ with $\lambda \ge \epsilon$ we get sufficient smoothness for free.[4]

As computing every gradient estimator requires $n$ evaluations of $\nabla\ell$, the total gradient complexity is $nT$, and we have the following suite of guarantees (see Appendix B.5 for proof).

**Theorem 1.** *Let Assumptions A1 and A2 hold, possibly trivially (with $H = \infty$ or $G_{\text{icdf}} = \infty$). Let $\epsilon \in (0, B)$ and write $\nu = \frac{H}{G^2}\epsilon$. With suitable choices of the batch size $n$ and iteration count $T$, the gradient methods (9) and (10) find $\bar{x}$ satisfying $\mathbb{E}\,\mathcal{L}(\bar{x}, P_0) - \inf_{x' \in \mathcal{X}} \mathcal{L}(x'; P_0) \le \epsilon$ with complexity $nT$ admitting the following bounds.*

- *For $\mathcal{L} = \mathcal{L}_{\text{CVaR}}$, we have $nT \lesssim \frac{(GR)^2}{\alpha\epsilon^2}\left(1 + \min\left\{\frac{\alpha G_{\text{icdf}}\sqrt{\log\frac{1}{\alpha}+\nu}}{GR}, \frac{B^2\sqrt{\log\frac{1}{\alpha}+\nu}}{GR\epsilon}, \frac{B^2}{\epsilon^2}\right\}\right)$.*

- *For $\mathcal{L} = \mathcal{L}_{\chi^2\text{-pen}}$ with $\lambda \le B$, we have $nT \lesssim \frac{(GR)^2 B}{\lambda\epsilon^2}\left(1 + \min\left\{\frac{B}{GR}\sqrt{\frac{\epsilon(1+\nu)}{\lambda}}, \frac{B}{\epsilon}\right\}\right)$.*

- *For $\mathcal{L} = \mathcal{L}_{\chi^2}$, we have $nT \lesssim \frac{(1+\rho)(GR)^2 B^2}{\epsilon^4}\log\frac{(1+\rho)B^2}{\epsilon^2}$.*

- *For any loss of the from (2), we have $nT \lesssim \frac{(GR)^2 G_{\text{icdf}}}{\epsilon^3}$.*

The smoothness parameter $H$ only appears in rates resulting from Nesterov acceleration. Even there, $H$ appears in lower-order terms in $\epsilon$ since $\nu = \frac{H}{G^2}\epsilon$. We also note that the final $G_{\text{icdf}}\epsilon^{-3}$ rate holds even when the uncertainty set is the entire simplex; therefore, when $G_{\text{icdf}} < \infty$ it is possible to approximately minimize the maximum loss [64] in sublinear time. Theorem 1 achieves the claimed rates of convergence in Table 1 in certain settings. In particular, it recovers the rates for $\mathcal{L}_{\text{CVaR}}$ and $\mathcal{L}_{\chi^2\text{-pen}}$ (the first and last column of the table) when $\nu \lesssim 1$, $\lambda \gtrsim (B/(GR))^2\epsilon$, and $\alpha \lesssim GR/G_{\text{icdf}}$. In the next section, we show how to attain the claimed optimal rates for $\mathcal{L}_{\text{CVaR}}$ and $\mathcal{L}_{\chi^2\text{-pen}}$ without conditions, returning to address the rates for the constrained $\chi^2$ objective $\mathcal{L}_{\chi^2}$ in Appendix E.

## 4  Multi-level Monte Carlo (MLMC) gradient estimators

In the previous section, we optimized the mini-batch surrogate $\overline{\mathcal{L}}(x;n)$ to the risk $\mathcal{L}(x;P_0)$, using Proposition 1 to guarantee the surrogate's fidelity for sufficiently large $n$. The increasing (linear) complexity of computing the estimator $\nabla\overline{\mathcal{L}}(x;S_1^n)$ as $n$ grows limits the (theoretical) efficiency of the method. To that end, in this section we revisit a multi-level Monte Carlo (MLMC) gradient estimator of Blanchet and Glynn [8] to form an unbiased approximation to $\nabla\overline{\mathcal{L}}(x;n)$ whose sample complexity is logarithmic in $n$. We provide new bounds on the variance of this MLMC estimator, leading immediately to improved (and, as we shall see, optimal) efficiency estimates for stochastic gradient methods using it.

To define the estimator, let $J \sim \min\{\text{Geo}(1/2), j_{\max}\}$ be a truncated geometric random variable supported on $\{1, \dots, j_{\max}\}$, and let $q(j) = \mathbb{P}(J = j) = 2^{-j+1\{j=j_{\max}\}}$. For a realization of $J$ we draw a sample of size $2^J n_0$ and compute the multi-level Monte-Carlo estimator as follows:

$$\widehat{\mathcal{M}}[\nabla\overline{\mathcal{L}}] := \nabla\overline{\mathcal{L}}(x; S_1^{n_0}) + \frac{1}{q(J)}\widehat{\mathcal{D}}_{2^J n_0}, \text{ where } \widehat{\mathcal{D}}_k := \nabla\overline{\mathcal{L}}(x; S_1^k) - \frac{\nabla\overline{\mathcal{L}}(x; S_1^{k/2}) + \nabla\overline{\mathcal{L}}(x; S_{k/2+1}^k)}{2}.$$

$$(11)$$

Our estimator differs from the proposal [8] in two aspects: the distribution of $J$ and the option to set $n_0 > 1$. As we further discuss in Appendix C.3, the former difference is crucial for our setting, while the latter is pratically and theoretically helpful yet not crucial. The following properties of the MLMC estimator are key to our analysis (see Appendix C.1 for proofs).

**Claim 2.** *The estimator $\widehat{\mathcal{M}}[\nabla\mathcal{L}]$ with parameters $n = 2^{j_{\max}} n_0$ satisfies*

$$\mathbb{E}\,\widehat{\mathcal{M}}[\nabla L] = \mathbb{E}\,\nabla\mathcal{L}(x; S_1^n) = \nabla\overline{\mathcal{L}}(x; n), \text{ requiring expected sample size } \mathbb{E}\,2^J n_0 = n_0(1 + \log_2(n/n_0)).$$

**Proposition 3** (Second moment of MLMC gradient estimator). *For all $x \in \mathcal{X}$, the multi-level Monte Carlo estimator with parameters $n$ and $n_0$ satisfies*

$$\mathbb{E}\left\|\widehat{\mathcal{M}}\big[\nabla\mathcal{L}_{\mathrm{CVaR}}\big]\right\|^2 \lesssim \left(1 + \frac{\log\frac{n}{n_0}}{\alpha n_0}\right)G^2 \ \ and \ \ \mathbb{E}\left\|\widehat{\mathcal{M}}\big[\nabla\mathcal{L}_{\chi^2\text{-pen}}\big]\right\|^2 \lesssim \left(1 + \frac{B\log\frac{n}{n_0}}{\lambda n_0}\right)G^2.$$

Claim 2 follows from a simple calculation, while the core of Proposition 3 is a sign-consistency argument for simplifying a 1-norm, similar to the proof of Proposition 2.

Further paralleling Proposition 2, we obtain similar bounds on the MLMC estimates of $\mathcal{L}_{\mathrm{CVaR}}$ and $\mathcal{L}_{\chi^2\text{-pen}}$ (in addition to their gradients), and demonstrate that similar bounds fail to hold for $\nabla\mathcal{L}_{\chi^2}$ (Proposition 7 in Appendix C.1). Therefore, directly using the MLMC estimator on $\nabla\mathcal{L}_{\chi^2}$ cannot provide guarantees for minimizing $\mathcal{L}_{\chi^2}$; instead, in Appendix E we develop a doubling scheme that minimizes the dual objective $\mathcal{L}_{\chi^2\text{-pen}}(x; P_0) + \lambda\rho$ jointly over $x$ and $\lambda$. This scheme relies on MLMC estimators for both the gradient $\nabla\mathcal{L}_{\chi^2\text{-pen}}$ and the derivative of $\mathcal{L}_{\chi^2\text{-pen}}$ with respect to $\lambda$.

Proposition 3 guarantees that the second moment of our gradient estimators remain bounded by a quantity that depends logarithmically on $n$. For these estimators, Proposition 6 thus directly provides complexity guarantees to minimize $\mathcal{L}_{\mathrm{CVaR}}$ and $\mathcal{L}_{\chi^2\text{-pen}}$. We also provide a high probability bound on the total complexity of the algorithm using a one-sided Bernstein concentration bound. We state the guarantee below and present a short proof in Appendix C.2.

**Theorem 2** (MLMC complexity guarantees). *For $\epsilon \in (0, B)$, set $n \asymp \frac{B^2}{\alpha\epsilon^2}$, $1 \lesssim n_0 \lesssim \frac{\log n}{\alpha}$ and $T \asymp \frac{(GR)^2}{n_0\alpha\epsilon^2}\log^2 n$. The stochastic gradient iterates (9) with $\tilde{g}(x) = \widehat{\mathcal{M}}[\nabla\mathcal{L}_{\mathrm{CVaR}}(x; \cdot)]$ satisfy $\mathbb{E}[\mathcal{L}_{\mathrm{CVaR}}(\bar{x}_T; P_0)] - \inf_{x \in \mathcal{X}}\mathcal{L}_{\mathrm{CVaR}}(x; P_0) \leq \epsilon$ with complexity at most*

$$n_0 \log_2\left(\frac{n}{n_0}\right)T + 5\sqrt{(n\log n)^2 + n_0 n T \log n} \lesssim \frac{(GR + B)^2}{\alpha\epsilon^2}\log^2\frac{B^2}{\alpha\epsilon^2} \ \ w.p \ \geq 1 - \frac{1}{n}.$$

*The same conclusion holds when replacing $\mathcal{L}_{\mathrm{CVaR}}$ with $\mathcal{L}_{\chi^2\text{-pen}}$ and $\alpha^{-1}$ with $1 + B/\lambda$.*

**Lower bounds.** We match the guarantees of Theorem 2 with lower bounds that hold in a standard *stochastic oracle* model [48, 43, 10], where algorithms interact with a problem instance by iteratively querying $x_t \in \mathcal{X}$ (for $t \in \mathbb{N}$) and observing $\ell(x_t; S)$ and $\nabla\ell(x_t; S)$ with $S \sim P_0$ (independent of $x_t$). All algorithms we consider fit into this model, with each gradient evaluation corresponding to an oracle query. Therefore, to demonstrate that our MLMC guarantees are unimprovable in the worst case (ignoring logarithmic factors), we formulate a lower bound on the number of queries any oracle-based algorithm requires.

**Theorem 3** (Minimax lower bounds). *Let $G, R, \alpha, \lambda > 0$, $\epsilon \in (0, GR/64)$, and sample space $\mathbb{S} = [-1, 1]$. There exists a numerical constant $c > 0$ such that the following holds.*

- *For each $d \geq 1$, domain $\mathcal{X} = \{x \in \mathbb{R}^d \mid \|x\| \leq R\}$, and any algorithm, there exists a distribution $P_0$ on $\mathbb{S}$ and convex $G$-Lipschitz loss $\ell : \mathcal{X} \times \mathbb{S} \to [0, GR]$ such that*

$$T \leq c\frac{(GR)^2}{\alpha\epsilon^2} \ \ implies \ \ \mathbb{E}[\mathcal{L}_{\mathrm{CVaR}}(x_T; P_0)] - \inf_{x' \in \mathcal{X}}\mathcal{L}_{\mathrm{CVaR}}(x'; P_0) > \epsilon.$$

- *There exists $d_\epsilon \lesssim (GR)^2\epsilon^{-2}\log\frac{GR}{\epsilon}$ such that for $\mathcal{X} = \{x \in \mathbb{R}^d \mid \|x\| \leq R\}$, the same conclusion holds when replacing $\mathcal{L}_{\mathrm{CVaR}}$ with $\mathcal{L}_{\chi^2\text{-pen}}$ and $\alpha$ with $\lambda/(GR)$.*

We present the proof in Appendix D. Our proof for the penalized $\chi^2$ lower bound leverages a classical high-dimensional hard instance construction for oracle-based optimization, while our proof for CVaR is information-theoretic. Consequently, the CVaR lower bound is stronger: it holds for $d = 1$ and extends to a global model where at every round the oracle provides the entire function $\ell(\cdot; S)$ rather than $\ell(x; S)$ and $\nabla\ell(x; S)$ at the query point $x$.

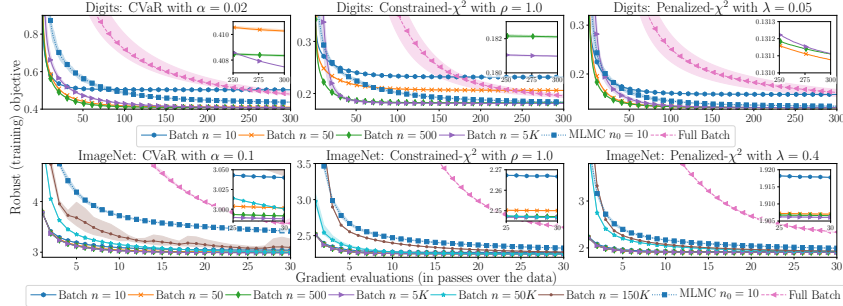

**Figure 1.** Convergence of DRO objective in our digits and ImageNet classification experiments. Shaded areas indicate range of variability across 5 repetitions (minimum to maximum), and the zoomed-in regions highlight the (often very low) "bias floor" of small batch sizes.

## 5 Experiments

We test our theoretical predictions with experiments on two datasets. Our main focus is measuring how the total work in solving the DRO problems depends on different gradient estimators. In particular, we quantify the tradeoffs in choosing the mini-batch size $n$ in the estimator $\nabla\mathcal{L}(x; S_1^n)$ of Section 3 and the effect of using the MLMC technique of Section 4. To ensure that we operate in practically meaningful settings, our experiments involve heterogeneous data, and we tune the DRO objective to improve the generalization performance of ERM on the hardest subpopulation. We provide a full account of experiments in Appendix F and summarize them below.

Our digit recognition experiment reproduces [22, Section 3.2], where the training data includes the 60K MNIST training images mixed with 600 images of typed digits from [19], while our ImageNet experiment uses the ILSVRC-2012 1000-way classification task. In each experiment we use DRO to learn linear classifiers on top of pretrained neural network features (i.e., training the head of the network), taking $\ell$ to be the logarithmic loss with squared-norm regularization; see Appendix F.1. Each experiments compares different gradient estimators for minimizing the $\mathcal{L}_{\text{CVaR}}$, $\mathcal{L}_{\chi^2}$ and $\mathcal{L}_{\chi^2\text{-pen}}$ objectives. Appendix F.2 details our hyper-parameter settings and their tuning procedures.

Figure 1 plots the training objective as optimization progresses. In Appendix F.4 we provide expanded figures that also report the robust generalization performance. We find that the benefits of DRO manifest mainly when the metric of interest is continuous (e.g., log loss) as opposed to the 0-1 loss.

Our analysis in Section 3.1 bounds the suboptimality of solutions resulting from using a mini-batch estimators with batch size $n$, showing it must vanish as $n$ increases. Figure 1 shows that smaller batch sizes indeed converge to suboptimal solutions, and that their suboptimality becomes negligible very quickly: essentially every batch size larger than 10 provides fairly small bias (with the exception of $\mathcal{L}_{\chi^2}$ in the digits experiment). The effect of bias is particularly weak for $\mathcal{L}_{\chi^2\text{-pen}}$, consistent with its superior theoretical guarantees. We note, however, that the suboptimality we see in practice is far smaller than the worst-case bounds in Proposition 1; we investigate this in detail in Appendix F.5.

While the MLMC estimator does not suffer from a bias floor (by design), it is also much slower to converge. This may appear confusing, since the MLMC convergence guarantees are optimal (for $\mathcal{L}_{\text{CVaR}}$ and $\mathcal{L}_{\chi^2\text{-pen}}$) while the mini-batch estimator achieves the optimal rate only under certain assumptions. Recall, however, that these assumptions are smoothness of the loss (which holds in our experiments) and—for CVaR—sufficiently rapid decay of the bias floor, which we verify empirically.

For batch sizes in the range 50–5K, the traces in Figure 1 look remarkably similar. This is consistent with our theoretical analysis for $\mathcal{L}_{\text{CVaR}}$ and $\mathcal{L}_{\chi^2\text{-pen}}$, which shows that the variance decreases linearly with the batch size and we may therefore (with Nesterov acceleration) increase the step size proportionally and expect the total work to remain constant. As theory predicts, this learning rate increase is only possible up to a certain batch size (roughly 5K in our experiments), after which larger batches become less efficient. Indeed, to reach within 2% of the optimal value, the full-batch method requires 27–36× more work than batch sizes 50–5K for ImageNet, and 9–16× more work for the digits experiment (see Table 4 and 5 for a precise breakdown of these numbers).

We also repeat our experiments with the dual SGM and prima-dual methods mentioned in Table 1 and compare them with them our proposed method; see Appendix F.6 for details.

## Broader Impact

The robustness of machine learning (ML) models, or lack thereof, has far-reaching present and future societal consequences: in autonomous vehicles [39, 18], medical diagnosis [53], facial recognition [11], credit scoring [29], and recidivism prediction [12, 1], failure of ML to perform robustly across sub-population or under distribution shift can have disastrous real-life consequences, particularly for members of underserved and/or under-represented groups.

Distributionally robust optimization (DRO) is emerging as a methodology for imposing the constraint that models perform uniformly well across subgroups, and several works conduct experiments demonstrating its benefit in promoting fairness [34, 24, 74] and robustness [68, 61] in ML. However, the computational experiments in these works are relatively small in scale, and there exist serious computational impediments to scaling up DRO. Consequently, the potential benefits of several DRO formulations remain unexplored.

The main contribution of our work is in strengthening the theoretical and algorithmic foundations of two fundamental DRO formulations. In particular, for $\chi^2$-divergence uncertainty sets we give the first proof that stochastic gradient methods can scale to large data similarly to they way they scale for standard empirical risk minimization. We believe that our algorithms will serve a basis for future experimentation with CVaR and $\chi^2$ divergence DRO, and we hope that the resulting findings would lead to more robust and fair machine learning algorithms with positive societal impact. Towards that end, we will release an implementation of our DRO gradient estimators that integrates seamlessly into the PyTorch optimization framework and is therefore suitable for application in a wide range of ML tasks.

In addition, we believe that our work is a step towards a suite of algorithms capable of solving a broader class of DRO problems at scale, including e.g., uncertainty set with explicit group structure as proposed in [38, 61]. We believe that such algorithm suite will empower machine learning researchers and engineers to create more reliable and ethical systems.

However, greater applicability and simplicity always comes with the risk of irresponsible and superficial use. In particular, we are concerned with the possibility that DRO might become a marketing scheme to sell off ML systems as robust without proper verification. Therefore, the development of robust training procedures must go hand-in-hand with the development of rigorous and independent evaluation methodologies for auditing of algorithms [36, 54, 41, 14, 45].

### Acknowledgments

The authors would like to thank Hongseok Namkoong for discussions and insights, as well as Nimit Sohoni for comments on an earlier draft.

### Sources of funding

DL, YC and JCD were supported by the NSF under CAREER Award CCF-1553086 and HDR 1934578 (the Stanford Data Science Collaboratory) and Office of Naval Research YIP Award N00014-19-2288. YC was supported by the Stanford Graduate Fellowship. AS is supported by a Microsoft Research Faculty Fellowship, NSF CAREER Award CCF-1844855, NSF Grant CCF-1955039, a PayPal research gift, and a Sloan Research Fellowship.

### Competing interests

The authors declare no competing interests.

## Footnotes

\*Equal contribution. Code is available on GitHub at https://github.com/daniellevy/fast-dro/.

[2]Our results hold also when $B$ denotes $\sup_{x \in \mathcal{X}, s, s' \in \mathbb{S}}\{\ell(x; s) - \ell(x; s')\}$. The Lipschitz loss and bounded domain assumptions imply $B \leq B_0 + GR$ if $\inf_{x \in \mathcal{X}} \ell(x; s) - \inf_{x' \in \mathcal{X}} \ell(x'; s') \leq B_0$ for all $s, s' \in \mathbb{S}$, which typically holds with $B_0 \approx 0$ in regression and classification problems.

[3]In the appendix we provide bounds on the variance of $\mathcal{L}(x; S_1^n)$ in addition to $\nabla \mathcal{L}(x; S_1^n)$.

[4]We can also handle the case $\lambda < \epsilon$ by adding a KL-divergence term to $\psi$ for $\mathcal{L}_{\chi^2\text{-pen}}$.

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
