[Supplementary Material]

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

[5]To obtain an estimate that has error $\lesssim \epsilon$ with high probability, we can use the median of a logarithmic number of iid copies of the batch estimator.

[6]For CVaR it is fact possible to compute $\overline{\mathcal{L}}(\bar{x}_T; n)$ in closed form via (22); we do that for the Digits experiment. Scaling the computation to ImageNet is nontrivial, so there we use an empirical estimate instead.

[7]The performance of the primal-dual method appears fairly insensitive to the use of momentum so we keep the parameter the same as in our previous experiments for simplicity.

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

# Appendix

## A  Extended preliminaries

In this section we collect several basic results which we use in subsequent derivations in the paper: Section A.1 gives several additional characterization of the robust objective $\mathcal{L}$, Section A.2 briefly discusses the computation of $\mathcal{L}$ and its costs, Section A.3 gives a short derivation of the complexity guarantees for "dual SGM" in Table 1, and Section A.4 introduces the notion of losses contained in a $\chi^2$ divergence ball. Finally, Section A.5 lists a few standard probabilistic bounds.

### A.1  Characterization of the robust objective

Here we give several equivalent characterizations of the robust objective

$$\mathcal{L}(x; P) := \sup_{Q \ll P : \mathrm{D}_\phi(Q, P) \leq \rho} \Big\{ \mathbb{E}_{S \sim Q}[\ell(x; S)] - \lambda \mathrm{D}_\psi(Q, P) \Big\}. \tag{12}$$

where $\psi$, $\phi$ are closed convex functions from $\mathbb{R}_+$ to $\mathbb{R}$ satisfying $\psi(1) = \phi(1) = 0$,

$$\mathrm{D}_\phi(Q, P) := \int \phi\Big(\frac{\mathrm{d}Q}{\mathrm{d}P}\Big) \mathrm{d}P, \text{ and } \mathrm{D}_\psi(Q, P) := \int \psi\Big(\frac{\mathrm{d}Q}{\mathrm{d}P}\Big) \mathrm{d}P.$$

For $\widehat{P}[s_1^n]$ uniform on $s_1, s_2, \ldots, s_n$ (which we abbreviate $s_1^n$), we write

$$\mathcal{L}(x; s_1^n) := \mathcal{L}(x; \widehat{P}[s_1^n]) = \sup_{q \in \Delta^n : \sum_{i \leq n} \frac{1}{n} \phi(nq_i) \leq \rho} \Big\{ \sum_{i \leq n} \big( q_i \ell(x; s_i) - \tfrac{1}{n} \psi(nq_i) \big) \Big\}. \tag{13}$$

#### A.1.1  Inverse-cdf formulation

Instead of expressing the objective in terms of distribution over $\mathbb{S}$, we can characterize the robust loss in terms of the inverse cdf of the distribution (over $\mathbb{R}$) of $\ell(x; S)$. Let $F^{-1}$ denotes the inverse cdf of $\ell(x; S)$ under $P$. Note that $\ell(x; S)$ with $S \sim P$ is equal in distribution to $F^{-1}(U)$ with $U \sim \mathsf{Unif}([0, 1])$. Therefore,

$$\mathcal{L}(x; P) := \sup_{Q' : \mathrm{D}_\phi(Q', \mathsf{Unif}([0,1])) \leq \rho} \Big\{ \mathbb{E}_{U \sim Q'}[F^{-1}(U)] - \lambda \mathrm{D}_\psi(Q', \mathsf{Unif}([0, 1])) \Big\}$$

$$= \sup_{r \in \mathcal{R}} \int_0^1 \Big[ r(u) F^{-1}(u) - \lambda \psi(r(u)) \Big] \mathrm{d}u, \tag{14}$$

where the last equality follows from writing $r(u) = \frac{\mathrm{d}Q'}{\mathrm{d}\mathsf{Unif}([0,1])}(u)$, and the set $\mathcal{R}$ is

$$\mathcal{R} := \Big\{ r : [0, 1] \to \mathbb{R}_+ \ \Big| \ \int_0^1 r(u) \mathrm{d}u = 1 \text{ and } \int_0^1 \phi(r(u)) \mathrm{d}u \leq \rho \Big\}. \tag{15}$$

#### A.1.2  Dual formulation

We can convert the maximization over $r$ in Eq. (14) (or $Q$ in (12)) with minimization over Lagrange multipliers for the constraint that $r$ sums to 1 and the $\phi$-divergence constraint, yielding

$$\mathcal{L}(x; P) = \inf_{\eta \in \mathbb{R}, \nu \geq 0} \Upsilon(x, \eta, \nu; P), \text{ where}$$

$$\Upsilon(x, \eta, \nu; P) := \int_0^1 \sup_{r \in \mathbb{R}_+} \Big[ r F^{-1}(u) - \eta(r - 1) - \nu(\phi(r) - \rho) - \lambda \psi(r) \Big] \mathrm{d}u, \tag{16}$$

where the strong duality follows Shapiro [66, Sec. 3.2]. Writing $(g)^*[v] := \sup_{t \in \mathrm{dom}(g)}\{vt - g(t)\}$ for the conjugate function of $g$, we may express $\Upsilon$ as

$$\Upsilon(x, \eta, \nu; P) = \int_0^1 (\nu\phi + \lambda\psi)^*[F^{-1}(u) - \eta] \mathrm{d}u + \eta + \nu\rho = \mathbb{E}(\nu\phi + \lambda\psi)^*[\ell(x; S) - \eta] + \eta + \nu\rho, \tag{17}$$

where the expectation is over $S \sim P$, i.e. the distribution from which we observe samples. On a finite sample $s_1^n$ we have

$$\Upsilon(x, \eta, \nu; s_1^n) := \Upsilon(x, \eta, \nu; \widehat{P}[s_1^n]) = \frac{1}{n}\sum_{i \leq n}(\nu\phi + \lambda\psi)^*[\ell(x; s_i) - \eta] + \eta + \nu\rho.$$

For pure-constraint objectives (with $\psi = 0$), $\Upsilon$ simplifies to

$$\psi = 0 \implies \Upsilon(x, \eta, \nu; P) = \nu\,\mathbb{E}_{S \sim P}\,\phi^*\left[\frac{\ell(x; S) - \eta}{\nu}\right] + \eta + \nu\rho. \tag{18}$$

For pure-penalty objective (with $\phi = 0$) the Lagrange multiplier $\nu$ is unnecessary and we have

$$\phi = 0 \implies \Upsilon(x, \eta; P) = \lambda\,\mathbb{E}_{S \sim P}\,\psi^*\left[\frac{\ell(x; S) - \eta}{\lambda}\right] + \eta. \tag{19}$$

Note that $\Upsilon$ is an expectation (i.e., an empirical risk) which means that to minimize $\mathcal{L}(x; P)$ we can, in principle, apply ERM jointly on $x, \eta$ and $\nu$, as we further discuss in Appendix A.3.

Finally, we note that any $Q^\star$ attaining the supremum in (12) is of the form

$$\frac{\mathrm{d}Q^\star}{\mathrm{d}P}(s) = (\nu^\star + \lambda\psi)^{*\prime}[\ell(x; s) - \eta^\star].$$

where $\eta^\star$ and $\nu^\star$ are optimal Lagrange multipliers in (16) and $(\nu^\star + \lambda\psi)^{*\prime}$ is a subderivative of $(\nu^\star + \lambda\psi)^*$. For $\phi = 0$ this specializes to

$$\frac{\mathrm{d}Q^\star}{\mathrm{d}P}(s) = \psi^{*\prime}\left[\frac{\ell(x; s) - \eta^\star}{\lambda}\right].$$

For a finite sample, we have

$$q_i^\star = \frac{1}{n}\psi^{*\prime}\left[\frac{\ell(x; s_i) - \eta^\star}{\lambda}\right]. \tag{20}$$

### A.1.3 Expressions for CVaR

Recall that CVaR at level $\alpha$ corresponds to $\phi = 0$ and $\psi = \mathbb{I}_{[0, 1/\alpha)}$. The dual expression of CVaR simplifies to [67, Example 6.16]

$$\mathcal{L}_{\mathrm{CVaR}}(x; P) = \inf_{\eta \in \mathbb{R}}\left\{\frac{1}{\alpha}\,\mathbb{E}_{S \sim P}(\ell(x; S) - \eta)_+ + \eta\right\}.$$

It also has a simple closed-form expression in terms of the inverse cdf of $\ell(x; S)$ [67, Theorem 6.2]:

$$\mathcal{L}_{\mathrm{CVaR}}(x; P) = \frac{1}{\alpha}\int_{1-\alpha}^1 F^{-1}(u)\mathrm{d}u. \tag{21}$$

We note that this last expression is a direct consequence of (14), since $\mathcal{R}$ is the set of measures never exceeding $\frac{1}{\alpha}$. On a finite sample $s_1^n$ this gives the closed-form expression

$$\mathcal{L}_{\mathrm{CVaR}}(x; s_1^n) = \frac{1}{\alpha n}\sum_{i=1}^{\lfloor \alpha n \rfloor}\ell(x; s_{(i)}) + \left(1 - \frac{\lfloor \alpha n \rfloor}{\alpha n}\right)\ell(x; s_{(\lfloor \alpha n \rfloor + 1)}), \tag{22}$$

where $s_{(1)}, \ldots, s_{(n)}$ are a permutation of $s_1^n$ satisfying $\ell(x; s_{(1)}) \geq \ell(x; s_{(2)}) \geq \cdots \geq \ell(x; s_{(n)})$. For $\alpha \leq 1/n$ we simply have $\mathcal{L}_{\mathrm{CVaR}}(x; s_1^n) = \max_{i \leq n} \ell(x; s_i)$.

The KL-divergence penalized CVaR at level $\alpha$ corresponds to $\psi(t) = \mathbb{I}_{[0, 1/\alpha]}(t) + t\log t - t + 1$, for which

$$\psi^*[v] = \begin{cases} e^v - 1 & v < \log\frac{1}{\alpha} \\ \frac{1}{\alpha} - 1 + \frac{1}{\alpha}(v - \log\frac{1}{\alpha}) & \text{otherwise,} \end{cases}$$

and the dual expression for $\mathcal{L}_{\mathrm{kl\text{-}CVaR}}$ is given by (19). In the special case $\alpha \leq 1/n$ the CVaR constraint becomes inactive, and we can minimize over $\eta$ in closed form to obtain the standard "soft max" objective $\mathcal{L}_{\mathrm{kl\text{-}CVaR}}(x; s_1^n) = \lambda\log\left(\frac{1}{n}\sum_{i \leq n}\exp(\ell(x; s_i)/\lambda)\right)$.

### A.1.4  Expressions for $\mathcal{L}_{\chi^2\text{-pen}}$ and $\mathcal{L}_{\chi^2}$

The penalized version of the $\chi^2$ objective corresponds to $\phi(t) = 0$ and $\psi(t) = \frac{1}{2}(t-1)^2$. Note that $D_\phi(Q, P)$ is invariant under $\psi(t) \mapsto \psi(t) + c \cdot (t-1)$ for any $c \in \mathbb{R}$ because $\int (\frac{dQ}{dP} - 1)dP = 0$. We find it more convenient to work with $\psi(t) = \frac{1}{2}(t-1)^2 + (t-1) = \frac{1}{2}(t^2 - 1)$, for which the conjugate is simply $\psi^*[v] = \frac{1}{2}((v)_+^2 + 1)$. The dual form (19) gives

$$\mathcal{L}_{\chi^2\text{-pen}}(x; P) = \inf_{\eta \in \mathbb{R}} \left\{ \frac{1}{2\lambda} \mathbb{E}_{S \sim P} \left(\ell(x; S) - \eta\right)_+^2 + \frac{\lambda}{2} + \eta \right\}. \tag{23}$$

The infimum is attained at the $\eta^\star$ solving $\mathbb{E}(\ell(x; S) - \eta^\star)_+ = \lambda$. In other words,

$$\eta^\star = \mathbb{E}[\ell(x; S) \mid \ell(x; S) \geq \eta^\star] - \frac{\lambda}{\mathbb{P}(\ell(x; S) \geq \eta^\star)} = \mathcal{L}_{\text{CVaR}}^{F(\eta^\star)}(x; P) - \frac{\lambda}{1 - F(\eta^\star)},$$

where $F(t) = \mathbb{P}(\ell(x; S) \leq t)$ is the cdf of $\ell(x; S)$. Letting $\mathfrak{G}(\eta^\star)$ denote the event that $\ell(x; S) \geq \eta^\star$, substituting back to the expression for $\mathcal{L}_{\chi^2\text{-pen}}$ gives

$$\mathcal{L}_{\chi^2\text{-pen}}(x; P) = \mathbb{E}[\ell(x; S) \mid \mathfrak{G}(\eta^\star)] + \frac{1}{2\lambda}\text{Var}[\ell(x; S) \mid \mathfrak{G}(\eta^\star)] + \frac{\lambda}{2}\left(\frac{1}{\mathbb{P}(\mathfrak{G}(\eta^\star))} - 1\right)^2.$$

In words, $\mathcal{L}_{\chi^2\text{-pen}}$ is a sum of a CVaR (at level $F(\eta^\star)$), a conditional variance regularization term and an outage probability regularization term. This expression simplifies considerably when $\lambda$ is sufficiently large. Specifically, we have,

$$\begin{aligned} \lambda \geq B &\implies \lambda \geq \mathbb{E}\,\ell(x; S) - F^{-1}(0) \\ &\implies \eta^\star = \mathbb{E}\,\ell(x; S) - \lambda \ \text{ and } \ \mathbb{P}(\mathfrak{G}(\eta^\star)) = 1 \\ &\implies \mathcal{L}_{\chi^2\text{-pen}}(x; P) = \mathbb{E}\,\ell(x; S) + \frac{1}{2\lambda}\text{Var}[\ell(x; S)]. \end{aligned} \tag{24}$$

That is, for sufficiently large $\lambda$ the objective $\mathcal{L}_{\chi^2\text{-pen}}$ is simply the empirical risk with variance regularization (see also [22]).

For a finite sample we have

$$\mathcal{L}_{\chi^2\text{-pen}}(x; s_1^n) = \frac{1}{2\lambda n} \sum_{i \leq n} (\ell(x; s_i) - \eta_n^\star)_+^2 + \frac{\lambda}{2} + \eta_n^\star.$$

Where $\eta_n^\star$ is the solution to $\sum_{i \leq n} (\ell(x; s_i) - \eta_n^\star)_+ = n\lambda$, or equivalently

$$\eta_n^\star = \frac{1}{i^\star} \sum_{i \leq i^\star} \ell(x; s_{(i)}) - \frac{\lambda n}{i^\star} \ \text{ for the unique } i^\star \text{ such that } \ \ell(x; s_{(i^\star+1)}) \leq \eta_n^\star \leq \ell(x; s_{(i^\star)}), \tag{25}$$

where $\{\ell(x; s_{(i)})\}$ are the sorted $\{\ell(x; s_i)\}$ and $\ell(x; s_{(n+1)}) := -\infty$.

An expression for $\mathcal{L}_{\chi^2}$ follows via (23)

$$\mathcal{L}_{\chi^2}(x; P) = \inf_{\lambda \geq 0} \left\{ \mathcal{L}_{\chi^2\text{-pen}}(x; P) + \lambda\rho \right\} = \inf_{\eta \in \mathbb{R}} \left\{ \sqrt{1 + 2\rho}\sqrt{\mathbb{E}_{S \sim P}\left(\ell(x; S) - \eta\right)_+^2} + \eta \right\}, \tag{26}$$

and the maximizing $Q$ is

$$\frac{dQ^\star}{dP}(s) = \frac{(\ell(x; s) - \eta^\star)_+}{\mathbb{E}_{S \sim P}(\ell(x; S) - \eta^\star)_+}. \tag{27}$$

### A.1.5  Expression for $\nabla\mathcal{L}$

Let $Q^\star$ by a distribution attaining the supremum in (12) and recall that $\nabla\ell(x; s)$ denotes an element in the sub-differential of $\ell(x; s)$ w.r.t. $x$. Then the following vector is a subgradient of $\mathcal{L}$ [37, Corollary 4.4.4],

$$\nabla\mathcal{L}(x; P) = \mathbb{E}_{S \sim Q^\star} \nabla\ell(x; S).$$

Similarly, for a sample of size $n$ and a maximizing $q^\star$, we have

$$\nabla\mathcal{L}(x; s_1^n) = \sum_{i \leq n} q_i^\star \nabla\ell(x; s_i). \tag{28}$$

### A.1.6 Smoothness of $\mathcal{L}_{\chi^2\text{-pen}}$ and $\mathcal{L}_{\text{kl-CVaR}}$

The smoothness of $\mathcal{L}$ (i.e., Lipschitz continuity of its gradient) plays a role in our mini-batch gradient estimator complexity guarantees. When the penalty term $\psi$ is strongly convex, the maximizing $Q^\star$ (or $q^\star$) is unique, and if $\nabla\ell$ is $H$-Lipschitz then $\mathcal{L}$ is differentiable [37, Corollary 4.4.5]. In particular, writing $Q_x^\star$ for the maximizing $Q$ at point $x$, we have

$$
\begin{aligned}
\|\nabla\mathcal{L}(x;P) - \nabla\mathcal{L}(y;P)\| &= \left\|\int\left\{\nabla\ell(x;s)\mathrm{d}Q_x^\star(y) - \nabla\ell(y;s)\mathrm{d}Q_y^\star(s)\right\}\right\| \\
&\leq \int\|\nabla\ell(x;s) - \nabla\ell(y;s)\|\mathrm{d}Q_x^\star(s) + \int\|\nabla\ell(y;s)\|\left|\frac{\mathrm{d}Q_x^\star}{\mathrm{d}P}(s) - \frac{\mathrm{d}Q_y^\star}{\mathrm{d}P}(s)\right|\mathrm{d}P(s) \\
&\leq H\|x-y\| + G\|Q_x^\star - Q_y^\star\|_1.
\end{aligned}
\tag{29}
$$

Therefore, if $Q_x^\star$ is Lipschitz w.r.t. $x$ in the 1-norm then $\nabla\mathcal{L}$ is Lipschitz as well. This is indeed the case $\mathcal{L}_{\chi^2\text{-pen}}$ and $\mathcal{L}_{\text{kl-CVaR}}$.

**Claim 1.** *Let Assumption A1 hold. For all $P$, $\nabla\mathcal{L}_{\text{kl-CVaR}}(x;P)$ and $\nabla\mathcal{L}_{\chi^2\text{-pen}}(x;P)$ are $(\frac{G^2}{\lambda} + H)$-Lipschitz in $x$, and $0 \leq \mathcal{L}_{\text{CVaR}}(x;P) - \mathcal{L}_{\text{kl-CVaR}}(x;P) \leq \lambda\log(1/\alpha)$ for all $x$.*

*Proof.* Since entropy is 1-strongly-convex w.r.t. the 1-norm, for $\mathcal{L}_{\text{kl-CVaR}}$ we have that the penalty $\lambda\psi$ is $\lambda$-strongly-convex w.r.t. the 1-norm and therefore [63, Lemma 2]

$$
\|Q_x^\star - Q_y^\star\|_1 \leq \frac{1}{\lambda}\|\ell(x;\cdot) - \ell(y;\cdot)\|_\infty \leq \frac{G}{\lambda}\|x-y\|,
$$

which by (29) implies that $\nabla\mathcal{L}_{\text{kl-CVaR}}$ is $(H + G^2/\lambda)$-Lipschitz as required. For $\mathcal{L}_{\chi^2\text{-pen}}$, we find it easier to argue for a finite sample $s_1^n$. By (13) we have $q_x^\star = \arg\max_{q\in\Delta^n}\left\{q^\top\ell(x) - \frac{1}{2}\lambda n\|q\|_2^2\right\}$, where $\ell_i(x) = \ell(x;s_i)$. Therefore, by $\lambda n$-strong-convexity w.r.t. the 2-norm, we have

$$
\|q_x^\star - q_y^\star\|_1 \leq \sqrt{n}\|q_x^\star - q_y^\star\|_2 \leq \frac{1}{\lambda\sqrt{n}}\|\ell(x) - \ell(y)\|_2 \leq \frac{G}{\lambda}\|x-y\|,
$$

establishing that $\nabla\mathcal{L}_{\chi^2\text{-pen}}$ is also $(H + G^2/\lambda)$-Lipschitz.

Finally, we note that $\mathcal{L}_{\text{kl-CVaR}}(x;P) \leq \mathcal{L}_{\text{CVaR}}$ because $D_\psi(Q,P) \geq 0$ for all $Q$. Conversely since any feasible $Q$ satisfies $\mathrm{d}Q/\mathrm{d}P \leq 1/\alpha$ we have $D_\psi(Q,P) = \int\mathrm{d}Q\log\frac{\mathrm{d}Q}{\mathrm{d}P} \leq \log\frac{1}{\alpha}$ and therefore $\mathcal{L}_{\text{kl-CVaR}}(x;P) \geq \mathcal{L}_{\text{CVaR}}(x;P) - \lambda\log\frac{1}{\alpha}$. $\qquad\square$

### A.2 Computational cost

To compute $\mathcal{L}(x;s_1^n)$ and its (sub)gradient from $\{\ell(x;s_i)\}_{i\leq n}$ and $\{\nabla\ell(x;s_i)\}_{i\leq n}$ we compute $q^\star$ that maximizes (13) and substitute it back in (28). The substitution requires $\tilde{O}(nd)$ work, so it remains to account for the work in computing $q^\star$.

For CVaR, this clearly amounts to sorting $\{\ell(x;s_i)\}_{i\leq n}$ and therefore takes $O(n\log n)$ time. Similarly, for $\mathcal{L}_{\chi^2\text{-pen}}$ we may find sort the losses and find $i^\star$ in (25), and hence $\eta_n^\star$ and $q^\star$, in $O(n)$ time. Alternatively, for any objective with $\phi = 0$ (including $\mathcal{L}_{\chi^2\text{-pen}}$ and $\mathcal{L}_{\text{kl-CVaR}}$ we can bisect directly on $\eta$, either to minimize the expression (19) or to satisfy the the simplex constraint $\sum_{i\leq n}q_i^\star = \frac{1}{n}\sum_{i\leq n}(\psi^*)'[(\ell(x;s_i) - \eta^\star)/\lambda] = 1$.

For $\mathcal{L}_{\chi^2}$ we may find $q^\star$ by performing similar bisection over $\eta$ via the expression (26), again either minimizing it or solving for the condition $\frac{1}{n}\sum_{i\leq n}(\ell(x;s_i) - \eta^\star)_+^2 = (1 + 2\rho)\left(\frac{1}{n}\sum_{i\leq n}(\ell(x;s_i) - \eta^\star)_+\right)^2$. Finding an $\varepsilon$ accurate solution via bisection requires roughly $n\log\frac{B}{\varepsilon}$ time.

Since we are interested in large-scale application, we assume that $d \gg \log(nB/\varepsilon)$ and therefore the time to compute the objective and its gradient is $O(nd)$.

For simplicity and stability, our code implements the computation of $q^\star$ using bisection over $\eta$ for each of $\mathcal{L}_{\chi^2\text{-pen}}, \mathcal{L}_{\chi^2}$ and $\mathcal{L}_{\text{kl-CVaR}}$.

### A.3 Stochastic gradient method on the dual objective

Here we discuss the convergence guarantees for a simple stochastic gradient method using the dual expression (16) for $\mathcal{L}(x; P_0)$ in order to minimize it over $x$. While several works consider such methods (see Section 1.1), we could not find direct reference for their runtime guarantees, and we therefore briefly derive it below.

Focusing on objectives with $\phi = 0$ (as in (19)), and writing $\gamma_x$ and $\gamma_\eta$ for step sizes, we write the iterations on $x$ and the Lagrange multiplier $\eta$ as

$$x_{t+1} = \Pi_{\mathcal{X}}(x_t - \gamma_x \nabla \Upsilon(x_t, \eta_t; P_0)) = \Pi_{\mathcal{X}}\left(x_t - \gamma_x \psi^{*\prime}\left[\frac{\ell(x; S_i) - \eta_t}{\lambda}\right]\nabla \ell(x; S_i)\right), \text{ and}$$

$$\eta_{t+1} = \Pi_{[\underline{\eta}, \overline{\eta}]}\left(\eta_t - \gamma_\eta \frac{\partial}{\partial \eta}\Upsilon(x_t, \eta_t; P_0)\right) = \Pi_{[\underline{\eta}, \overline{\eta}]}\left(\eta_t + \gamma_\eta \psi^{*\prime}\left[\frac{\ell(x; S_i) - \eta_t}{\lambda}\right] - \gamma_\eta\right), \quad (30)$$

Where $S_1, S_2, \ldots$ are drawn iid from $P_0$.

For CVaR, we have $(\psi^*)'[v] = \frac{1}{\alpha}1_{\{v \geq 0\}}$ and we may restrict $\eta$ to the range $[\underline{\eta}, \overline{\eta}] = [0, B]$, as the optimal $\eta$ is the value at risk level $\alpha$ and therefore in the range of $\ell$. For $\mathcal{L}_{\chi^2\text{-pen}}$ we have $(\psi^*)'[v] = (v)_+$ and we may take $[\underline{\eta}, \overline{\eta}] = [-\lambda, B]$ due to the condition $\mathbb{E}(\ell(x; S) - \eta^\star)_+ = \lambda$. In these settings, the method (30) has the following guarantee

**Claim 3.** *Let $\epsilon \in (0, B)$. For CVaR and a suitable choice of $\gamma_x, \gamma_\eta$ the average iterate $\bar{x}_T = \frac{1}{T}\sum_{t \leq T} x_t$ satisfies*

$$\mathbb{E}\,\mathcal{L}_{\mathrm{CVaR}}(\bar{x}_T; P_0) - \min_{x' \in \mathcal{X}} \mathcal{L}_{\mathrm{CVaR}}(x'; P_0) \leq \epsilon \text{ for } T \asymp \frac{(GR)^2 + B^2}{\alpha^2 \epsilon^2}.$$

*Similarly, for $\chi^2$ penalty we have*

$$\mathbb{E}\,\mathcal{L}_{\chi^2\text{-pen}}(\bar{x}_T; P_0) - \min_{x' \in \mathcal{X}} \mathcal{L}_{\chi^2\text{-pen}}(x'; P_0) \leq \epsilon \text{ for } T \asymp \frac{(GR)^2 + B^2}{\epsilon^2}\left(1 + \frac{B^2}{\lambda^2}\right).$$

*Proof.* By Proposition 6, the expected sub-optimality of $\bar{x}_T$ is $\lesssim (\Gamma_x R + \Gamma_\eta(\overline{\eta} - \underline{\eta}))/\sqrt{T}$, where $\Gamma_x^2$ (respectively $\Gamma_\eta$) is an upper bound on the second moment of $\nabla \Upsilon(x, \eta; S)$ (respectively $\frac{\partial}{\partial \eta}\Upsilon(x, \eta; S)$). For $\mathcal{L}_{\mathrm{CVaR}}$ we have $\Gamma_x \leq G/\alpha$, $\Gamma_\eta \leq 1/\alpha$ and $\overline{\eta} - \underline{\eta} = B$. For $\mathcal{L}_{\chi^2\text{-pen}}$ we have $\Gamma_x \leq G(1 + B/\lambda)$, $\Gamma_\eta = 1 + B/\lambda$ and $\overline{\eta} - \underline{\eta} = B + \lambda$. The result follows from substituting $T \asymp (\Gamma_x^2 R^2 + \Gamma_\eta^2(\overline{\eta} - \underline{\eta})^2)\epsilon^{-2}$. $\square$

### A.4 Uncertainty sets contained in $\chi^2$ divergence balls

A number of our results hold for general subclass of the objective (12) with the following property.

**Definition 1** ($\chi^2$-bounded objective)**.** *An objective $\mathcal{L}(x; P_0)$ is $C$-$\chi^2$-bounded if for all $x$ and all $Q^\star$ attaining the supremum in (12) we have $\mathrm{D}_{\chi^2}(Q^\star, P_0) \leq C$.*

The three objectives we focus on are $\chi^2$-bounded.

**Claim 4.** *The objectives $\mathcal{L}_{\mathrm{kl\text{-}CVaR}}$, $\mathcal{L}_{\chi^2}$ and $\mathcal{L}_{\chi^2\text{-pen}}$ are $\chi^2$-bounded with constants $C = \frac{1}{\alpha} - 1$, $C = \rho$ and $C = B/\lambda$, respectively.*

*Proof.* That $\mathcal{L}_{\chi^2}$ is $\rho$-$\chi^2$-bounded is obvious from definition. For $\mathcal{L}_{\chi^2\text{-pen}}$ we have

$$\mathcal{L}_{\chi^2\text{-pen}}(x; P_0) = \mathbb{E}_{S \sim Q^\star}\, \ell(x; S) - \lambda \mathrm{D}_{\chi^2}(Q^\star, P_0)$$
$$\geq \mathbb{E}_{S \sim P_0}\, \ell(x; S) - \lambda \mathrm{D}_{\chi^2}(P_0; P_0) = \mathbb{E}_{S \sim P_0}\, \ell(x; S)$$

and consequently

$$\mathrm{D}_{\chi^2}(Q^\star, P_0) \leq \frac{\mathbb{E}_{S \sim Q^\star}\ell(x; S) - \mathbb{E}_{S \sim P_0}\, \ell(x; S)}{\lambda} \leq \frac{B}{\lambda}.$$

Finally, for $\mathcal{L}_{\mathrm{kl\text{-}CVaR}}$ every feasible $Q$ satisfies $dQ/dP_0 \leq 1/\alpha$ and therefore

$$\mathrm{D}_{\chi^2}(Q, P_0) = \int \left(\frac{dQ}{dP_0}(s)\right)^2 dP_0(s) - 1 \leq \frac{1}{\alpha}\int \left(\frac{dQ}{dP_0}(s)\right)dP_0(s) - 1 = \frac{1}{\alpha} - 1.$$

$\square$

### A.5 General results

We conclude this section of the appendix by stating three general results that aid our analysis. First, we give a lemma stating that a binomial random variable with parameters $n$ and $\alpha$ has a constant probability of being at least $\sqrt{\alpha(1-\alpha)n}$ below its mean.

**Lemma 1.** *Let $n \in \mathbb{N}$ and $\alpha \in (0,1)$. There exists a numerical constant $C \in \mathbb{R}$ such that*

$$\mathbb{P}\big(\mathsf{Bin}(n,\alpha) \leq n\alpha - \sqrt{n\alpha(1-\alpha)}\big) \geq \mathbb{P}(\mathcal{N}(0,1) \leq -1) - \frac{C}{\sqrt{\alpha(1-\alpha)n}}.$$

*Proof.* Note that $\mathbb{P}\big(\mathsf{Bin}(n,\alpha) \leq n\alpha - \sqrt{n\alpha(1-\alpha)}\big) = \mathbb{P}(Y\sqrt{n} \leq -1)$ where $Y = \frac{1}{n}\frac{\mathsf{Bin}(n,\alpha)-n\alpha}{\sqrt{\alpha(1-\alpha)}}$ is the mean of $n$ independent random variable with zero mean, unit variance, and absolute third moment $\rho = \frac{\alpha^2 + (1-\alpha)^2}{\sqrt{\alpha(1-\alpha)}} \leq \frac{1}{\sqrt{\alpha(1-\alpha)}}$. The Berry-Esseen theorem [25, Theorem 3.4.17] states that for such $Y$ we have $|\mathbb{P}(Y\sqrt{n} \leq t) - \mathbb{P}(\mathcal{N}(0,1) \leq t)| \leq C\rho/\sqrt{n}$, for all $t \in \mathbb{R}$; substituting $t = 1$ and $\rho \leq \frac{1}{\sqrt{\alpha(1-\alpha)}}$ concludes the proof. $\qquad\square$

Second, we state the Efron-Stein inequality in vector form, which follows from applying the standard scalar bound element-wise.

**Lemma 2** (Efron-Stein inequality [9, Theorem 3.1])**.** *Let $X_1^{n+1}$ be i.i.d random variables and $f : \mathcal{X}^n \to \mathbb{R}^m$. Let $I$ be uniform on $\{1, \dots, n\}$ and let $\tilde{X}_1^N$ be such that $\tilde{X}_i = X_i$ for $i \neq I$ and $\tilde{X}_I = X_{n+1}$. Then*

$$\mathrm{Var}[f(X_1^n)] \leq \frac{n}{2}\,\mathbb{E}\,\|f(X_1^n) - f(\tilde{X}_1^n)\|^2. \tag{31}$$

Third, we give a general lemma on the variance of sampling without replacement, which we specialize to the simplex for later use.

**Lemma 3.** *Let $p \in \Delta^k$ and let $\mathcal{I}$ be a random subset of $[k]$ of size $k/2$. Then*

$$\mathbb{E}\left(\sum_{i \in \mathcal{I}} p_i - \frac{1}{2}\right)^2 \leq \frac{1}{2}\left\|p - \tfrac{1}{k}\mathbf{1}\right\|^2 = \frac{1}{2k}\mathrm{D}_{\chi^2}(p, \tfrac{1}{k}\mathbf{1}).$$

*Proof.* Let us denote $q = p - \frac{1}{k}\mathbf{1}$. We have

$$
\mathbb{E}\left(\sum_{i \leq k} p_i \mathbf{1}_{\{i \in \mathcal{I}\}} - \frac{1}{2}\right)^2 = \mathbb{E}\left(\sum_{i \leq k} q_i \mathbf{1}_{\{i \in \mathcal{I}\}}\right)^2
$$

$$
\overset{(i)}{=} \frac{1}{2}\sum_{i \leq k} q_i^2 + \sum_{i \neq j} q_i q_j\,\mathbb{E}\,\mathbf{1}_{\{i \in \mathcal{I} \text{ and } j \in \mathcal{I}\}}
$$

$$
\overset{(ii)}{=} \frac{1}{2}\|q\|^2 + \frac{k-2}{4(k-1)}\sum_{i \leq k}\sum_{j \neq i} q_i q_j = \frac{1}{2}\|q\|^2 + \frac{k-2}{4(k-1)}\sum_{i \leq k} q_i(1-q_i)
$$

$$
\overset{(iii)}{=} \left(\frac{1}{2} - \frac{k-2}{4(k-1)}\right)\|q\|^2 \leq \frac{1}{2}\|q\|^2,
$$

where $(i)$ stems from $\mathbb{P}(i \in \mathcal{I}) = \frac{1}{2}$, $(ii)$ from $\mathbb{P}(i \in I \text{ and } j \in I) = \frac{1}{2}\frac{k/2-1}{k-1}$ and $(iii)$ from $\sum_{i \leq k} q_i = 0$. Noting that $\mathrm{D}_{\chi^2}(p, \frac{1}{k}\mathbf{1}) = k\|q\|^2$ concludes the proof. $\qquad\square$

## B  Proofs from Section 3

This section completes the proof and discussion of the results in Section 3. First, in Section B.1, we prove the bias bounds in Proposition 1 and argue their tightness in the worst case. Section B.2 provides additional discussion of the smoothness and Lipschitz inverse-cdf assumptions sometimes used in this section. Then, in Section B.3 we bound the variance of the mini-batch estimators for

$\chi^2$-bounded penalty objectives and their gradient, obtaining Proposition 2 as a corollary. We also argue that similar bounds do not hold for the $\chi^2$ constraint objective. In Section B.4 we review the standard convergence guarantees for stochastic gradients iterations with and without Nesterov acceleration, and in Section B.5 we combine all these ingredients to prove Theorem 1.

## B.1  Bias of batch estimator

### B.1.1  Proof of Proposition 1

**Proposition 1** (Bias of the batch estimator). *For all $x \in \mathcal{X}$ and $n \in \mathbb{N}$ we have*

$$0 \leq \mathcal{L}(x; P_0) - \overline{\mathcal{L}}(x; n) \lesssim \begin{cases} B \min\{1, (\alpha n)^{-1/2}\} & \text{for } \mathcal{L} = \mathcal{L}_{\mathrm{CVaR}} & (5) \\ B\sqrt{(1+\rho)(\log n)/n} & \text{for } \mathcal{L} = \mathcal{L}_{\chi^2} & (6) \\ B^2(\lambda n)^{-1} & \text{for } \mathcal{L} = \mathcal{L}_{\chi^2\text{-pen}} & (7) \\ G_{\mathrm{icdf}}\, n^{-1} & \text{for any loss (2),} & (8) \end{cases}$$

*where the bound* (8) *holds under Assumption A1.*

*Proof.* We first show that the bound $\mathcal{L} \geq \overline{\mathcal{L}}$ holds for any loss of the form (12) and then proceed to show each of the bounds (5)–(8). We remark here that the bound (6) actually holds for any $\rho$-$\chi^2$-bounded objective (Definition 1).

**Proof of $\mathcal{L}(x; P_0) \geq \overline{\mathcal{L}}(x; n)$.** The dual expression (17) gives
$$\mathcal{L}(x; P_0) = \inf_{\eta \in \mathbb{R}, \nu \geq 0} \mathbb{E}_{S \sim P_0}\{(\nu\phi + \lambda\psi)^*[\ell(x; S) - \eta] + \eta + \nu\rho\}$$

$$= \inf_{\eta \in \mathbb{R}, \nu \geq 0} \mathbb{E}_{S_1^n \sim P_0^n}\left\{\frac{1}{n}\sum_{i \leq n}(\nu\phi + \lambda\psi)^*[\ell(x; S_i) - \eta] + \eta + \nu\rho\right\}$$

$$\geq \mathbb{E}_{S_1^n \sim P_0^n} \inf_{\eta \in \mathbb{R}, \nu \geq 0}\left\{\frac{1}{n}\sum_{i \leq n}(\nu\phi + \lambda\psi)^*[\ell(x; S_i) - \eta] + \eta + \nu\rho\right\} = \mathbb{E}\,\mathcal{L}(x; S_1^n) = \overline{\mathcal{L}}(x; n),$$

where the inequality follows from exchanging the expectation and the infimum.

**Proof of the CVaR bias bound** (5)**.** By Eq. (22) we have

$$\overline{\mathcal{L}}_{\mathrm{CVaR}}(x; n) = \mathbb{E}\,\mathcal{L}_{\mathrm{CVaR}}(x; S_1^n) = \frac{1}{\alpha n}\sum_{i=1}^{\lfloor \alpha n \rfloor} \mathbb{E}\,\ell(x; S_{(i)}) + \left(1 - \frac{\lfloor \alpha n \rfloor}{\alpha n}\right)\mathbb{E}\,\ell(x; S_{(\lfloor \alpha n \rfloor + 1)}),$$

where $\ell(x; S_{(i)})$ is the $i$th order statistic of $\ell(x; S_1^n)$ (in decreasing order). Recalling that $F$ denotes the cdf of $\ell(x; S)$, we may write $\ell(x; S) = F^{-1}(U)$ with $U$ uniform on $[0, 1]$. Therefore, $\ell(x; S_{(i)}) = F^{-1}(U_{(i)})$ where $U_{(i)} \sim \mathsf{Beta}(n - i + 1, i)$ is the $i$th order statistic of $n$ iid $\mathsf{Unif}([0, 1])$ random variables [57, Sec. 4.6]. Taking expectation, we have

$$\mathbb{E}_{S_1^n \sim P_0^n}\,\ell(x; S_{(i)}) = \int_0^1 F_Z^{-1}(u) f_{\mathsf{Beta}(n-i+1,i)}(u)\mathrm{d}u,$$

where $f_{\mathsf{Beta}(a,b)}$ is the density function of the Beta random variable of parameters $a, b$. Substituting back, we have

$$\overline{\mathcal{L}}_{\mathrm{CVaR}}(x; n) = \frac{1}{\alpha}\int_0^1 \mathcal{I}_\alpha(u)F^{-1}(u)\mathrm{d}u, \quad \text{where}$$

$$\mathcal{I}_\alpha(u) = \frac{1}{n}\sum_{i=1}^{\lfloor \alpha n \rfloor} f_{\mathsf{Beta}(n-i+1,i)}(u) + \left(\alpha - \frac{\lfloor \alpha n \rfloor}{n}\right) f_{\mathsf{Beta}(n-\lfloor \alpha n \rfloor, \lfloor \alpha n \rfloor + 1)}(u). \quad (32)$$

Using

$$\begin{aligned} \frac{1}{n}\sum_{i=1}^n f_{\mathsf{Beta}(n-i+1,i)}(u) &= \frac{1}{n}\sum_{i=1}^n \frac{n!}{(n-i)!(i-1)!}u^{n-i}(1-u)^{i-1} \\ &= \sum_{i=0}^{n-1}\binom{n-1}{i-1}(1-u)^i u^{n-1-i} = 1, \end{aligned} \quad (33)$$

we have that

$$1 - \mathcal{I}_\alpha(u) \leq \frac{1}{n} \sum_{i=\lfloor \alpha n \rfloor + 1}^{n} f_{\mathsf{Beta}(n-i+1,i)}(u).$$

Recalling Eq. (21) for $\mathcal{L}_{\mathsf{CVaR}}(x; P_0)$, and recalling that $F^{-1}(u) \in [0, B]$ for all $u$ by assumption, we bound the bias as

$$
\begin{aligned}
\mathcal{L}_{\mathsf{CVaR}}(x; P_0) - \overline{\mathcal{L}}_{\mathsf{CVaR}}(x; n) &= \frac{1}{\alpha} \int_0^1 \left[ 1_{\{u \geq 1-\alpha\}} - \mathcal{I}_\alpha(u) \right] F^{-1}(u) \mathrm{d}u \\
&\leq \frac{B}{\alpha} \int_{1-\alpha}^1 \left[ 1_{\{u \geq 1-\alpha\}} - \mathcal{I}_\alpha(u) \right] \mathrm{d}u \\
&\leq \frac{B}{\alpha} \int_{1-\alpha}^1 \left( \frac{1}{n} \sum_{i=\lfloor \alpha n \rfloor + 1}^{n} f_{\mathsf{Beta}(n-i+1,i)}(u) \right) \mathrm{d}u \\
&= \frac{B}{\alpha n} \sum_{i=\lfloor \alpha n \rfloor + 1}^{n} \mathbb{P}(\mathsf{Beta}(n-i+1,i) \geq 1 - \alpha). \qquad (34)
\end{aligned}
$$

To conclude, it suffices to bound the tail probability of the Beta random variables. We have [see, e.g., 57, Ex. 5 in Sec. 4.6]

$$
\begin{aligned}
\mathbb{P}(\mathsf{Beta}(n-i+1,i) \geq 1 - \alpha) &= 1 - \mathbb{P}(\mathsf{Beta}(n-i+1,i) \leq 1 - \alpha) \\
&= \mathbb{P}(\mathsf{Bin}(n; 1-\alpha) \leq n - i) = \mathbb{P}(\mathsf{Bin}(n; \alpha) \geq i),
\end{aligned}
$$

and the multiplicative Chernoff bound [46, Theorem 4.3] gives

$$\mathbb{P}(\mathsf{Bin}(n; \alpha) \geq i) \leq \exp\left( -\frac{i - n\alpha}{3} \min\left\{ \frac{i - n\alpha}{n\alpha}, 1 \right\} \right).$$

Therefore, for $\alpha n \geq 9$,

$$
\begin{aligned}
\sum_{i=\lfloor \alpha n \rfloor + 1}^{n} \mathbb{P}(\mathsf{Bin}(n; \alpha) \geq i) &\leq \sum_{i=\lfloor \alpha n \rfloor + 1}^{2\lfloor \alpha n \rfloor} \exp\left( -\frac{(i - n\alpha)^2}{3n\alpha} \right) + \sum_{i=2\lfloor \alpha n \rfloor + 1}^{\infty} \exp\left( -\frac{i - n\alpha}{3} \right) \\
&\leq 1 + \int_0^\infty \exp\left( -\frac{u^2}{3n\alpha} \right) \mathrm{d}u + \exp\left( -\frac{2\lfloor \alpha n \rfloor + 1 - \alpha n}{3} \right) \sum_{i=0}^{\infty} e^{-i/3} \\
&\leq 1 + \frac{\sqrt{3\pi\alpha n}}{2} + \frac{e^{-3}}{1 - e^{-1/3}} \leq 3\sqrt{\alpha n}.
\end{aligned}
$$

Substituting into (34) and using $\mathcal{L}_{\mathsf{CVaR}}(x; P_0) \leq B$ when $\alpha n \leq 9$ gives the final bound

$$\mathcal{L}_{\mathsf{CVaR}}(x; P_0) - \overline{\mathcal{L}}_{\mathsf{CVaR}}(x; n) \leq B \min\left\{ \frac{3}{\sqrt{\alpha n}}, 1 \right\}. \qquad (35)$$

**Proof of the bound (6).** We start with the expression (14) specialized for the $\mathcal{L}_{\chi^2}$,

$$\mathcal{L}_{\chi^2}(x; P_0) = \sup_{r \in \mathcal{R}} \int_0^1 r(\beta) F^{-1}(1 - \beta) \mathrm{d}\beta,$$

where

$$\mathcal{R} = \left\{ r : [0, 1] \to \mathbb{R}_+ \ \middle| \ \|r\|_1 = 1, \ \|r\|_2^2 \leq 1 + 2\rho, \text{ and } r \text{ is non-increasing} \right\};$$

The restriction of $\mathcal{R}$ to non-increasing functions is "free" since $F^{-1}$ is non-decreasing. Our strategy is to relate $F^{-1}$ to CVaR and then apply the corresponding bias bounds (5)—this type of transformation is closely related to the Kusuoka representation of coherent risk measures [42]. Specifically, note that

$$\mathcal{L}_{\mathsf{CVaR}}^\alpha = \frac{1}{\alpha} \int_0^\alpha F_Z^{-1}(1 - \beta) \mathrm{d}\beta \implies F_Z^{-1}(1 - \alpha) = \frac{\mathrm{d}}{\mathrm{d}\alpha}(\alpha \mathcal{L}_{\mathsf{CVaR}}^\alpha).$$

Therefore, for any $r \in \mathcal{R}$ integration by parts gives

$$\int_0^1 r(\beta) F_Z^{-1}(1-\beta) \mathrm{d}\beta = \int_0^1 r(\alpha) \frac{\mathrm{d}}{\mathrm{d}\alpha}(\alpha \mathcal{L}_{\mathrm{CVaR}}^\alpha) \mathrm{d}\alpha = r(1) \mathcal{L}_{\mathrm{CVaR}}^1 - \int_0^1 r'(\alpha) \alpha \mathcal{L}_{\mathrm{CVaR}}^\alpha \mathrm{d}\alpha.$$

The CVaR bias bound (35) tells us that $\mathcal{L}_{\mathrm{CVaR}}^\alpha \leq \overline{\mathcal{L}}_{\mathrm{CVaR}}^\alpha + \mathrm{bb}(\alpha)$ where $\mathrm{bb}(\alpha) = 3B \min\left\{\sqrt{\frac{1}{\alpha n}}, 1\right\}$. Moreover, we may write $\overline{\mathcal{L}}_{\mathrm{CVaR}}^\alpha = \frac{1}{\alpha}\int_0^\alpha \mathbb{E}\,\widehat{F}^{-1}(1-\beta)\mathrm{d}\beta$, where $\widehat{F}$ denotes the empirical cdf of the losses $\ell(x; S_1), \ldots, \ell(x; S_n)$. Noting that $r'(\alpha) \leq 0$ for all $\alpha$, we may write

$$-\int_0^1 r'(\alpha)\alpha \mathcal{L}_{\mathrm{CVaR}}^\alpha \mathrm{d}\alpha \leq -\int_0^1 r'(\alpha)\alpha \overline{\mathcal{L}}_{\mathrm{CVaR}}^\alpha \mathrm{d}\alpha - \int_0^1 r'(\alpha)\alpha \cdot \mathrm{bb}(\alpha)\mathrm{d}\alpha$$

$$= \mathbb{E}\int_0^1 r(\beta)\widehat{F}^{-1}(1-\beta)\mathrm{d}\beta - r(1)\overline{\mathcal{L}}_{\mathrm{CVaR}}^1 + \int_0^1 [r(\alpha) - r(1)](\alpha \cdot \mathrm{bb}(\alpha))'\mathrm{d}\alpha,$$

where in the final equality we used again integration by parts along with $\mathbb{E}\,\widehat{F}^{-1}(1-\alpha) = \frac{\mathrm{d}}{\mathrm{d}\alpha}(\alpha \overline{\mathcal{L}}_{\mathrm{CVaR}})$.

Substituting back and using $\mathcal{L}_{\mathrm{CVaR}}^1 = \overline{\mathcal{L}}_{\mathrm{CVaR}}^1 = \mathbb{E}\,\ell(x; S)$, we obtain

$$\int_0^1 r(\beta)F_Z^{-1}(1-\beta)\mathrm{d}\beta - \mathbb{E}\int_0^1 r(\beta)\widehat{F}^{-1}(1-\beta)\mathrm{d}\beta \leq \sup_{r \in \mathcal{R}}\int_0^1 [r(\alpha) - r(1)](\alpha \cdot \mathrm{bb}(\alpha))'\mathrm{d}\alpha =: E$$

Taking a supremum over $r \in \mathcal{R}$, we conclude that

$$\mathcal{L}_{\chi^2}(x; P_0) = \sup_{r \in \mathcal{R}}\int_0^1 r(\beta)F_Z^{-1}(1-\beta)\mathrm{d}\beta \leq \sup_{r \in \mathcal{R}}\mathbb{E}\int_0^1 r(\beta)\widehat{F}^{-1}(1-\beta)\mathrm{d}\beta + E$$

$$\leq \mathbb{E}\sup_{r \in \mathcal{R}}\int_0^1 r(\beta)\widehat{F}^{-1}(1-\beta)\mathrm{d}\beta + E = \overline{\mathcal{L}}_{\chi^2}(x; n) + E. \tag{36}$$

It remains to bound the quantity $E$, which we do via the the Cauchy-Schwarz inequality and the definition of $\mathcal{R}$, which gives

$$E = \int_0^1 [r(\alpha) - r(1)](\alpha \cdot \mathrm{bb}(\alpha))'\mathrm{d}\alpha \leq \|r\|_2 \|(\alpha \cdot \mathrm{bb}(\alpha)'\|_2 \leq \sqrt{1 + 2\rho} \cdot \|(\alpha \cdot \mathrm{bb}(\alpha))'\|_2$$

for all $r \in \mathcal{R}$. We calculate $(\alpha \cdot \mathrm{bb}(\alpha))' = \mathrm{bb}(0)1_{\{\alpha \leq 1/n\}} + \frac{1}{2}\mathrm{bb}(\alpha)1_{\{\alpha > 1/n\}}$, so that

$$\|(\alpha \cdot \mathrm{bb}(\alpha)'\|_2^2 = \frac{\mathrm{bb}^2(0)}{n}\left(1 + \int_{1/n}^1 \frac{\mathrm{d}\beta}{4\beta}\right) \leq (3B)^2 \cdot \frac{4 + \log n}{4n}.$$

for all $r \in \mathcal{R}$, giving the required bound.

**Remark 1.** The bound (36) hold for any loss (12) and not just $\mathcal{L}_{\chi^2}$. Moreover, the final bound using Cauchy-Schwarz is equally valid for any $\rho$-$\chi^2$-bounded uncertainty set. In particular, consider the Cressie-Read uncertainty sets [15] corresponding to $k$-norm the constraint $\|r\|_k^2 \leq 1 + 2\rho$. For $k > 2$ they satisfy $\|r\|_2^2 \leq 1 + 2\rho$ and our bias bounds holds (using Hölder's inequality instead of Cauchy-Schwarz removes the logarithmic factor). For $k \in (1, 2)$ Hölder's inequality gives bounds decaying as $n^{-(k-1)/k}$.

**Proof of the bound** (8) Starting with CVaR, we return to the expression (32) for the bias and note that

$$[1_{\{u \geq 1-\alpha\}} - \mathcal{I}_\alpha(u)]F^{-1}(u) \leq [1_{\{u \geq 1-\alpha\}} - \mathcal{I}_\alpha(u)]\{F^{-1}(1-\alpha) + G_{\mathrm{icdf}} \cdot (u - (1-\alpha))\}$$

holds for all $u$, because when $u < 1 - \alpha$ we have that $1_{\{u \geq 1-\alpha\}} - \mathcal{I}_\alpha(u) \leq 0$ and so we increase the LHS by replacing $F^{-1}$ with an under-estimate, while for $u \geq 1 - \alpha$ we have $1 - \mathcal{I}_\alpha(u) \geq 0$ due

to (33) and we increase the LHS be replacing it with an $F^{-1}$ with an over-estimate. Substituting into (32) and calculating gives

$$\mathcal{L}_{\mathrm{CVaR}}(x; P_0) - \overline{\mathcal{L}}_{\mathrm{CVaR}}(x; n)$$

$$\leq \frac{1}{\alpha} \int_0^1 \left(1_{\{u \geq 1-\alpha\}} - \mathcal{I}_\alpha(u)\right) \left[F^{-1}(1-\alpha) + G_{\mathrm{icdf}} \cdot (u - [1-\alpha])\right] \mathrm{d}u$$

$$\overset{(i)}{=} \frac{G_{\mathrm{icdf}}}{\alpha} \int_0^1 \left(1_{\{u \geq 1-\alpha\}} - \mathcal{I}_\alpha(u)\right) u \, \mathrm{d}u$$

$$\overset{(ii)}{=} G_{\mathrm{icdf}} \left[\frac{1}{2\alpha}\left(1 - (1-\alpha)^2\right) - \frac{1}{\alpha n} \sum_{i=n-\lfloor \alpha n \rfloor + 1}^{n} \frac{i}{n+1} - \left(1 - \frac{\lfloor \alpha n \rfloor}{\alpha n}\right) \frac{n - \lfloor \alpha n \rfloor}{n+1}\right]$$

$$= G_{\mathrm{icdf}} \left[1 - \frac{\alpha}{2} - \frac{1}{\alpha n (n+1)} \frac{\lfloor \alpha n \rfloor}{2}(2n - \lfloor \alpha n \rfloor + 1) - \left(1 - \frac{\lfloor \alpha n \rfloor}{\alpha n}\right) \frac{n - \lfloor \alpha n \rfloor}{n+1}\right]$$

$$= G_{\mathrm{icdf}} \left[1 - \frac{\alpha}{2} - \frac{1}{2} \frac{\lfloor \alpha n \rfloor}{\alpha n (n+1)} (\lfloor \alpha n \rfloor + 1) - \frac{n - \lfloor \alpha n \rfloor}{n+1}\right]$$

$$= G_{\mathrm{icdf}} \left[\frac{1}{n+1} + \frac{\lfloor \alpha n \rfloor}{n+1}\left[1 - \frac{\lfloor \alpha n \rfloor}{2\alpha n}\right] - \frac{\alpha}{2} - \frac{1}{2} \frac{\lfloor \alpha n \rfloor}{\alpha n (n+1)}\right]$$

$$\leq G_{\mathrm{icdf}} \left[\frac{1}{n+1} + \frac{\alpha n}{2(n+1)} - \frac{\alpha}{2}\right] \leq \frac{G_{\mathrm{icdf}}}{n+1}. \tag{37}$$

Above, $(i)$ uses the fact that $\frac{1}{\alpha}\mathcal{I}_\alpha$ is a convex combination of densities to deduce that

$$\int_0^1 \left(1_{\{u \geq 1-\alpha\}} - \mathcal{I}_\alpha(u)\right) \left[F^{-1}(1-\alpha) - G_{\mathrm{icdf}} \cdot (1-\alpha)\right] \mathrm{d}u = 0,$$

and $(ii)$ uses the definition (32) of $\mathcal{I}_\alpha$ along with the fact that $\mathbb{E}\,\mathrm{Beta}(a,b) = \frac{a}{a+b}$.

This bound extends to any $\mathcal{L}$ of the form (12) via (36), since we have $\mathrm{bb}(\alpha) = G_{\mathrm{icdf}}/(n+1)$ independent of $\alpha$ and consequently $(\alpha \cdot \mathrm{bb}(\alpha))' = G_{\mathrm{icdf}}/(n+1)$, giving

$$E = \sup_{r \in \mathcal{R}} \int_0^1 [r(\alpha) - r(1)](\alpha \cdot \mathrm{bb}(\alpha))' \mathrm{d}\alpha = \frac{G_{\mathrm{icdf}}}{n+1} \cdot \sup_{r \in \mathcal{R}} \int_0^1 [r(\alpha) - r(1)] \mathrm{d}\alpha \leq \frac{G_{\mathrm{icdf}}}{n+1},$$

since $\int r(\alpha) \mathrm{d}\alpha = 1$ for all $r \in \mathcal{R}$ regardless of $\phi$ and $\psi$.

**Penalized-$\chi^2$** We use the shorthand $Z = \ell(x, S)$ and for a sample $S_1^n$ we let $Z_i = \ell(x, S_i)$. By Eq. (23),

$$\mathcal{L}_{\chi^2\text{-pen}}(x; P_0) = \Upsilon(\eta^\star; P_0) = \mathbb{E}\,\frac{(Z - \eta^\star)_+^2}{2\lambda} + \eta^* + \frac{\lambda}{2},$$

where $\eta^*$ is the unique solution to $\mathbb{E}(Z - \eta^\star)_+ = \lambda$. (We omit the dependence of $\Upsilon$ on $x$ as $x$ is constant throughout). Similarly, we have that

$$\mathcal{L}_{\chi^2\text{-pen}}(x; S_1^n) = \Upsilon(\eta_n; S_1^n), \quad \text{where} \quad \Upsilon(\eta_n; S_1^n) := \sum_{i=1}^n \frac{(Z_i - \eta)_+^2}{2\lambda n} + \eta + \frac{\lambda}{2}$$

and $\eta_n$ is the unique solution to $\frac{1}{n}\sum_{i=1}^n (Z_i - \eta)_+ = \lambda$. Convexity of $\Upsilon$ w.r.t. $\eta$ gives us

$$\Upsilon(\eta_n; S_1^n) \geq \Upsilon(\eta^\star; S_1^n) + \Upsilon'(\eta^\star; S_1^n)(\eta_n - \eta^\star).$$

Taking expectation, we observe that $\mathbb{E}\,\Upsilon(\eta_n; S_1^n) = \overline{\mathcal{L}}_{\chi^2\text{-pen}}(x; n)$ and $\mathbb{E}\,\Upsilon(\eta^\star; S_1^n) = \Upsilon(\eta^\star; P_0) = \mathcal{L}_{\chi^2\text{-pen}}(x; P_0)$. Therefore, by the Cauchy-Schwarz inequality

$$\overline{\mathcal{L}}_{\chi^2\text{-pen}}(x; n) - \mathcal{L}_{\chi^2\text{-pen}}(x; P_0) = \mathbb{E}\,\Upsilon'(\eta^\star; S_1^n)(\eta_n - \eta^\star)$$

$$\overset{(\star)}{=} \mathbb{E}\,\Upsilon'(\eta^\star; S_1^n)(\eta_n - \mathbb{E}\,\eta_n) \geq -\sqrt{\mathrm{Var}\Upsilon'(\eta^\star; S_1^n)}\sqrt{\mathrm{Var}\eta_n}, \tag{38}$$

where $(\star)$ uses that $\mathbb{E}\,\Upsilon'(\eta^\star; S_1^n) = \mathbb{E}\,\Upsilon'(\eta^\star; P_0) = 0$ by the definition of $\eta^\star$, and therefore we may replace $\eta_n - \eta^\star$ with $\eta_n - \mathbb{E}\,\eta_n$. We now proceed to bound each variance separately. First, we have

$$\mathrm{Var}\,\Upsilon'(\eta^\star; S_1^n) = \mathbb{E}\left[\frac{1}{n}\sum_{i=1}^n\left(\frac{(Z_i - \eta^*)_+}{\lambda} - 1\right)\right]^2 = \frac{1}{n}\,\mathbb{E}\left(\frac{(Z - \eta^*)_+}{\lambda} - 1\right)^2$$

$$= \frac{1}{n}\left[\mathbb{E}\,\frac{(Z - \eta^\star)_+^2}{\lambda^2} - 1\right] = \frac{1}{n}\left[\frac{1}{\lambda}(2\mathcal{L}_{\chi^2\text{-pen}}(x; P_0) - 2\eta^* - \lambda) - 1\right] \leq \frac{2B}{\lambda n}, \quad (39)$$

where in the final transition we used $\mathcal{L}_{\chi^2\text{-pen}}(x; P_0) \leq B$ and $\eta^\star \geq -\lambda$ due to $\mathbb{E}(Z - \eta^\star)_+ = \lambda$ and $Z \geq 0$.

To handle the second variance we use the Efron-Stein inequality (Lemma 2). Let $I$ be uniformly distributed on $[n]$, and define

$$\tilde{Z}_1^n = (Z_1, \ldots, Z_{I-1}, Z_I', Z_{I+1}, \ldots, Z_n),$$

where $Z'$ is an i.i.d. copy of $Z$. Let $\tilde{\eta}_n$ be the solution to $\frac{1}{n}\sum_{i=1}^n(\tilde{Z}_i - \eta)_+ = \lambda$. Then,

$$\mathrm{Var}\,\eta_n \leq \frac{n}{2}\,\mathbb{E}(\eta_n - \tilde{\eta}_n)^2 \quad (40)$$

Define the random set

$$\mathcal{A} := \{i \mid Z_i - \eta_n > 0\}.$$

Recalling that $\sum_{i=1}^n(\tilde{Z}_i - \tilde{\eta}_n)_+ = \sum_{i=1}^n(Z_i - \eta_n)_+ = \lambda n$, we have

$$0 = \sum_{i\in[n]}\left\{(Z_i - \eta_n)_+ - (\tilde{Z}_i - \tilde{\eta}_n)_+\right\} \leq \sum_{i\in\mathcal{A}}\left\{(Z_i - \eta_n) - (\tilde{Z}_i - \tilde{\eta}_n)_+\right\}$$

$$\leq \sum_{i\in\mathcal{A}}\left\{(Z_i - \eta_n) - (\tilde{Z}_i - \tilde{\eta}_n)\right\} = |\mathcal{A}|(\tilde{\eta}_n - \eta_n) + (Z_I - Z_I')\mathbf{1}_{\{I\in\mathcal{A}\}},$$

and therefore $\eta_n - \tilde{\eta}_n \leq \frac{B\mathbf{1}_{\{I\in\mathcal{A}\}}}{|\mathcal{A}|}$. Similarly defining $\tilde{\mathcal{A}} := \{i \mid \tilde{Z}_i - \tilde{\eta}_n > 0\}$ and applying the same argument with $\tilde{\eta}_n$ and $\eta_n$ swapped allows us to conclude that

$$(\eta_n - \tilde{\eta}_n)^2 \leq B^2\max\left\{\frac{\mathbf{1}_{\{I\in\mathcal{A}\}}}{|\mathcal{A}|^2}, \frac{\mathbf{1}_{\{I\in\tilde{\mathcal{A}}\}}}{|\tilde{\mathcal{A}}|^2}\right\} \leq B^2\left(\frac{\mathbf{1}_{\{I\in\mathcal{A}\}}}{|\mathcal{A}|^2} + \frac{\mathbf{1}_{\{I\in\tilde{\mathcal{A}}\}}}{|\tilde{\mathcal{A}}|^2}\right).$$

Taking expectation, we obtain

$$\mathbb{E}(\eta_n - \tilde{\eta}_n)^2 \leq 2B^2\,\mathbb{E}\left[\frac{\mathbf{1}_{\{I\in\mathcal{A}\}}}{|\mathcal{A}|^2}\right] = \frac{2B^2}{n}\,\mathbb{E}\left[\frac{1}{|\mathcal{A}|}\right],$$

where the final transition follows from $\mathbb{E}[\,\mathbf{1}_{\{I\in\mathcal{A}\}} \mid |\mathcal{A}|] = |\mathcal{A}|/n$ (since $I$ is uniform on $[n]$). Assume for the moment that $\lambda \leq B$. Then we must have $\eta_n \geq 0$ and moreover $|\mathcal{A}| \geq n\min\{1, \lambda/B\}$ with probability 1. Substituting back into (40), we get the variance bound

$$\mathrm{Var}\,\eta_n \leq \frac{B^3}{\lambda n}. \quad (41)$$

Combining (41), (39) and (38) gives the result for $\lambda \leq B$.

In the edge case that $\lambda \geq B$, Eq. (24) gives us that

$$\overline{\mathcal{L}}_{\chi^2\text{-pen}}(x; n) = \mathbb{E}\,\frac{1}{n}\sum_{i\leq n}Z_i + \frac{1}{2\lambda}\,\mathbb{E}\,\mathrm{Var}[Z_1^n] = \mathbb{E}\,Z + \frac{n-1}{2n\lambda}\mathrm{Var}[Z]$$

$$= \mathcal{L}(x; P_0) - \frac{1}{2\lambda}\mathrm{Var}[Z] \geq \mathcal{L}(x; P_0) - \frac{B^2}{2\lambda n}.$$

We note that in this case may easily form an unbiased estimator of $\mathcal{L}_{\chi^2\text{-pen}}$ by using the standard unbiased variance estimator. $\qquad\square$

### B.1.2 Worst-case tightness of bias bounds

**Proposition 4.** *For $p \in [0, 1]$, let $P_0 = \mathsf{Bernoulli}(p_0)$ and $\ell(x; s) = B \cdot s$. The following results hold.*

- *Set $p_0 = \alpha$, then*

$$\mathcal{L}_{\mathrm{CVaR}}(x; P_0) - \overline{\mathcal{L}}_{\mathrm{CVaR}}(x; n) \gtrsim \frac{B\sqrt{1 - \alpha}}{\sqrt{\alpha n}}.$$

- *Set $p_0 = (1 + 2\rho)^{-1}$, then*

$$\mathcal{L}_{\chi^2}(x; P_0) - \overline{\mathcal{L}}_{\chi^2}(x; n) \gtrsim B\sqrt{\frac{\rho}{n}}.$$

- *Set $p_0 = \lambda / B \leq 1/2$, then*

$$\mathcal{L}_{\chi^2\text{-pen}}(x; P_0) - \overline{\mathcal{L}}_{\chi^2\text{-pen}}(x; n) \gtrsim \frac{B^2}{\lambda n}.$$

*Proof.* As before, we treat each case separately.

**CVaR.** First, note that $\mathcal{L}_{\mathrm{CVaR}}(x; P_0) = B$ since for $Q$ such that $Q(1) = 1$ we have $\frac{\mathrm{d}Q}{\mathrm{d}P_0}(s) = \frac{1}{\alpha} 1_{\{s=1\}}$ and therefore $Q \in \mathcal{U}_{\mathrm{CVaR}}^\alpha(P_0)$. Second, for a sample $S_1^n \in \{0, 1\}^n$ we have

$$\mathcal{L}_{\mathrm{CVaR}}(x; S_1^n) = B \max\left\{1, \frac{1}{\alpha n} \sum_{i \in [n]} S_i\right\}.$$

Therefore

$$\mathcal{L}_{\mathrm{CVaR}}(x; P_0) - \overline{\mathcal{L}}_{\mathrm{CVaR}}(x; n) = \frac{B}{\alpha n} \mathbb{E}\left(n\alpha - \sum_{i \in [n]} S_i\right)_+$$

$$\geq \frac{B\sqrt{1 - \alpha}}{\sqrt{\alpha n}} \mathbb{P}\big(\mathsf{Bin}(n, \alpha) \leq n\alpha - \sqrt{n\alpha(1 - \alpha)}\big) \gtrsim \frac{B\sqrt{1 - \alpha}}{\sqrt{\alpha n}},$$

where the final bound follows from the Berry-Esseen theorem (see Lemma 1).

**Constrained-$\chi^2$.** The $\chi^2$ divergence between two Bernoulli random variables is

$$\mathrm{D}_{\chi^2}(\mathsf{Bernoulli}(q), \mathsf{Bernoulli}(p)) = \frac{1}{2} p \left(\frac{q}{p} - 1\right)^2 + \frac{1}{2}(1 - p)\left(\frac{1 - q}{1 - p} - 1\right)^2 = \frac{(q - p)^2}{2p(1 - p)}.$$

Therefore, for any $p \in (0, 1)$, the element in $\mathcal{U}_{\chi^2}^\rho(\mathsf{Bernoulli}(p))$ that maximizes $Q(1)$ is $Q = \mathsf{Bernoulli}(q)$ with $q = \min\left\{1, p + \sqrt{2\rho}\sqrt{p(1 - p)}\right\}$. Set $p_0 = \frac{1}{1 + 2\rho}$ and note that the function $f(p) = p + \sqrt{2\rho}\sqrt{p(1 - p)} = p + \sqrt{(1 - p_0)/p_0}\sqrt{p(1 - p)}$ satisfies $f(p_0) = 1$ and $f'(p) \geq \frac{1}{2p_0}$ for all $p \leq p_0$. Therefore, we have

$$\mathcal{L}_{\chi^2}(x; \mathsf{Bernoulli}(p)) \leq B\left[1 - \left(\frac{p_0 - p}{2p_0}\right)_+\right]$$

for all $p \in (0, 1)$, with equality at $p = p_0$. In particular, setting $P_0 = \mathsf{Bernoulli}(p_0)$ implies $\mathcal{L}_{\chi^2}(x; P_0) = B$ and for a sample $S_1^n \sim P_0^n$ with $\hat{p} = \frac{1}{n}\sum_{i \in [n]} S_i$ we have $\mathcal{L}_{\chi^2}(x; S_1^n) \leq B\left(1 - \frac{p_0 - \hat{p}}{2p_0}\right)$. Therefore

$$\mathcal{L}_{\chi^2}(x; P_0) - \overline{\mathcal{L}}_{\chi^2}(x; n) \geq \frac{B}{2p_0} \mathbb{E}(p_0 - \hat{p})_+ = \frac{B}{2p_0 n} \cdot \mathbb{E}\left(np_0 - \sum_{i \in [n]} S_i\right)_+ \overset{(\star)}{\gtrsim} \frac{B\sqrt{1 - p_0}}{\sqrt{p_0 n}} = B\sqrt{\frac{2\rho}{n}},$$

where $(\star)$ follows from the CVaR case and for the final equality we substitute the definition of $p_0$.

**Penalized-$\chi^2$.** For any $p \in (0,1)$ we have

$$\mathcal{L}(x; \text{Bernoulli}(p)) = \sup_{q \in [0,1]} \left\{ qB - \lambda D_{\chi^2}(\text{Bernoulli}(q), \text{Bernoulli}(p)) \right\}$$

$$= \sup_{q \in (0,1)} \left\{ qB - \frac{\lambda(q-p)^2}{2p(1-p)} \right\} = \begin{cases} pB\left(1 + \frac{(1-p)B}{2\lambda}\right) & p \leq \lambda/B \\ B - \frac{\lambda(1-p)}{2p} & \text{otherwise.} \end{cases}$$

Simplifying, we have,

$$\mathcal{L}(x; \text{Bernoulli}(p)) \leq \frac{B+\lambda}{2} + \frac{B^2}{2\lambda} \cdot \left[ \left(p - \frac{\lambda}{B}\right) - \frac{B}{\lambda} \frac{\left(p - \frac{\lambda}{B}\right)_+^2}{1 + \frac{B}{\lambda}\left(p - \frac{\lambda}{B}\right)_+} \right],$$

with equality at $p = \lambda/B$. Taking $p_0 = \lambda/B$ and and for a sample $S_1^n \sim P_0^n$ letting $\hat{p} = \frac{1}{n}\sum_{i \in [n]} S_i$, we have

$$\mathcal{L}_{\chi^2\text{-pen}}(x; P_0) - \overline{\mathcal{L}}_{\chi^2\text{-pen}}(x; n) \geq -\frac{B}{2p_0}\mathbb{E}(\hat{p} - p_0) + \frac{B}{2p_0^2}\mathbb{E}\frac{(\hat{p} - p_0)_+^2}{1 + \frac{1}{p_0}(\hat{p} - p_0)_+}.$$

Since $\mathbb{E}\hat{p} = p_0$, we may lower bound this as

$$\mathcal{L}_{\chi^2\text{-pen}}(x; P_0) - \overline{\mathcal{L}}_{\chi^2\text{-pen}}(x; n) \geq \frac{B(1-p_0)}{4np_0}\mathbb{P}(\sqrt{n^{-1}p_0(1-p_0)} \leq \hat{p} - p_0 \leq p_0).$$

We have

$$\mathbb{P}(\sqrt{n^{-1}p_0(1-p_0)} \leq \hat{p} - p_0 \leq p_0)$$
$$\geq \mathbb{P}(\text{Bin}(n, p_0) \geq np_0 + \sqrt{np_0(1-p_0)}) - \mathbb{P}(\text{Bin}(n, p_0) \geq 2np_0) \gtrsim 1$$

by Berry-Esseen and Chernoff, and the result follows by substituting $p_0 = \lambda/B$. $\qquad\square$

## B.2 Discussion of additional assumptions

### B.2.1 Smoothness of $\ell$

The guarantees for the accelerated gradient iterations (10), detailed in Appendix B.4, require the objective function be smooth, i.e., have Lipschitz gradient. However, the degree of smoothness need not be high: as Nesterov [52] and subsequent work [43, 23] observed, even if $\nabla\mathcal{L}$ is order $G^2/\epsilon$ Lipschitz, acceleration allows finding an $\epsilon$ accurate solution in roughly $GR/\epsilon$ steps (a quadratic improvement over the SGM rate), as long as the gradient variance is itself of order $\epsilon$; the accelerated rates in Theorem 1 stem from this fact.

By Claim 1, for $\mathcal{L}$ to have roughly $G^2/\epsilon$ Lipschitz gradient, the loss gradients $\nabla\ell$ have to be $H = G^2/\epsilon$ Lipschitz. This is in fact a weak assumption, because every $G$-Lipschitz loss $\ell$ has a *smoothed* version $\tilde{\ell}$ that satisfies $|\tilde{\ell}(x; s) - \ell(x; s)| \lesssim \epsilon$ for all $x, s$ and that $\nabla\tilde{\ell}(x; s)$ is $G^2/\epsilon$ Lipschitz. For example, we may replace the hinge loss $\ell(x; s) = (1 - x^\top s)_+$ with $\tilde{\ell}(x; s) = \epsilon\log(1 + \exp([1 - x^\top s]/\epsilon))$. More generally, the smoothing [33]

$$\tilde{\ell}(x; s) = \inf_{y \in \mathcal{X}} \left\{ \ell(y; s) + \frac{G^2}{2\epsilon}\|y - x\|^2 \right\} \tag{42}$$

works for any $G$-Lipschitz $\ell$.

In practice, we are often at liberty to replace the original loss $\ell$ with its smoothed version $\tilde{\ell}$ and minimize the resulting objective $\tilde{\mathcal{L}}$ which is guaranteed to be sufficiently smooth and approximates $\mathcal{L}$ to accuracy $\epsilon$. Indeed, in the "statistical learning" model where we observe the entire $\ell(\cdot; S)$ per sample of $S \sim P_0$, we can apply the smoothing (42) to enforce the smoothness requirement without loss of generality. Therefore, our smoothness assumption can fail to hold only in situation where $\ell$ is non-smooth and $\ell$ and $\nabla\ell$ are strict black-boxes, so we cannot compute (42) without multiple black-box queries.

### B.2.2 Lipschitz inverse-cdf

The inverse-cdf of $\ell(x; S)$ is Lipschitz if and only if the distribution of $\ell(x; S)$ has positive density in the interval $[\min_{s \in \mathbb{S}} \ell(x; s), \max_{s \in \mathbb{S}} \ell(x; s)]$. This is a rather strong assumption that fails whenever $\mathbb{S}$ is discrete or $\ell(x; S)$ is distributed as two separate bulks. However, the conclusions of our analysis under the Lipschitz inverse-cdf assumption hold under two natural relaxations.

**Near-Lipschitz inverse-cdf and discrete loss distributions.** Note that if $F^{-1}$ satisfies $|F^{-1}(u) - \tilde{F}^{-1}(u)| \leq \delta$ for all $u \in [0, 1]$ and a $G_{\mathrm{icdf}}$-Lipschitz $\tilde{F}^{-1}(u)$, then we can repeat the proof of the bound (8) to show that $\mathcal{L}(x; P_0) - \overline{\mathcal{L}}(x; n) \leq \delta + \frac{G_{\mathrm{icdf}}}{n+1}$ for all objectives of the form (12). Moreover, suppose that $P_0$ is uniform on $N$ elements $s_1^N$ such that $\ell(x; s_i)$ is increasing in $i$, and suppose that it holds that

$$\ell(x; s_{i+1}) - \ell(x; s_i) \leq \frac{G_{\mathrm{icdf}}}{N}. \tag{43}$$

That is, the increments in the loss are not too far from uniform. Then, the piecewise linear function $\tilde{F}^{-1}$ connecting the steps in $F^{-1}$ is $G_{\mathrm{icdf}}$-Lipschitz and satisfies $|F^{-1}(u) - \tilde{F}^{-1}(u)| \leq G_{\mathrm{icdf}}/N$. Therefore, the assumption (43) implies that for any mini-batch size $n < N$, we have $\mathcal{L}(x; P_0) - \overline{\mathcal{L}}(x; n) \leq 2G_{\mathrm{icdf}}/(n+1)$. We note also that the assumption $n < N$ is essentially without loss of generality, since for $n = N$ we can simply use a full-batch method with no bias at all.

**CVaR bias bounds with locally Lipschitz inverse-cdf.** The proof of the bound (37) also works if $F^{-1}$ is Lipschitz in a small neighborhood of the CVaR cutoff $1 - \alpha$, because for values of $u$ that are roughly $\sqrt{n^{-1}\alpha}$ far from $1 - \alpha$ we may bound $|1_{\{u \geq 1-\alpha\}} - \mathcal{I}_\alpha(u)|$ via tail bounds, as in the proof of the bound (5). Therefore, we expect the bias of $\mathcal{L}_{\mathrm{CVaR}}(x; P_0) - \overline{\mathcal{L}}(x; n)$ to vanish with rate $1/n$ whenever the distribution of $\ell(x; S)$ has a density at the $1 - \alpha$ quantile loss value. Prior work shows that, from an asymptotic perspective, the converse is also true: when $\ell(x; S)$ does not have a density at the $1 - \alpha$ quantile, the bias vanishes with asymptotic rate $n^{-1/2}$ [cf. 71, Theorem 2].

### B.3 Proofs of variance bounds

We give a more general statement of the variance bound using the notion of $C$-$\chi^2$-bounded objectives (Definition 1); Proposition 2 follows immediately from Claim 4.

**Proposition 2'.** *Let $\mathcal{L}$ be an objective of the form (12). If $\mathcal{L}$ is $C$-$\chi^2$-bounded, we have that for all $n \in \mathbb{N}$ and $x \in \mathcal{X}$*

$$\mathrm{Var}[\mathcal{L}(x; S_1^n)] \leq \frac{2(1 + C)}{n} B^2.$$

*If in addition $\phi = 0$ and $\psi$ is strictly convex, we have*

$$\mathrm{Var}[\nabla \mathcal{L}(x; S_1^n)] \leq \frac{8(1 + C)}{n} G^2.$$

*Proof.* We first show the bound on the objective variance. By the the Efron-Stein inequality (see Lemma 2), we have

$$\mathrm{Var}[\mathcal{L}(x; S_1^n)] \leq \frac{n}{2} \mathbb{E} \left( \mathcal{L}(x; S_1^n) - \mathcal{L}(x; \tilde{S}_1^n) \right)^2, \tag{44}$$

where $S$ and $\tilde{S}$ are identical except in a random entry $I$ for which $\tilde{S}_I$ is an iid copy of $S_I$. Let $q$ and $\tilde{q}$ denote the maximizers of (13) for samples $S_1^n$ and $\tilde{S}_1^n$ respectively. In addition, let $Z_i = \ell(x; S_i)$ and $\tilde{Z}_i = \ell(x; \tilde{S}_i)$. Clearly, $\mathcal{L}(x; S_1^n)$ is convex in $Z$ and satisfies $\frac{\partial}{\partial Z} \mathcal{L}(x; S_1^n) = q$. Therefore,

$$\mathcal{L}(x; S_1^n) - \mathcal{L}(x; \tilde{S}_1^n) \leq \left\langle \frac{\partial}{\partial Z} \mathcal{L}(x; S_1^n), Z - \tilde{Z} \right\rangle = q_I(\ell(x; S_I) - \ell(x; \tilde{S}_I)).$$

Applying the argument again with $S$ and $\tilde{S}$ swapped, we find that

$$|\mathcal{L}(x; S_1^n) - \mathcal{L}(x; \tilde{S}_1^n)| \leq \max\{q_I, \tilde{q}_I\} |\ell(x; S_I) - \ell(x; \tilde{S}_I)| \leq B \sqrt{q_I^2 + \tilde{q}_I^2}.$$

Therefore, using the fact the $q_I$ and $\tilde{q}_I$ are identically distributed, we have

$$\mathbb{E}(\mathcal{L}(x; S_1^n) - \mathcal{L}(x; \tilde{S}_1^n))^2 \leq 2B^2 \, \mathbb{E} \, q_I^2 = \frac{2G^2}{n} \mathbb{E}\|q\|_2^2 = \frac{4B^2}{n^2}(C + 1),$$

where the final bound is due to $\|q\|_2^2 = \frac{1}{n}(2D_{\chi^2}(q, \frac{1}{n}\mathbf{1}) + 1)$ and the $C$-$\chi^2$-bounded property of $\mathcal{L}$. Substituting back into (44) gives the claimed objective variance bound.

Next, to show the bound on the gradient variance we invoke Efron-Stein elementwise to obtain

$$\mathrm{Var}[\nabla\mathcal{L}(x; S_1^n)] \leq \frac{n}{2} \mathbb{E} \|\nabla\mathcal{L}(x; S_1^n) - \nabla\mathcal{L}(x; \tilde{S}_1^n)\|^2.$$

By the expression (28) for $\nabla\mathcal{L}$ we have

$$\|\nabla\mathcal{L}(x; S_1^n) - \nabla\mathcal{L}(x; \tilde{S}_1^n)\| = \left\| \sum_{i \neq I} (q_i - \tilde{q}_i)\nabla\ell(x; S_i) + q_I\nabla\ell(x; S_I) - \tilde{q}_I\nabla\ell(x; \tilde{S}_I) \right\|$$

$$\leq G\left( \sum_{i \neq I} |q_i - \tilde{q}_i| + q_I + \tilde{q}_I \right),$$

where the bound follows from the triangle inequality and the fact that $\ell$ is $G$-Lipschitz.

Now, observe that $q_i = \frac{1}{n}\psi^{*\prime}[(\ell(x; S_i) - \eta)/\lambda]$ for some $\eta \in \mathbb{R}$ by Eq. (20). Similarly, $q_i = \frac{1}{n}\psi^{*\prime}[(\ell(x; S_i) - \tilde{\eta})/\lambda]$ for some $\tilde{\eta}$. Since $\psi$ is strictly convex we have that $\psi^{*\prime}$ is continuous and monotonic non-decreasing. Consequently, either $q_i \geq \tilde{q}_i$ for all $i \neq I$ (if $\eta \leq \tilde{\eta}$), or $q_i \leq \tilde{q}_i$ for all $i \neq I$ (if $\eta \geq \tilde{\eta}$). In either case, we have

$$\sum_{i \neq I} |q_i - \tilde{q}_i| = \left| \sum_{i \neq I} (q_i - \tilde{q}_i) \right| = |q_I - \tilde{q}_I|,$$

where the final equality used the fact that $\sum_{i \leq n} q_i = \sum_{i \leq n} \tilde{q}_i = 1$. Substituting back, we find that

$$\|\nabla\mathcal{L}(x; S_1^n) - \nabla\mathcal{L}(x; \tilde{S}_1^n)\| \leq 2G \max\{q_I, \tilde{q}_I\} \leq 2G\sqrt{q_I^2 + \tilde{q}_I^2}.$$

The remainder of the proof is identical to that of the objective variance bound, except with $2G$ replacing $B$. $\qquad\square$

Proposition 2' implies that the variance of the $\chi^2$ constraint objective $\mathcal{L}_{\chi^2}$ is at most $2(1 + \rho)B^2/n$. However, our gradient variance bound requires $\phi = 0$ and therefore does not apply to $\nabla\mathcal{L}_{\chi^2}$. The following proposition shows that the requirement $\phi = 0$ is necessary, since no upper bound of the from $O(1)(1 + \rho)G^2/n$ holds for $\mathrm{Var}[\nabla\mathcal{L}_{\chi^2}]$.

**Proposition 5** (Variance of the mini-batch gradient estimator for $\mathcal{L}_{\chi^2}$). *For any $n > 4$ and $\rho \geq 0$, there exists a distribution $P_0$ over $\mathbb{S} = \{0, 1, 2\}$ and a $G$-Lipschitz loss $\ell : [-1, 1] \times \mathbb{S} \to [0, 1]$ such that*

$$\mathrm{Var}\left[\nabla\mathcal{L}_{\chi^2}(0; S_1^n)\right] \gtrsim \frac{\rho^2}{(1 + \rho)^2}G^2.$$

*Proof.* We construct $P_0$ as follows,

$$\mathbb{P}(S = 2) = p_2 = 1 - 2^{1/n} \approx \frac{\log 2}{n} \text{ and } \mathbb{P}(S = 1) = p_1 = \frac{1}{1 + 2\rho}.$$

(Note that we may assume without loss of generality that $\rho \gtrsim 1/n$, so that $P(S = 0) = 1 - p_1 - p_2 > 0$, since for $\rho = 0$ we already have a standard $1/n$ lower bound on the variance). We set the loss values to be

$$\ell(0; 0) = 0 \ , \ \ell(0; 1) = \frac{1}{30n} \text{ and } \ell(0; 2) = 1,$$

and the loss gradients as

$$\nabla\ell(0; 2) = \nabla\ell(0; 0) = -G \text{ and } \ell(0; 1) = G.$$

The source of high variance in this construction is that, for a sample $S_1^n$, the maximizing $q^\star$ behaves very differently when $S_i = 2$ for some $i$ and when $S_i \neq 2$ for all $i$. In the former case, we show that $q^\star$ puts significant mass on samples with $S_i \neq 1$, so $\nabla\mathcal{L}(0; S_1^n) < G(1 - c)$ for some $c \gtrsim \rho/(1 + \rho)$. In the latter case, we show that with constant probability $q^\star$ places mass only on samples with $S_i = 1$,

and so $\nabla\mathcal{L}(0; S_1^n) = G$. Since either scenario occurs with constant probability, the variance bound follows.

To provide a detailed proof, let $C_k(S_1^n) = \sum_{i \leq n} 1_{\{S_i = k\}}$ be the number of samples with value $k$, for $k \in \{0, 1, 2\}$, and consider the events

$$\mathfrak{E}_a(S_1^n) = \{C_2(S_1^n) = 0 \text{ and } C_1(S_1^n) \geq np_1\}$$

and

$$\mathfrak{E}_b(S_1^n) = \{C_2(S_1^n) = 1 \text{ and } C_1(S_1^n) < np_1\}.$$

Note that we chose $p_2$ such that $\mathbb{P}(C_2(S_1^n) = 0) = (1 - p_2)^n = \frac{1}{2}$ and that $\mathbb{P}(C_1(S_1^n) \geq np_1) \gtrsim 1$ since $np_1$ is roughly the median of $C_1(S_1^n)$. Similarly, $\mathbb{P}(C_2(S_1^n) = 1) = np_0(1 - p_0)^{n-1} \approx \frac{\log 2}{2}$ and $\mathbb{P}(C_1(S_1^n) < np_1) \gtrsim 1$. Therefore,

$$\mathbb{P}(\mathfrak{E}_a(S_1^n)) \gtrsim 1 \text{ and } \mathbb{P}(\mathfrak{E}_b(S_1^n)) \gtrsim 1. \tag{45}$$

We bound $\nabla\mathcal{L}_{\chi^2}$ conditional on each event in turn.

Under event $\mathfrak{E}_a(S_1^n)$, the empirical loss distribution is Bernoulli with parameter $C_1(S_1^n)/n \geq p_1 = 1/(1 + 2\rho)$ and consequently $q^\star$ places mass only on samples with value 1 (see further discussion in the proof of Proposition 4). Therefore, we have

$$\mathbb{E}[\nabla\mathcal{L}_{\chi^2}(0; S_1^n) \mid \mathfrak{E}_a(S_1^n)] = G. \tag{46}$$

To bound the gradient under event $\mathfrak{E}_b(S_1^n)$, assume that without loss of generality that $S_1 = 2$ is the unique sample with that value. We consider separately the cases $q_1^\star > 2/3$ and $q_1^\star \leq 2/3$. In the former, we clearly have $\nabla\mathcal{L}_{\chi^2}(0; S_1^n) \leq -q_1^\star G + (1 - q_1^\star)G < -G/3$. In the latter case, we recall Eq. (27) showing that $q^\star$ is of the form

$$q_i^\star = \frac{(\ell(x; S_i) - \eta^\star)_+}{\sum_{j \leq n}(\ell(x; S_j) - \eta^\star)_+}$$

for some $\eta^\star \in \mathbb{R}$. The fact that $q_1^\star \leq 2/3$ and that there are at most $n$ samples with value $\ell(0; 1) = 1/(30n)$ gives the following bound on $\eta^\star$

$$\frac{2}{3} \geq q_1^\star = \frac{\ell(0; S_1) - \eta^\star}{\sum_{j \leq n}(\ell(x; S_j) - \eta^\star)_+} \geq \frac{1 - \eta^\star}{31/30 - n\eta^\star} \implies \eta^\star \leq -\frac{1}{3n}.$$

Suppose $S_j = 0$ and $S_i = 1$, then

$$r = \frac{q_j^\star}{q_i^\star} = \frac{\ell(0; 0) - \eta^\star}{\ell(0; 1) - \eta^\star} = 1 - \frac{\ell(0; 1)}{\ell(0; 1) - \eta^\star} \geq \frac{7}{8}.$$

Assuming that $S_1 = 2$, we have that the total weight under $q^\star$ of samples with gradient $-G$ is

$$q_1^\star + (1 - q_1^\star)\frac{rC_0(S_1^n)}{C_1(S_1^n) + rC_0(S_1^n)} \geq \frac{7}{8}(1 - p_1) = \frac{7\rho}{4(1 + 2\rho)},$$

which implies $\nabla\mathcal{L}_{\chi^2}(0; S_1^n) \leq -\frac{7\rho}{4(1+2\rho)}G + (1 - \frac{7\rho}{4(1+2\rho)})G \leq G(1 - \frac{7\rho}{2(1+2\rho)})$. We conclude that

$$\mathbb{E}[\nabla\mathcal{L}_{\chi^2}(0; S_1^n) \mid \mathfrak{E}_b(S_1^n)] \leq G(1 - c) \text{ for } c = \min\left\{\frac{4}{3}, \frac{7\rho}{2(1 + 2\rho)}\right\} \gtrsim \frac{\rho}{1 + \rho}. \tag{47}$$

Let $\tilde{S}_1^n$ be an independent copy of $S_1^n$. We combine our conclusions (45), (46) and (47) to form a variance bound as follows,

$$\text{Var}[\nabla\mathcal{L}_{\chi^2}(0; S_1^n)] = \frac{1}{2}\mathbb{E}\left(\nabla\mathcal{L}_{\chi^2}(0; S_1^n) - \nabla\mathcal{L}_{\chi^2}(0; \tilde{S}_1^n)\right)^2$$

$$\geq \frac{1}{2}\mathbb{E}\left[\left(\nabla\mathcal{L}_{\chi^2}(0; S_1^n) - \nabla\mathcal{L}_{\chi^2}(0; \tilde{S}_1^n)\right)^2 \,\Big|\, \mathfrak{E}_a(S_1^n), \mathfrak{E}_b(\tilde{S}_1^n)\right]\mathbb{P}(\mathfrak{E}_a(S_1^n), \mathfrak{E}_b(\tilde{S}_1^n))$$

$$\geq \frac{1}{2}c^2G^2 \cdot \mathbb{P}(\mathfrak{E}_a(S_1^n)) \cdot \mathbb{P}(\mathfrak{E}_b(\tilde{S}_1^n)) \gtrsim \frac{\rho^2}{(1 + \rho)^2}G^2.$$

$\square$

## B.4 Convergence rates of stochastic gradient methods

We state below the classical convergence rates for standard and accelerated stochastic gradient methods, under a somewhat non-standard assumption that the stochastic gradient estimates are unbiased for a uniform approximation of the objective function with additive error $\delta$.

**Proposition 6** (Convergence of stochastic gradient methods [43, Corollary 1])**.** *Let $F : \mathcal{X} \to \mathbb{R}$ and $\overline{F} : \mathcal{X} \to \mathbb{R}$ satisfy $0 \leq F(x) - \overline{F}(x) \leq \delta$ for all $\mathcal{X} \in \mathbb{R}$. Assume that $\overline{F}$ is convex and that a stochastic gradient estimator $\tilde{g}$ satisfies $\mathbb{E}\,\tilde{g}(x) \in \partial \overline{F}(x)$ and $\mathbb{E}\,\|\tilde{g}(x)\|^2 \leq \Gamma^2$ for all $x \in \mathcal{X}$. For $T \in \mathbb{N}$, the iterate $\bar{x}_T$ in the sequence (9) with $\eta \asymp \frac{R}{T^{1/2}\Gamma}$ satisfies*

$$\mathbb{E}\,F(\bar{x}_T) - \inf_{x'} F(x') \lesssim \delta + \frac{\Gamma R}{\sqrt{T}}. \tag{48}$$

*If in addition $\nabla \overline{F}$ is $\Lambda$-Lipschitz and $\mathrm{Var}[\tilde{g}(x)] \leq \sigma^2$ for all $x \in \mathcal{X}$, the iterate $y_T$ in the sequence (10) with $\eta \asymp \min\{\frac{1}{\Lambda}, \frac{R}{T^{3/2}\sigma}\}$ and $\theta_t = \frac{2}{t+1}$ satisfies*

$$\mathbb{E}\,F(y_T) - \inf_{x'} F(x') \lesssim \delta + \frac{\Lambda R^2}{T^2} + \frac{\sigma R}{\sqrt{T}}. \tag{49}$$

*Proof.* [43] gives us the rates (48) and (49) but for $\overline{F}$ rather than $F$. That is, it guarantees that SGM finds $\bar{x}_T$ such that

$$\mathbb{E}\,\overline{F}(\bar{x}_T) - \inf_{x'} \overline{F}(x') \lesssim \frac{\Gamma R}{\sqrt{T}}.$$

To remove the bars, we use $0 \leq F(x) - \overline{F}(x) \leq \delta$ to write

$$-\inf_{x'} F(x') \geq -\inf_{x'} \overline{F}(x') \quad \text{and} \quad F(\bar{x}_T) \geq \overline{F}(\bar{x}_T) + \delta.$$

$\square$

**Remark 2.** In the unconstrained case $\mathcal{X} = \mathbb{R}^d$, the recursion (10) reduces to the more familiar form

$$v_{t+1} = \omega_t v_t - \eta \tilde{g}(x_t), \quad x_{t+1} = x_t + \omega_{t+1} v_{t+1} - \eta \tilde{g}(x_t), \tag{50}$$

where $\omega_t = (1 - \theta_{t-1})\frac{\theta_t}{\theta_{t-1}}$ is a time-varying "momentum" parameter; the sequences $y_t, z_t$ are related to $v_t$ via $v_t = \frac{\theta_t}{\omega_t}(z_t - y_t)$ and $y_{t+1} = x_t - \eta \tilde{g}(x_t)$.

## B.5 Proofs of complexity bounds

**Theorem 1.** *Let Assumptions A1 and A2 hold, possibly trivially (with $H = \infty$ or $G_{\mathrm{icdf}} = \infty$). Let $\epsilon \in (0, B)$ and write $\nu = \frac{H}{G^2}\epsilon$. With suitable choices of the batch size $n$ and iteration count $T$, the gradient methods (9) and (10) find $\bar{x}$ satisfying $\mathbb{E}\,\mathcal{L}(\bar{x}, P_0) - \inf_{x' \in \mathcal{X}} \mathcal{L}(x'; P_0) \leq \epsilon$ with complexity $nT$ admitting the following bounds.*

- *For $\mathcal{L} = \mathcal{L}_{\mathrm{CVaR}}$, we have $nT \lesssim \frac{(GR)^2}{\alpha \epsilon^2}\left(1 + \min\left\{\frac{\alpha G_{\mathrm{icdf}}\sqrt{\log\frac{1}{\alpha}+\nu}}{GR}, \frac{B^2\sqrt{\log\frac{1}{\alpha}+\nu}}{GR\epsilon}, \frac{B^2}{\epsilon^2}\right\}\right)$.*

- *For $\mathcal{L} = \mathcal{L}_{\chi^2\text{-pen}}$ with $\lambda \leq B$, we have $nT \lesssim \frac{(GR)^2 B}{\lambda \epsilon^2}\left(1 + \min\left\{\frac{B}{GR}\sqrt{\frac{\epsilon(1+\nu)}{\lambda}}, \frac{B}{\epsilon}\right\}\right)$.*

- *For $\mathcal{L} = \mathcal{L}_{\chi^2}$, we have $nT \lesssim \frac{(1+\rho)(GR)^2 B^2}{\epsilon^4}\log\frac{(1+\rho)B^2}{\epsilon^2}$.*

- *For any loss of the from (2), we have $nT \lesssim \frac{(GR)^2 G_{\mathrm{icdf}}}{\epsilon^3}$.*

*Proof.* To prove each bound in the theorem we choose $n$ large enough via one of the bounds in Proposition 1 and then choose $T$ to guarantee $\epsilon$-accurate solution via Proposition 6. For a (potentially random) point $\bar{x} \in \mathcal{X}$ and robust risk $\mathcal{L}$, we define the shorthand

$$\mathsf{err}(x; \mathcal{L}) := \mathbb{E}\,\mathcal{L}(\bar{x}; P_0) - \inf_{x \in \mathcal{X}} \mathcal{L}(x; P_0).$$

We summarize our choices of $n$ and $T$ for different robust objectives, under different assumptions in Table B.5. In the statement of the theorem, we sometimes upper bound $a \vee b := \max\{a, b\}$ by $a + b$ for readability, and state the tighter rates here.

| Loss | $\nabla$ est. | $n$ | $T$ | complexity $= nT$ |
|---|---|---|---|---|
| $\mathcal{L}_{\text{CVaR}}$ | $\nabla\mathcal{L}_{\text{CVaR}}$ | $\frac{B^2}{\alpha\epsilon^2}$ | $\frac{(GR)^2}{\epsilon^2}$ | $\frac{(GR)^2 B^2}{\alpha\epsilon^4}$ |
| $\mathcal{L}_{\chi^2}$ | $\nabla\mathcal{L}_{\chi^2}$ | $\frac{(1+\rho)B^2}{\epsilon^2}\log\frac{(1+\rho)B^2}{\epsilon^2}$ | $\frac{(GR)^2}{\epsilon^2}$ | $\frac{(1+\rho)(GR)^2 B^2}{\epsilon^4}\log\frac{(1+\rho)B^2}{\epsilon^2}$ |
| $\mathcal{L}_{\chi^2\text{-pen}}$ | $\nabla\mathcal{L}_{\chi^2\text{-pen}}$ | $\frac{B^2}{\lambda\epsilon}$ | $\frac{(GR)^2}{\epsilon^2}$ | $\frac{(GR)^2 B^2}{\lambda\epsilon^3}$ |
| any $\mathcal{L}$ in (2) | $\nabla\mathcal{L}$ | $\frac{G_{\text{icdf}}}{\epsilon}$ | $\frac{(GR)^2}{\epsilon^2}$ | $\frac{(GR)^2 G_{\text{icdf}}}{\epsilon^3}$ |
| $\mathcal{L}_{\text{CVaR}}$ | $\nabla\mathcal{L}_{\text{kl-CVaR}}$ | $\frac{B^2}{\alpha\epsilon^2}$ | $\frac{GR\sqrt{\log\frac{1}{\alpha}+\nu}}{\epsilon}\vee\frac{(GR)^2}{B^2}$ | $\frac{(GR)^2}{\alpha\epsilon^2}\left(1\vee\frac{B^2}{GR\epsilon}\sqrt{\log\frac{1}{\alpha}+\nu}\right)$ |
| $\mathcal{L}_{\text{CVaR}}$ | $\nabla\mathcal{L}_{\text{kl-CVaR}}$ | $\frac{G_{\text{icdf}}}{\epsilon}$ | $\frac{GR\sqrt{\log\frac{1}{\alpha}+\nu}}{\epsilon}\vee\frac{(GR)^2}{\alpha G_{\text{icdf}}\epsilon}$ | $\frac{(GR)^2}{\epsilon^2}\left(\frac{1}{\alpha}\vee\frac{G_{\text{icdf}}}{GR}\sqrt{\log\frac{1}{\alpha}+\nu}\right)$ |
| $\mathcal{L}_{\chi^2\text{-pen}}$ | $\nabla\mathcal{L}_{\chi^2\text{-pen}}$ | $\frac{B^2}{\lambda\epsilon}$ | $\frac{GR\sqrt{1+\nu}}{\epsilon}\vee\frac{(GR)^2}{B\epsilon}$ | $\frac{GRB}{\lambda\epsilon^2}\left(B\sqrt{1+\nu}\vee GR\right)$ |

**Table 2.** Parameter settings for Theorem 1. For $\mathcal{L}_{\text{kl-CVaR}}$ we take $\lambda\asymp\frac{\epsilon}{\log(\frac{1}{\alpha})}$. For $\mathcal{L}_{\chi^2\text{-pen}}$ assume $\lambda\geq\epsilon$. $\nu:=H\epsilon/G^2$. We use the shorthand $a\vee b$ for $\max\{a,b\}$.

**CVaR.** We distinguish between the different possible assumptions on the loss $\ell$ and distribution $P_0$ as they yield different rates.

(a) Non-smooth $\ell$: let $\bar{x}_T$ be the iterates of (9), the sub-optimality guarantee of (48) and the bias bound of Proposition 1 yield

$$\text{err}(\bar{x}_T;\mathcal{L}_{\text{CVaR}})\lesssim\frac{B}{\sqrt{\alpha n}}+\frac{GR}{\sqrt{T}}.$$

In that case, setting $n\asymp\frac{B^2}{\alpha\epsilon^2}$ guarantees that the bias is smaller than $\epsilon$ and setting $T\asymp\frac{(GR)^2}{\epsilon^2}$ yields that $\text{err}(\bar{x}_T;\mathcal{L}_{\text{CVaR}})\lesssim\epsilon$.

(b) Smooth $\ell$: if $\ell$ is $H$-smooth, we consider the $\mathcal{L}_{\text{kl-CVaR}}$ objective with $\lambda=\frac{\epsilon}{\log\frac{1}{\alpha}}$. This guarantees that, for all $x\in\mathcal{X}$

$$\mathcal{L}_{\text{kl-CVaR}}(x;P_0)\leq\mathcal{L}_{\text{CVaR}}(x;P_0)\leq\mathcal{L}_{\text{kl-CVaR}}(x;P_0)+\epsilon.$$

$\overline{\mathcal{L}}_{\text{kl-CVaR}}$ being $(\frac{G^2\log(1/\alpha)}{\epsilon}+H)$-smooth, the final iterate of the sequence (10) achieves

$$\text{err}(y_T;\mathcal{L}_{\text{CVaR}})\lesssim\epsilon+\frac{B}{\sqrt{\alpha n}}+\frac{(GR)^2(\log\frac{1}{\alpha}+\nu)}{\epsilon T^2}+\frac{GR}{\sqrt{\alpha nT}}.$$

To make sure that the second and third terms are smaller than $\epsilon$, we set $T=\frac{(GR)^2}{\alpha n\epsilon^2}\vee\frac{GR}{\epsilon}\sqrt{\log\frac{1}{\alpha}+\nu}$. To guarantee small bias, we set $n\asymp\frac{B^2}{\alpha\epsilon^2}$; the resulting complexity is

$$nT\asymp\frac{(GR)^2}{\alpha\epsilon^2}\max\left\{1,\frac{B^2}{GR\epsilon}\sqrt{\log\frac{1}{\alpha}+\nu}\right\}.$$

(c) Smooth $\ell$ and inverse cdf Lipschitz: in this case, the regret guarantees of the iterates of (49) is

$$\text{err}(y_T;\mathcal{L}_{\text{CVaR}})\lesssim\epsilon+\frac{G_{\text{icdf}}}{n}+\frac{(GR)^2(\log\frac{1}{\alpha}+\nu)}{\epsilon T^2}+\frac{GR}{\sqrt{\alpha nT}}.$$

We once again set $T=\frac{(GR)^2}{\alpha n\epsilon^2}\vee\frac{GR}{\epsilon}\sqrt{\log\frac{1}{\alpha}+\nu}$, and choosing $n\asymp\frac{G_{\text{icdf}}}{\epsilon}$ yields the result.

**Penalized-$\chi^2$.** We distinguish between whether or not $\ell$ is smooth.

(a) Non-smooth $\ell$: for the sequence of iterates of (9), we have

$$\mathsf{err}(\bar{x}_T; \mathcal{L}_{\chi^2\text{-pen}}) \lesssim \frac{B^2}{\lambda n} + \frac{GR}{\sqrt{T}},$$

and setting $n \asymp \frac{B^2}{\lambda \epsilon}$ and $T \asymp \frac{(GR)^2}{\epsilon^2}$ yields the fist rate.

(b) Smooth $\ell$: We now turn to acceleration, we have

$$\mathsf{err}(y_T; \mathcal{L}_{\chi^2\text{-pen}}) \lesssim \frac{B^2}{\lambda n} + \frac{R^2\left(\frac{G^2}{\lambda} + H\right)}{T^2} + GR\sqrt{\frac{1 + \frac{B}{\lambda}}{nT}}.$$

First, noting that $\lambda \geq \epsilon$ guarantees that $R^2(\frac{G^2}{\lambda} + H) \leq \frac{(GR)^2(1+\nu)}{\epsilon}$. Furthermore, we simplify the variance term since $B/\lambda \geq 1$. We thus set $T \asymp \frac{GR}{\epsilon}\sqrt{1 + \nu} \vee \frac{(GR)^2 B}{\lambda n \epsilon^2}$ and choose $n \asymp \frac{B^2}{\lambda \epsilon^2}$. This yields the final result

$$nT \lesssim \frac{GRB}{\lambda \epsilon^2}\big(B\sqrt{1 + \nu} \vee (GR)\big).$$

**Constrained-$\chi^2$.** This case is straightforward—without any bound on the variance in the worst-case, we turn to the basic SGM guarantee (48); we have

$$\mathsf{err}(\bar{x}_T; \mathcal{L}_{\chi^2}) \lesssim B\sqrt{1 + 2\rho}\sqrt{\frac{\log n}{n}} + \frac{GR}{\sqrt{T}}.$$

We set $T \asymp \frac{(GR)^2}{\epsilon^2}$ and $n \asymp \frac{(1+2\rho)B^2}{\epsilon^2}\log((1 + 2\rho)B^2\epsilon^{-2})$. We then have

$$B\sqrt{1 + 2\rho}\sqrt{\frac{\log n}{n}} = \epsilon\sqrt{1 + \frac{\log\log(\frac{(1+2\rho)B^2}{\epsilon^2})}{\log(\frac{(1+2\rho)B^2}{\epsilon^2})}} \leq \sqrt{2}\epsilon,$$

and this concludes the proof.

**Lipschitz inverse-cdf.** The sequence of iterates (9) yield error

$$\mathsf{err}(\bar{x}_T; \mathcal{L}) \leq \frac{G_{\text{icdf}}}{n} + \frac{GR}{\sqrt{T}},$$

and setting $n \asymp \frac{G_{\text{icdf}}}{\epsilon}, T \asymp \frac{(GR)^2}{\epsilon^2}$ concludes the proof of the theorem. $\qquad\square$

## C  Proofs of Section 4

We now provide additional discussion of the multilevel Monte Carlo estimator for general functions $\mathsf{F}$, whose form we restate here for convenience

$$\widehat{\mathcal{M}}[\mathsf{F}] \coloneqq \mathsf{F}(x; S_1^{n_0}) + \frac{1}{q(J)}\widehat{\mathcal{D}}_{2^J n_0}, \text{ where } \widehat{\mathcal{D}}_k \coloneqq \mathsf{F}(x; S_1^k) - \frac{\mathsf{F}(x; S_1^{k/2}) + \mathsf{F}(x; S_{k/2+1}^k)}{2}. \quad (51)$$

Section C.1 provides upper bounds on the moments of $\widehat{\mathcal{M}}$ for estimating $\mathcal{L}_{\text{kl-CVaR}}, \mathcal{L}_{\chi^2\text{-pen}}$, and their gradients, proving Claim 2 and Proposition 3. In that section we also prove that similar second moment bounds do not always hold for $\nabla\mathcal{L}_{\chi^2}$. In Section C.2 we prove the complexity guarantees in Theorem 2, and we conclude in Section C.3 with a comparison of some of our design choices to the original proposal of Blanchet and Glynn [8].

## C.1  Proofs of moment bounds

**Claim 2'.** *For any function* $\mathsf{F}$*, the estimator* $\widehat{\mathcal{M}}[\mathsf{F}]$ *with parameters* $n = 2^{j_{\max}} n_0$ *satisfies*

$$\mathbb{E}\,\widehat{\mathcal{M}}[\mathsf{F}] = \mathbb{E}\,\mathsf{F}(S_1^n), \ \textit{requiring expected sample size}\ \mathbb{E}\,2^J n_0 = n_0(1 + \log_2(n/n_0)).$$

*Proof.* For any even $k$, $\mathbb{E}\,\widehat{\mathcal{D}}_k = \mathbb{E}\,\mathsf{F}(S_1^k) - \mathbb{E}\,\mathsf{F}(S_1^{k/2})$. Therefore, the expectation of $\widehat{\mathcal{M}}[\mathsf{F}]$ telescopes: $\mathbb{E}\,\widehat{\mathcal{M}}[\mathsf{F}] = \mathbb{E}[\mathsf{F}(S_1^{n_0})] + \sum_{j=1}^{j_{\max}} \mathbb{E}\,\widehat{\mathcal{D}}_{2^j n_0} = \mathbb{E}[\mathsf{F}(S_1^n)]$. The expected number of samples follows from direct calculation: $\mathbb{E}[2^J] = \sum_{j=1}^{j_{\max}} 2^j \mathbb{P}(J = j) = j_{\max} + 1$. $\qquad\square$

We have the following bound on the second moment of the estimator,

$$\mathbb{E}\left\|\widehat{\mathcal{M}}[\mathsf{F}]\right\|^2 \le 2\left\|\mathsf{F}(S_1^{n_0})\right\|^2 + \sum_{j=1}^{j_{\max}} \frac{2}{q(j)}\,\mathbb{E}\left\|\widehat{\mathcal{D}}_{2^j n_0}\right\|^2 \le 2\left\|\mathsf{F}(S_1^{n_0})\right\|^2 + \sum_{j=1}^{j_{\max}} 2^{j+1}\,\mathbb{E}\left\|\widehat{\mathcal{D}}_{2^j n_0}\right\|^2. \quad (52)$$

For $\chi^2$-bounded (Definition 1) pure-penalty losses such as $\mathcal{L}_{\text{kl-CVaR}}$ and $\mathcal{L}_{\chi^2\text{-pen}}$, we argue that $\mathbb{E}\|\widehat{\mathcal{D}}_k\|^2 \lesssim 1/k$, so that $2^j\,\mathbb{E}\|\widehat{\mathcal{D}}_{2^j n_0}\|^2 \lesssim 1/n_0$. Substituting into the bound (52) gives the following guarantees, from which Proposition 3 follows immediately via Claim 4.

**Proposition 3'.** *Let $\mathcal{L}$ be an objective of the form* (12) *with $\phi = 0$ and strictly convex $\psi$. If $\mathcal{L}$ is $C$-$\chi^2$-bounded, we have that for all $x \in \mathcal{X}$, the multi-level Monte Carlo estimator with parameters $n$ and $n_0$ satisfies*

$$\mathbb{E}\left(\widehat{\mathcal{M}}[\mathcal{L}]\right)^2 \le 2B^2\left(1 + \frac{2C}{n_0}\log_2(n/n_0)\right) \ \textit{and}\ \ \mathbb{E}\left\|\widehat{\mathcal{M}}[\nabla\mathcal{L}]\right\|^2 \le 2G^2\left(1 + \frac{2C}{n_0}\log_2(n/n_0)\right).$$

*Proof.* The proof follows similarly to the proof of Proposition 2', where the key step is to bound $\widehat{\mathcal{D}}_k$ for $k \in 2\mathbb{N}$. We distinguish between estimating the gradient and the loss as, for the latter, one needs to account for estimating the regularizer $\mathrm{D}_\psi$.

**Gradient estimator.** We start with the proof of the second moment of the gradient estimator. Let $k \in 2\mathbb{N}$ and let $q, q'$ and $q''$ be the maximizer of (13) for $S_1^k, S_1^{k/2}$ and $S_{k/2+1}^k$ respectively. We have

$$\|\widehat{\mathcal{D}}_k\| = \left\|\sum_{i \le k}\left(q_i - \tfrac{1}{2}q_i' 1_{\{i \le k/2\}} - \tfrac{1}{2}q_{i-k/2}'' 1_{\{i > k/2\}}\right)\nabla\ell(x; S_i)\right\|$$

$$\le G\sum_{i \le k/2}|q_i - \tfrac{1}{2}q_i'| + G\sum_{i > k/2}|q_i - \tfrac{1}{2}q_{i-k/2}''|.$$

For $i \in \{1, \ldots, k/2\}$, it holds that $q_i = \frac{1}{n}\psi^{*\prime}[(\ell(x; S_i) - \eta)/\lambda]$ and $q_i' = \frac{2}{n}\psi^{*\prime}[(\ell(x; S_i) - \eta')/\lambda]$ for $\eta, \eta' \in \mathbb{R}$. Since that $\psi$ is strictly convex, $\psi^{*\prime}$ is increasing and $q_i - \tfrac{1}{2}q_i'$ is of constant sign for $i \in \{1, \ldots, k/2\}$. Therefore,

$$\sum_{i \le k/2}\left|q_i - \frac{1}{2}q_i'\right| = \left|\sum_{i \le k/2}\left(q_i - \frac{1}{2}q_i'\right)\right| = \left|\sum_{i \le k/2}q_i - \frac{1}{2}\right|.$$

By symmetry, it thus holds that

$$\mathbb{E}\,\|\widehat{\mathcal{D}}_k\|^2 \le 4G^2\,\mathbb{E}\left(\sum_{i \le k/2}q_i - \frac{1}{2}\right)^2 \overset{(i)}{\le} \frac{2}{k}\mathrm{D}_{\chi^2}(q, \tfrac{1}{k}\mathbf{1}) \overset{(ii)}{\le} \frac{2CG^2}{k},$$

where $(i)$ is due to Lemma 3 and $(ii)$ follows from the assumption that $\mathcal{L}$ is $C$-$\chi^2$-bounded. Substituting into (52), we have

$$\mathbb{E}\,\|\widehat{\mathcal{M}}[\nabla\mathcal{L}]\|^2 \le 2G^2 + 4CG^2\sum_{j \le 1}^{j_{\max}}\frac{1}{q(j)2^j n_0}$$

$$\le 2G^2 + \frac{4CG^2}{n_0}\left(j_{\max} - \frac{1}{2}\right)$$

$$\le G^2\left(2 + \frac{4C}{n_0}\log_2(n/n_0)\right).$$

This concludes the argument for the gradient.

**Loss estimator.** With the same notation, let us define $\tilde{q} := [\frac{1}{2}q', \frac{1}{2}q''] \in \Delta^k$. We first prove that $\widehat{\mathcal{D}}_k \geq 0$. Indeed, we have

$$\mathcal{L}(x; S_1^k) = \sum_{i \leq k} q_i \ell(x; S_i) - \mathrm{D}_\psi(q, \tfrac{1}{k}\mathbf{1}) \overset{(i)}{\geq} \sum_{i \leq k} \tilde{q}_i \ell(x; S_i) - \mathrm{D}_\psi(\tilde{q}, \tfrac{1}{k}\mathbf{1})$$

$$\overset{(ii)}{=} \frac{1}{2}\mathcal{L}(x; S_1^{k/2}) + \frac{1}{2}\mathcal{L}(x; S_{k/2+1}^k),$$

where $(i)$ is because $q$ is the maximizer for $S_1^k$ and $(ii)$ because the $\psi$-divergence tensorizes, i.e., $\mathrm{D}_\psi(\tilde{q}, \tfrac{1}{k}\mathbf{1}) = \frac{1}{2}\mathrm{D}_\psi(q', \tfrac{2}{k}\mathbf{1}) + \frac{1}{2}\mathrm{D}_\psi(q'', \tfrac{2}{k}\mathbf{1})$. This guarantees that $\widehat{\mathcal{D}}_k = \mathcal{L}(x; S_1^k) - \frac{1}{2}\mathcal{L}(x; S_1^{k/2}) - \frac{1}{2}\mathcal{L}(x; S_{k/2+1}^k) \geq 0$.

Let us now upper bound $\widehat{\mathcal{D}}_k$. To that end, we define $\tilde{q}' = 2q_1^{k/2} + \delta$ where $\delta \in \mathbb{R}^{k/2}$ is a fixed-sign vector such that $\tilde{q}'$ lies in $\Delta^{k/2}$. More precisely, if $\tilde{q}'^\top \mathbf{1} > 1$, $\delta$ decreases the mass of the largest coordinate until $\tilde{q}'_{(1)} = \frac{2}{k}$ and iterates along the sorted coordinates until $\tilde{q}' \in \Delta^{k/2}$. If $\tilde{q}'^\top \mathbf{1} < 1$, $\delta$ similarly increases the smallest coordinate to $\frac{2}{k}$ until $\tilde{q}' \in \Delta^{k/2}$. Without loss of generality, we can assume that $\psi$ attains its minimum at $t = 1$ (otherwise may replaced it by $\psi(t) - \psi'(1)(t-1)$ without changing the objective). Therefore, since $\tilde{q}'$ is closer to $\frac{2}{k}\mathbf{1}$ than $2q_1^{k/2}$, it holds that

$$\mathrm{D}_\psi(\tilde{q}', \tfrac{2}{k}\mathbf{1}) = \frac{2}{k}\sum_{i \leq k/2} \psi(\tfrac{k\tilde{q}'}{2}) \leq \frac{2}{k}\sum_{i \leq k/2} \psi(kq_i)$$

Finally, we know that $q'$ is optimal for $S_1^{k/2}$ and so

$$\mathcal{L}(x; S_1^{k/2}) \geq \sum_{i \leq k/2} \tilde{q}'_i \ell(x; S_i) - \mathrm{D}_\psi(\tilde{q}', \tfrac{2}{k}\mathbf{1})$$

$$\geq 2\sum_{i \leq k/2} q_i \ell(x; S_i) - \frac{2}{k}\sum_{i \leq k/2} \psi(kq_i) - \sum_{i \leq k/2} [-\delta_i]_+ B$$

$$= 2\sum_{i \leq k/2} q_i \ell(x; S_i) - \frac{2}{k}\sum_{i \leq k/2} \psi(kq_i) - 2\left[\sum_{i \leq k/2} q_i - \frac{1}{2}\right]_+ B.$$

The same argument for the indices $\{k/2 + 1, \dots, k\}$ yields

$$\widehat{\mathcal{D}}_k \leq 2B\left\{\left[\sum_{i=1}^{k/2} q_i - \frac{1}{2}\right]_+ + \left[\sum_{i=k/2+1}^k q_i - \frac{1}{2}\right]_+\right\} = 2B\left|\sum_{i=1}^{k/2} q_i - \frac{1}{2}\right|.$$

Therefore, we have

$$\mathbb{E}(\widehat{\mathcal{D}}_k)^2 \leq 4B^2 \mathbb{E}\left[\sum_{i \leq k/2} q_i - \frac{1}{2}\right]^2 \overset{(i)}{\leq} \frac{2CB^2}{k},$$

where $(i)$ follows from Lemma 3 and the $C$-$\chi^2$-boundedness of $\mathcal{L}$. Substituting into (52) yields the desired bound on $\widehat{\mathcal{M}}[\mathcal{L}]$. $\qquad\square$

Having established the gradient estimator upper bounds for pure-penalty objectives, we demonstrate that similar bounds *do not* extend to the case of $\chi^2$ constraint.

**Proposition 7** (Lower bound in the case of constrained-$\chi^2$). *For every $\rho \geq 1$, $n_0$ and $n \geq 4$, there exists a distribution $P_0$ over $\mathbb{S} = \{0, 1, 2\}$ and a $G$-Lipschitz loss $\ell : [-1, 1] \times \mathbb{S} \to \mathbb{R}_+$ such that the multi-level Monte Carlo gradient estimator with parameters $n_0$ and $n$ satisfies*

$$\mathbb{E}\left\|\widehat{\mathcal{M}}[\nabla\mathcal{L}_{\chi^2}]\right\|^2 \gtrsim \frac{n}{n_0}G^2.$$

*Proof.* We reuse the construction and notation in the proof of Proposition 5 and so do not repeat it. For a sample $S_1^n$, we consider the event $\mathfrak{E}_a(S_1^{n/2})$ where $S_i \neq 2$ for all $i \leq n/2$ and there are at least $np_1/2$ samples with value 1. We argue in the proof of Proposition 5 (Eq. (46)) that under this event we have

$$\mathbb{E}[\nabla \mathcal{L}_{\chi^2}(0; S_1^{n/2}) \mid \mathfrak{E}_a(S_1^{n/2})] = G.$$

Moreover, we have

$$\nabla \mathcal{L}_{\chi^2}(0; S_{n/2+1}^n) \geq -G$$

with probability 1, so overall

$$\mathbb{E}\left[\frac{1}{2}\nabla \mathcal{L}_{\chi^2}(0; S_1^{n/2}) + \frac{1}{2}\nabla \mathcal{L}_{\chi^2}(0; S_{n/2+1}^n) \,\middle|\, \mathfrak{E}_a(S_1^{n/2})\right] \geq 0.$$

We also consider the event $\mathfrak{E}_b(S_1^n)$ that there is exactly one sample with value 2 and less the $np_1$ samples with value 1. As per the proof of Proposition 5 (Eq. (47)) we have

$$\mathbb{E}[\nabla \mathcal{L}_{\chi^2}(0; S_1^n) \mid \mathfrak{E}_b(S_1^n)] \leq -\frac{1}{6}G,$$

where we used $\rho \geq 1$.

Moreover, by the arguments in the proof of Proposition 5, we have

$$\mathbb{P}\left(\mathfrak{E}_a(S_1^{n/2}) \cap \mathfrak{E}_b(S_1^n)\right) \gtrsim 1.$$

Therefore, since $\widehat{\mathcal{D}}_n = \nabla \mathcal{L}_{\chi^2}(0; S_1^n) - \frac{1}{2}\nabla \mathcal{L}_{\chi^2}(0; S_1^{n/2}) - \frac{1}{2}\nabla \mathcal{L}_{\chi^2}(0; S_1^{n/2})$, we have

$$\mathbb{E}\|\widehat{\mathcal{D}}_n\|^2 \geq \mathbb{E}\left[\|\widehat{\mathcal{D}}_n\|^2 \mid \mathfrak{E}_a(S_1^{n/2}) \cap \mathfrak{E}_b(S_1^n)\right]\mathbb{P}\left(\mathfrak{E}_a(S_1^{n/2}) \cap \mathfrak{E}_b(S_1^n)\right)$$

$$\geq \frac{G^2}{36}\mathbb{P}\left(\mathfrak{E}_a(S_1^{n/2}) \cap \mathfrak{E}_b(S_1^n)\right) \gtrsim G^2.$$

The proof is complete by noting that

$$\mathbb{E}\left|\widehat{\mathcal{M}}[\nabla \mathcal{L}_{\chi^2}(0; \cdot)]\right|^2 \geq 2^{j_{\max}-1}\,\mathbb{E}\|\widehat{\mathcal{D}}_n\|^2 = \frac{n}{2n_0}\,\mathbb{E}\|\widehat{\mathcal{D}}_n\|^2 \gtrsim \frac{n}{n_0}G^2.$$

$\square$

Since the number $T$ of SGM iterations must be proportional to the second moment of the gradient estimator, Proposition 7 tells us that in the worst case we might have to set $T \asymp n(GR)^2/\epsilon^2$, in which case we might as well use a mini-batch estimator with batch size $n$ and run $(GR)^2/\epsilon^2$ SGM steps.

## C.2 Proof of complexity bounds

**Theorem 2** (MLMC complexity guarantees). *For $\epsilon \in (0, B)$, set $n \asymp \frac{B^2}{\alpha \epsilon^2}$, $1 \lesssim n_0 \lesssim \frac{\log n}{\alpha}$ and $T \asymp \frac{(GR)^2}{n_0 \alpha \epsilon^2}\log^2 n$. The stochastic gradient iterates (9) with $\tilde{g}(x) = \widehat{\mathcal{M}}[\nabla \mathcal{L}_{\text{CVaR}}(x; \cdot)]$ satisfy $\mathbb{E}[\mathcal{L}_{\text{CVaR}}(\bar{x}_T; P_0)] - \inf_{x \in \mathcal{X}}\mathcal{L}_{\text{CVaR}}(x; P_0) \leq \epsilon$ with complexity at most*

$$n_0 \log_2\left(\frac{n}{n_0}\right)T + 5\sqrt{(n\log n)^2 + n_0 n T \log n} \lesssim \frac{(GR+B)^2}{\alpha \epsilon^2}\log^2 \frac{B^2}{\alpha \epsilon^2} \quad w.p \;\geq 1 - \frac{1}{n}.$$

*The same conclusion holds when replacing $\mathcal{L}_{\text{CVaR}}$ with $\mathcal{L}_{\chi^2\text{-pen}}$ and $\alpha^{-1}$ with $1 + B/\lambda$.*

*Proof.* The convergence guarantee of Proposition 6 and the second moment bound of Proposition 3 directly give that iterates of the form (9) with the MLMC gradient estimator guarantees a regret smaller than $\epsilon$ for $n \asymp \frac{B^2}{\alpha \epsilon^2}$, $1 \lesssim n_0 \lesssim \frac{\log n}{\alpha}$ and $T \asymp \frac{(GR)^2}{n_0 \alpha \epsilon^2}\log^2 n$. However, since the multilevel estimator randomizes the batch size, it remains to show that the number of samples concentrates below the claimed bound. Let $K_t = n_0 2^{J_t}$ be the batch size at time $t$, and note that

$$\mathbb{E}\,K_1 = n_0 \log_2 \frac{2n}{n_0},$$
$$\mathbb{E}\,K_1^2 = 3n_0 n - 2n_0^2 \leq 3n_0 n, \quad \text{and}$$
$$K_1 \leq n \quad \text{with probability 1.}$$

Therefore, since $K_1^T$ are iid, a one-sided Bernstein bound [73, Prop. 2.14] implies that

$$\mathbb{P}\left[\sum_{t \leq T} K_t \geq n_0 \log_2(2n/n_0)T + \delta\right] \leq \exp\left(-\frac{\delta^2}{6n_0nT + \frac{n\delta}{3}}\right).$$

Solving in $\delta$ for the RHS to be equal to $\frac{1}{n}$ yields $\delta = \frac{n \log n}{3}(1 + \sqrt{1 + 216\frac{Tn_0}{n \log n}})$. We replace $n, n_0$ and $T$ by their values and conclude the proof. $\qquad\square$

### C.3 Comparison with Blanchet and Glynn [8]

There are two differences between our MLMC estimator and the proposal of Blanchet and Glynn [8]. First, we take $J$ to be a truncated $\mathsf{Geo}(1/2)$ random variable while they suggest $J \sim \mathsf{Geo}(2^{-3/2})$ without truncation—as we further discuss below, this modification is crucial for ensuring a useful second moment bound in our setting. The second difference is that we allow for a minimum sample size $n_0 > 1$ as opposed to $n_0 = 1$ in [8]. This modification is somewhat less important, as $n_0 = 1$ suffices for optimal gradient complexity, but choosing slightly larger $n_0$ is helpful in practice and can provably reduce the sequential depth of SGM by logarithmic factors.

Let us discuss in more detail the choice $p = 1/2$ in our construction of $J \sim \min\{\mathsf{Geo}(p), j_{\max}\}$. Inspection of Claim 2 shows that $p < 1/2$ implies that the expected sample cost is $\mathbb{E}\, 2^J n_0 \leq \frac{n_0}{1-2p}$ independent of $n = 2^{j_{\max}}n_0$, so in principle we could compute unbiased estimates even for $\mathbb{E}\, \mathsf{F}(S_1^\infty)$, i.e., the population objective. However, any $p < 1/2$ would result in overly large second moments: substituting $q(j) \propto p^{-j}$ and $\mathbb{E}\|\widehat{\mathcal{D}}_k\|^2 \asymp 1/k$ in (52) would result in bounds scaling with $(n/n_0)^{\log_2 1/(2p)}$. Therefore, $p = 1/2$ is the only value for which both the second moment and expected number of samples are sub-polynomial in $n$. In contrast, Blanchet and Glynn [8] apply the MLMC estimator to more regular functionals for which $\|\widehat{\mathcal{D}}_k\|^2 \lesssim 1/k^2$, and consequently can use a smaller value for $p$.

## D   Lower bound proofs

This section proves our lower bounds, which we restate for ease of reference.

**Theorem 3** (Minimax lower bounds). *Let* $G, R, \alpha, \lambda > 0$, $\epsilon \in (0, GR/64)$, *and sample space* $\mathbb{S} = [-1, 1]$. *There exists a numerical constant* $c > 0$ *such that the following holds.*

- *For each* $d \geq 1$, *domain* $\mathcal{X} = \{x \in \mathbb{R}^d \mid \|x\| \leq R\}$, *and any algorithm, there exists a distribution* $P_0$ *on* $\mathbb{S}$ *and convex G-Lipschitz loss* $\ell : \mathcal{X} \times \mathbb{S} \to [0, GR]$ *such that*

$$T \leq c\frac{(GR)^2}{\alpha\epsilon^2} \text{ implies } \mathbb{E}[\mathcal{L}_{\mathrm{CVaR}}(x_T; P_0)] - \inf_{x' \in \mathcal{X}} \mathcal{L}_{\mathrm{CVaR}}(x'; P_0) > \epsilon.$$

- *There exists* $d_\epsilon \lesssim (GR)^2\epsilon^{-2} \log \frac{GR}{\epsilon}$ *such that for* $\mathcal{X} = \{x \in \mathbb{R}^d \mid \|x\| \leq R\}$, *the same conclusion holds when replacing* $\mathcal{L}_{\mathrm{CVaR}}$ *with* $\widehat{\mathcal{L}}_{\chi^2\text{-pen}}$ *and* $\alpha$ *with* $\lambda/(GR)$.

Since our proofs for CVaR and $\chi^2$ penalty are quite different, we present them separately in Theorems 3a and 3b, respectively.

### D.1 CVaR lower bound

To prove the CVaR lower bound we use the following standard Le Cam reduction from stochastic optimization to hypothesis testing.

**Lemma 4.** *[21, Chapter 5] Let* $\mathcal{P}$ *be a set of distributions and* $P_{-1}, P_1 \in \mathcal{P}$ *and define*

$$\mathsf{d}_{\mathrm{opt}}(P_1, P_{-1}) \coloneqq \sup\{\delta' \geq 0 \mid \text{no } x \in \mathcal{X} \text{ is } \delta'\text{-optimal for both } \mathcal{L}(\cdot; P_{-1}) \text{ and } \mathcal{L}(\cdot; P_1)\}.$$

*Then for any measurable mapping* $\hat{x}_n : \mathbb{S}^n \to \mathcal{X}$ *we have*

$$\sup_{P \in \mathcal{P}} \mathbb{E}_{S_1^n \sim P^n}\{\mathcal{L}(\hat{x}_n(S_1^n); P)\} - \inf_{x' \in \mathcal{X}} \mathcal{L}(x'; P) \geq \frac{\mathsf{d}_{\mathrm{opt}}(P_1, P_{-1})}{2}\left(1 - \sqrt{\frac{n}{2}\mathsf{D}_{\mathrm{kl}}(P_{-1}, P_1)}\right),$$

Armed with Lemma 4, we state and prove the lower bound for CVaR.

**Theorem 3a** (CVaR lower bound). *Let $G, R, \alpha > 0$, $\epsilon \in (0, GR/64)$, $\mathbb{S} = [-G, G]$, $\mathcal{X} = [-R, R]$, and $\ell(x, s) = x \cdot s$. For any (potentially randomized) mapping $\hat{x}_n : \mathbb{S}^n \to \mathcal{X}$ there exists a distribution $P_0$ over $\mathbb{S}$ such that,*

$$n \le \frac{(GR)^2}{2048\alpha\epsilon^2} \quad \text{implies} \quad \mathbb{E}\,\mathcal{L}_{\mathrm{CVaR}}(\hat{x}_n(S_1^n); P_0) - \inf_{x \in \mathcal{X}} \mathcal{L}_{\mathrm{CVaR}}(x; P_0) \ge \epsilon.$$

*Proof.* Let us first assume that $\alpha \le \frac{1}{2}$. For $\delta \le \min\{\alpha, 1 - 2\alpha\}$, $\mu > 0$ and $v \in \{-1, 1\}$ we consider the distributions $P_v$ such that for $S_v \sim P_v$ we have

$$S_v = G \cdot \begin{cases} \mu & \text{with probability } \alpha + \delta v \\ -1 & \text{with probability } 1 - \alpha - \delta v \end{cases}. \tag{53}$$

For $x \in [-R, R]$, we let $\ell(x; s) = x \cdot s$. Since the CVaR objective is positively homogeneous, we have

$$\mathcal{L}_{\mathrm{CVaR}}(x; P_v) = |x| \cdot \mathcal{L}_{\mathrm{CVaR}}(\mathrm{sign}(x); P_v).$$

It therefore suffices to compute $\mathcal{L}_{\mathrm{CVaR}}(\pm 1; P_{\pm 1})$. A quick calculation yields

$$\mathcal{L}_{\mathrm{CVaR}}(1; S_1) = G\mu, \quad \mathcal{L}_{\mathrm{CVaR}}(-1, S_1) = G,$$

$$\mathcal{L}_{\mathrm{CVaR}}(1; S_{-1}) = G\mu\left(1 - \frac{\delta}{\alpha}\right) - G\frac{\delta}{\alpha} \quad \text{and} \quad \mathcal{L}_{\mathrm{CVaR}}(-1; S_{-1}) = G.$$

We thus have a closed-form expression for the CVaR objective: for $P_1$ we have

$$\mathcal{L}_{\mathrm{CVaR}}(x; P_1) = -Gx1_{\{x \le 0\}} + Gx\mu1_{\{x \ge 0\}},$$

which clearly attains its minimum at $x = 0$ where it has value 0. Choosing $\mu$ such that

$$\mu = \frac{\delta}{2\alpha}\left(1 - \frac{\delta}{2\alpha}\right)^{-1}$$

gives $\mathcal{L}_{\mathrm{CVaR}}(1; S_{-1}) = -G\mu$ and

$$\mathcal{L}_{\mathrm{CVaR}}(x; P_{-1}) = -Gx1_{\{x \le 0\}} - Gx\mu1_{\{x \ge 0\}},$$

which attains its minimum at $x = R$ where it has value $-GR\mu$. We therefore have that

$$\mathsf{d}_{\mathrm{opt}}(P_1, P_{-1}) = \frac{GR\mu}{2} \ge \frac{GR\delta}{4\alpha}. \tag{54}$$

Moreover, we have $t \log t - t + 1 \le (t - 1)^2$ for all $t \ge 0$, so that $\mathrm{D}_{\mathrm{kl}}(Q, P) \le 2\mathrm{D}_{\chi^2}(Q, P)$ for all $Q, P$, and in particular

$$\mathrm{D}_{\mathrm{kl}}(P_{-1}, P_1) \le 2\mathrm{D}_{\chi^2}(P_{-1}, P_1) = \frac{4\delta^2}{(1 - \alpha - \delta)(\alpha + \delta)} \le \frac{8\delta^2}{\alpha}, \tag{55}$$

where that last transition used $\delta \le \alpha$ and $\alpha \le 1/2$.

We take

$$\delta = \sqrt{\frac{\alpha}{16(n + \alpha^{-1})}},$$

where so that $\mathrm{D}_{\mathrm{kl}}(P_{-1}, P_1) \le 1/(2n)$ and Lemma 4 combined with (54) and (55) gives

$$\sup_{P \in \mathcal{P}} \mathbb{E}_{S_1^n \sim P^n}\{\mathcal{L}(\hat{x}_n(S_1^n); P)\} - \inf_{x' \in \mathcal{X}} \mathcal{L}(x'; P) \ge \frac{GR}{32\sqrt{\alpha n + 1}},$$

and the result follows from substituting $n \le \frac{(GR)^2}{2048\alpha\epsilon^2}$. When $\alpha \ge 1/2$ the result follows from the standard lower bound for stochastic convex optimization (e.g. [21, Thm. 5.2.10]). $\qquad\square$

## D.2 Penalized-$\chi^2$ lower bound

Computation of $\mathcal{L}_{\chi^2\text{-pen}}(x; P_{\pm})$ for the CVaR lower bound construction (53) shows that the argument does not easily transfer to the penalized-$\chi^2$ objective because—as opposed to constrained-$\chi^2$ and CVaR—it is not positive homogeneous in $x$.

Sidestepping this difficulty, we prove our lower bound using the different machinery of high-dimensional hard instances for oracle-based optimization [48]. We consider two standard oracles. First is the deterministic first-order oracle, that for a function $f : \mathbb{R}^d \to \mathbb{R}$ and a query $x$ returns

$$\mathsf{O}_f^{\mathrm{D}}(x) := (f(x), \nabla f(x)),$$

where we recall that $\nabla f(x)$ is an arbitrary element of $\partial f(x)$. Second is the stochastic oracle, that for a loss function $\ell : \mathcal{X} \times \mathbb{S} \to \mathbb{R}$ and distribution $P_0$ returns the randomized mapping

$$\mathsf{O}_{\ell,P_0}^{\mathrm{S}}(x) := (\ell(x; S), \nabla \ell(x; S)), \quad \text{for } S \sim P_0.$$

We construct the hard instance for $\mathcal{L}_{\chi^2\text{-pen}}$ based on the standard hard instance for non-stochastic convex optimization, whose properties are as follows.

**Proposition 8** (Braun et al. [10], Theorem V.1). *Let $\epsilon, G, R > 0$. There exists $d_\epsilon \lesssim (GR)^2 \epsilon^{-2} \log \frac{GR}{\epsilon}$ such that the following holds for $\mathcal{X} = \{x \in \mathbb{R}^{d_\epsilon} \mid \|x\| \leq R\}$. For any (possibly randomized) algorithm there exists $f_\epsilon : \mathcal{X} \to [0, GR]$ convex and $G$-Lipschitz such the query $x_T$ $\mathsf{O}_{f_\epsilon}^{\mathrm{D}}$ at iteration $T$ satisfies*

$$T \leq c \frac{(GR)^2}{\epsilon^2} \quad \text{implies} \quad \mathbb{E} f_\epsilon(x_T) - \inf_{\|x'\| \leq R} f_\epsilon(x') \geq \epsilon,$$

*for a numerical constant $c > 0$.*

In other words, any "dimension-free" algorithm needs to interact $\Omega(\epsilon^{-2})$ times with the deterministic oracle to obtain an $\epsilon$-suboptimal point. With this result, we prove our lower bound for optimizing $\mathcal{L}_{\chi^2\text{-pen}}$.

**Theorem 3b** (Penalized-$\chi^2$ lower bound). *Let $G, R, \lambda > 0$ and $\epsilon \in (0, GR)$. There exists $d_\epsilon \lesssim (GR)^2 \epsilon^{-2} \log \frac{GR}{\epsilon}$ such that the following holds for $\mathcal{X} = \{x \in \mathbb{R}^{d_\epsilon} \mid \|x\| \leq R\}$ and $\mathbb{S} \subseteq [-1, 1]$. For every algorithm there exists a distribution $P_0$ over $\mathbb{S}$ and $\ell : \mathcal{X} \times \mathbb{S} \to [-GR, GR]$ convex and $G$-Lipschitz in $x$, such that the query $x_T$ to $\mathsf{O}_{\ell,P_0}^{\mathrm{S}}$ at iteration $T$ satisfies*

$$T \leq c \frac{(GR)^3}{\lambda \epsilon^2} \quad \text{implies} \quad \mathbb{E}[\mathcal{L}_{\chi^2\text{-pen}}(x_T; P_0)] - \min_{x' \in \mathcal{X}} \mathcal{L}_{\chi^2\text{-pen}}(x'; P_0) > \epsilon,$$

*for $c > 0$ independent of $G, R, \lambda$ and $\epsilon$.*

*Proof.* Consider any convex and $G$ Lipschitz $f : \mathcal{X} \to [0, GR]$, define $\mathbb{S} := \{0, 1\}$ and $P_0 = \text{Bernoulli}(\frac{\lambda}{GR})$, and construct the following loss

$$\ell(x; S) := \begin{cases} f(x) & \text{if } S = 1 \\ -GR & \text{if } S = 0. \end{cases}$$

(If $\lambda > GR$ the result follows from the standard $(GR)^2/\epsilon^2$ lower bound for convex optimization). Expressing the resulting objective $\mathcal{L}_{\chi^2\text{-pen}}$ with the dual form (23) gives

$$\mathcal{L}_{\chi^2\text{-pen}}(x; P_0) = \inf_{\eta \in \mathbb{R}} \left\{ \frac{\lambda}{2} + \eta + \frac{1}{2\lambda} \left[ \frac{\lambda}{GR}(f_\epsilon(x) - \eta)_+^2 + \left(1 - \frac{\lambda}{GR}\right)(-GR - \eta)_+^2 \right] \right\}$$

$$= f(x) - \frac{GR - \lambda}{2},$$

since $\eta^\star = f(x) - GR \geq -GR$. We get that minimizing $\mathcal{L}_{\chi^2\text{-pen}}$ is equivalent to optimizing $f$.

Fix an algorithm interacting with $\mathsf{O}_{\ell,P_0}^{\mathrm{S}}$ and note that it implies a (randomized) algorithm interacting with $\mathsf{O}_f^{\mathrm{D}}$. Therefore we may take $f = f_\epsilon$, the hard function for this algorithm that Proposition 8 guarantees. Note that an algorithm interacting with $\mathsf{O}_{\ell,P_0}^{\mathrm{S}}$ receives information on $f_\epsilon$ only when

$S = 1$. Therefore, the worst-case expected optimality gap when minimizing $\mathcal{L}_{\chi^2\text{-pen}}$ with $T$ queries to $\mathsf{O}^{\mathsf{S}}_{\ell, P_0}$ is identical to the worst-case expected optimality gap when minimizing $f_\epsilon$ with $\mathsf{Bin}(T, \frac{\lambda}{GR})$ queries. Therefore, Proposition 8 tells us that for some $c' > 0$,

$$\mathbb{E}[\mathcal{L}(x_T; P_0) - \inf_{\|x'\| \leq R} \mathcal{L}(x'; P_0)] \geq \epsilon \cdot \mathbb{P}\left(\mathsf{Bin}\left(T, \frac{\lambda}{GR}\right) \leq c' \cdot \frac{(GR)^2}{\epsilon^2}\right).$$

Substituting $T \leq \frac{c}{4} \cdot \frac{(GR)^3}{\lambda \epsilon^2}$ gives that $\mathbb{P}(\mathsf{Bin}(T, \frac{\lambda}{GR}) \leq c' \cdot \frac{(GR)^2}{\epsilon^2}) \geq \frac{1}{2}$ by a standard Chernoff bound. The result follows by properly adjusting the constant factors (e.g., replacing $\epsilon$ with $2\epsilon$). $\qquad\square$

# E   A doubling scheme for minimizing $\mathcal{L}_{\chi^2}$

In this section, we obtain a stronger guarantee for minimizing the $\chi^2$-constraint robust objective. Namely, we leverage duality relationships to approximate the constrained objective $\mathcal{L}_{\chi^2}$ via its penalized counterpart, $\mathcal{L}_{\chi^2\text{-pen}}$. We adjust notation to make the dependence of $\mathcal{L}^\lambda_{\chi^2\text{-pen}}$ on $\lambda$ explicit.

Our starting point is the recognition that, by duality (cf. [66, Sec. 3.2]),

$$\mathcal{L}_{\chi^2}(x; P_0) = \inf_{\lambda \geq 0} \left\{ \mathcal{L}^\lambda_{\chi^2\text{-pen}}(x; P_0) + \lambda \rho \right\} = \inf_{\lambda \geq 0} \sup_{Q \ll P_0} \left\{ \mathbb{E}_Q \, \ell(x; S) - \lambda \left[ \mathrm{D}_{\chi^2}(Q, P_0) - \rho \right] \right\}$$

for any distribution $P_0$. For $0 \leq \underline{\lambda} \leq \overline{\lambda}$, we may thus consider the approximation

$$\mathcal{L}_{\chi^2[\underline{\lambda}, \overline{\lambda}]}(x; P_0) := \min_{\lambda \in [\underline{\lambda}, \overline{\lambda}]} f_\rho(x, \lambda) \text{ where } f_\rho(x, \lambda) := \mathcal{L}^\lambda_{\chi^2\text{-pen}}(x; P_0) + \lambda \rho.$$

By restricting $\lambda$ to an appropriate range, we can then approximate $\mathcal{L}_{\chi^2}$ by its truncated version, as the next lemma shows.

**Lemma 5.** *For all $P_0$, $\rho$ and $\epsilon$,*

$$\min_{x \in \mathcal{X}} \mathcal{L}_{\chi^2[\frac{\epsilon}{2\rho}, \frac{B}{\rho}]}(x; P_0) \leq \min_{x' \in \mathcal{X}} \mathcal{L}_{\chi^2}(x'; P_0) + \frac{\epsilon}{2}.$$

*Proof.* Le $x^\star, \lambda^\star = \arg\min_{x \in \mathcal{X}, \lambda \geq 0} \{f_\rho(x, \lambda)\}$, noting that $\min_{x' \in \mathcal{X}} \mathcal{L}_{\chi^2}(x'; P_0) = f_\rho(x^\star, \lambda^\star)$. For any $x, \lambda$ let $Q^\star_{x,\lambda}$ be the maximizing $Q$ in (12) for these values of $x, \lambda$. Moreover, let $\mathrm{D}(x, \lambda) = \mathrm{D}_{\chi^2}(Q^\star_{x,\lambda}, P_0)$. By Claim 4, for all $\lambda > B/\rho$ we have that $\mathrm{D}(x, \lambda) < \rho$, and consequently $\lambda > \lambda^\star$, i.e., $\lambda^\star \leq B/\rho$, and hence that upper bound has no impact on accuracy.

When in addition we have $\lambda^\star \geq \epsilon/(2\rho)$ then clearly $\min_{x \in \mathcal{X}} \mathcal{L}_{\chi^2[\frac{\epsilon}{2\rho}, \frac{B}{\rho}]}(x; P_0) = \mathcal{L}_{\chi^2[0, \infty]}(x; P_0) = \min_{x' \in \mathcal{X}} \mathcal{L}_{\chi^2}(x'; P_0)$. Otherwise, if $\lambda^\star < \epsilon/(2\rho) =: \lambda_\epsilon$ we may write

$$\begin{aligned}
\min_{x \in \mathcal{X}} \mathcal{L}_{\chi^2[\frac{\epsilon}{2\rho}, \frac{B}{\rho}]}(x; P_0) &\overset{(i)}{\leq} f_\rho(x^\star, \lambda_\epsilon) \overset{(ii)}{\leq} f_\rho(x^\star, \lambda^\star) + \left[ \tfrac{\partial}{\partial \lambda} f_\rho(x^\star, \lambda_\epsilon) \right](\lambda_\epsilon - \lambda^\star) \\
&= \min_{x' \in \mathcal{X}} \mathcal{L}_{\chi^2}(x'; P_0) + [\rho - \mathrm{D}(x^\star, \lambda_\epsilon)](\lambda_\epsilon - \lambda^\star) \\
&\overset{(iii)}{\leq} \min_{x' \in \mathcal{X}} \mathcal{L}_{\chi^2}(x'; P_0) + \lambda_\epsilon \rho = \min_{x' \in \mathcal{X}} \mathcal{L}_{\chi^2}(x'; P_0) + \frac{\epsilon}{2}.
\end{aligned}$$

Where we used $(i)$ that $x^\star$ and $\lambda_\epsilon$ are feasible points in the joint minimization of $f_\rho(x, \lambda)$ over $x \in \mathcal{X}$ and $\lambda \in [\lambda_\epsilon, B/\rho]$; $(ii)$ the convexity of $f$ in $\lambda$; and $(iii)$ the fact that $\mathrm{D}(x^\star, \lambda_\epsilon) \geq 0$ and $\lambda^\star \leq \lambda_\epsilon$. $\qquad\square$

Our strategy is therefore to jointly minimize $f_\rho(x, \lambda) = \mathcal{L}^\lambda_{\chi^2\text{-pen}}(x; P_0) + \lambda \rho$ over both $x \in \mathcal{X}$ and $\lambda \in [\underline{\lambda}, \overline{\lambda}]$ (rather than $[0, \infty]$), using the approximation guarantee in Lemma 5 to argue that the restriction of $\lambda$ will have limited effect on the quality of the resulting solution. We iterate the projected stochastic gradient method with the multi-level Monte Carlo (MLMC) gradient estimator (11) via

$$x_{t+1} = \Pi_\mathcal{X}\left( x_t - \gamma_x \widehat{\mathcal{M}}\left[ \nabla \mathcal{L}^{\lambda_t}_{\chi^2\text{-pen}}(x_t) \right] \right), \; \lambda_{t+1} = \Pi_{[\underline{\lambda}, \overline{\lambda}]}\left( \lambda_t - \gamma_\lambda \widehat{\mathcal{M}}\left[ \tfrac{\partial}{\partial \lambda} \mathcal{L}^{\lambda_t}_{\chi^2\text{-pen}}(x_t) + \rho \right] \right). \tag{56}$$

If we can bound the moments of the MLMC-approximated gradients $\widehat{\mathcal{M}}$, we can then leverage standard stochastic gradient analyses to prove convergence. We use the following bound.

**Lemma 6.** *We have*

$$\mathbb{E}\left(\widehat{\mathcal{M}}\left[\tfrac{\partial}{\partial\lambda}\mathcal{L}^\lambda_{\chi^2\text{-pen}}(x;\cdot)+\rho\right]\right)^2 \lesssim \frac{B^2}{\lambda^2}\left(1+\frac{B\log\frac{n}{n_0}}{\lambda n_0}\right)+\rho^2.$$

*Proof.* Recall the definition (51) of the MLMC estimator of a general $\mathsf{F}$ and the expression (52) for its second moment. Suppose that $\mathsf{F}(\cdot)=\mathsf{F}_1(\cdot)+\mathsf{F}_2(\cdot)+c$, where $c$ is a constant. Then

$$\mathbb{E}\|\widehat{\mathcal{D}}_k[\mathsf{F}]\|^2 = \mathbb{E}\|\widehat{\mathcal{D}}_k[\mathsf{F}_1+\mathsf{F}_2]\|^2 \le 2\,\mathbb{E}\|\widehat{\mathcal{D}}_k[\mathsf{F}_1]\|^2 + 2\,\mathbb{E}\|\widehat{\mathcal{D}}_k[\mathsf{F}_2]\|^2.$$

Consequently, by (52), we have

$$\mathbb{E}\|\widehat{\mathcal{M}}[\mathsf{F}]\|^2 \le 2c^2 + 2\,\mathbb{E}\|\widehat{\mathcal{M}}[\mathsf{F}_1]\|^2 + 2\,\mathbb{E}\|\widehat{\mathcal{M}}[\mathsf{F}_2]\|^2. \tag{57}$$

We apply this observation to $\widehat{\mathcal{M}}\left[\tfrac{\partial}{\partial\lambda}\mathcal{L}^\lambda_{\chi^2\text{-pen}}(x;\cdot)+\rho\right]$ by noting that

$$\tfrac{\partial}{\partial\lambda}\mathcal{L}^\lambda_{\chi^2\text{-pen}}(x;S^n_1) = -\mathrm{D}_{\chi^2}(q^\star;\tfrac{1}{n}\mathbf{1}) = \frac{1}{\lambda}\left(\mathcal{L}^\lambda_{\chi^2\text{-pen}}(x;S^n_1)-\frac{1}{n}\sum_{i\le n}q^\star_i\ell(x;S_i)\right).$$

Proposition 3' gives us the bound $\mathbb{E}\left(\widehat{\mathcal{M}}[\mathcal{L}^\lambda_{\chi^2\text{-pen}}(x;S^n_1)]\right)^2 \lesssim B^2\left(1+\frac{B}{\lambda n_0}\log\frac{n}{n_0}\right)$. Moreover, we have that

$$\mathbb{E}\left(\widehat{\mathcal{M}}\Big[\frac{1}{n}\sum_{i\le n}q^\star_i\ell(x;S_i)\Big]\right)^2 \lesssim B^2\left(1+\frac{B}{\lambda n_0}\log\frac{n}{n_0}\right)$$

By exactly the same argument that proves the gradient second moment bound in Proposition 3'. The result then follows by substituting into (57). $\qquad\square$

Therefore, we may find an $\epsilon$ approximate minimizer with complexity roughly $B^3\overline{\lambda}^2/(\underline{\lambda}^3\epsilon^2)$:

**Lemma 7.** *Fix $\epsilon \in (0,B)$ and $\overline{\lambda} \ge \underline{\lambda} > 0$. For a suitable setting of the parameters $n_0, n, T, \gamma_x$ and $\gamma_\lambda$, the average $\bar{x}_T = \sum_{t\le T} x_t$ of the iterates (56) satisfies $\mathbb{E}\,\mathcal{L}_{\chi^2[\underline{\lambda},\overline{\lambda}]}(\bar{x}_T;P_0) \le \min_{x\in\mathcal{X}}\mathcal{L}_{\chi^2[\underline{\lambda},\overline{\lambda}]}(x;P_0)+\epsilon$, with complexity*

$$\lesssim \left(1+\frac{B}{\underline{\lambda}}\right)\frac{(GR)^2+B^2\overline{\lambda}^2/\underline{\lambda}^2+\overline{\lambda}^2\rho^2}{\epsilon^2}\log^2\left(1+\frac{B}{\underline{\lambda}\epsilon}\right) \text{ with probability } \ge 1-\frac{\epsilon^2}{B^2}.$$

*Proof.* We take $n\asymp\frac{B}{\underline{\lambda}\epsilon}$ to guarantee bias below $\epsilon/2$ by Proposition 1, and we take $n_0\asymp\frac{B}{\underline{\lambda}}\log n$ to guarantee that

$$\Gamma^2_x := \sup_{x\in\mathcal{X},\lambda\in[\underline{\lambda},\overline{\lambda}]}\mathbb{E}\left\|\widehat{\mathcal{M}}\big[\nabla\mathcal{L}^\lambda_{\chi^2\text{-pen}}(x;\cdot)\big]\right\|^2 \lesssim G^2$$

and, by Lemma 6,

$$\Gamma^2_\lambda := \sup_{x\in\mathcal{X},\lambda\in[\underline{\lambda},\overline{\lambda}]}\mathbb{E}\left(\widehat{\mathcal{M}}\big[\tfrac{\partial}{\partial\lambda}\mathcal{L}^\lambda_{\chi^2\text{-pen}}(x;\cdot)+\rho\big]\right)^2 \lesssim \frac{B^2}{\underline{\lambda}^2}+\rho^2.$$

Let $\bar{\lambda}_T=\sum_{t\le T}\lambda_t$ be the average of the $\lambda$ iterates in (56). By appropriate choice of $\eta$ and $\eta'$ we guarantee (via Proposition 6) that

$$\mathbb{E}\,\mathcal{L}_{\chi^2[\underline{\lambda},\overline{\lambda}]}(\bar{x}_T;P_0) \le \mathbb{E}\,f_\rho(\bar{x}_T,\bar{\lambda}_T) \le \min_{x\in\mathcal{X},\lambda\in[\underline{\lambda},\overline{\lambda}]}f_\rho(x,\lambda)+\mathrm{err}_T = \min_{x\in\mathcal{X}}\mathcal{L}_{\chi^2[\underline{\lambda},\overline{\lambda}]}(x;P_0)+\mathrm{err}_T,$$

where

$$\mathrm{err}_T \lesssim \frac{\epsilon}{2}+\frac{\Gamma_x R+\Gamma_\lambda(\overline{\lambda}-\underline{\lambda})}{\sqrt{T}}.$$

Therefore, by taking

$$T \asymp \frac{\Gamma^2_x R^2+\Gamma^2_\lambda\overline{\lambda}^2}{\epsilon^2} \asymp \frac{(GR)^2+B^2\overline{\lambda}^2/\underline{\lambda}^2+\overline{\lambda}^2\rho^2}{\epsilon^2}$$

we guarantee that $\mathrm{err}_T \le \epsilon$, and the complexity bound follows from substituting $n_0, n$ and $T$ in the high probability upper bound $n_0\log_2\left(\frac{n}{n_0}\right)T+5\sqrt{(n\log n)^2+n_0 nT\log n}$ shown in Theorem 2. $\quad\square$

Directly substituting $\underline{\lambda} = \frac{\epsilon}{2\rho}$ and $\overline{\lambda} = \frac{B}{\rho}$ results in a guarantee scaling as $\epsilon^{-5}$, which is worse than the mini-batch rate of $\epsilon^{-4}$. To improve on this, we divide $[\frac{\epsilon}{2\rho}, \frac{B}{\rho}]$ into $K = \log_2 \frac{B}{\epsilon}$ sub-intervals $[\lambda^{(i+1)}, \lambda^{(i)}]$ satisfying $\lambda^{(i+1)}/\lambda^{(i)} = 2$. We then perform the stochastic gradient method (56) on each of these intervals $[\lambda^{(i+1)}, \lambda^{(i)}]$ in turn, yielding estimates $\bar{x}^{(i)}$ that are each $\lesssim \epsilon$-suboptimal for the approximate objective $\mathcal{L}_{\chi^2[\lambda^{(i+1)}, \lambda^{(i)}]}$. Using the bounded ratio $\lambda^{(i+1)}/\lambda^{(i)} = 2$, this requires complexity roughly $1/(\lambda^{(i+1)}\epsilon^2) \lesssim \rho/\epsilon^3$, giving the following theorem.

**Theorem 4.** *Fix $\epsilon \in (0, B)$, and for $i \in \mathbb{N}$ set $\lambda^{(i)} = \frac{B}{\rho}2^{-i+1}$ and let $\bar{x}^{(i)}$ be an $\epsilon/2$-approximate minimizer of $\mathcal{L}_{\chi^2[\lambda^{(i+1)}, \lambda^{(i)}]}$ computed via stochastic gradient iterations according to Lemma 7. Then, for $1 + K = \lceil \log_2 \frac{2B}{\epsilon} \rceil$ and some $i^\star \leq K$ we have $\mathbb{E}\, \mathcal{L}_{\chi^2}(\bar{x}^{(i^\star)}; P_0) \leq \min_{x \in \mathcal{X}} \mathcal{L}_{\chi^2}(x; P_0) + \epsilon$. Computing $\bar{x}^{(1)}, \ldots, \bar{x}^{(K)}$ requires a total number of $\nabla \ell$ evaluations*

$$\lesssim \frac{(GR)^2(\rho B + \epsilon \log_2 \frac{B}{\epsilon})}{\epsilon^3} \log^2\left(1 + \frac{\rho B}{\epsilon^2}\right) \quad \text{with probability} \ \geq 1 - \frac{\epsilon}{B}.$$

*Proof.* By Lemma 7, finding an $\epsilon$ approximate solution in the interval $[\lambda^{(i+1)}, \lambda^{(i)}]$ requires

$$\lesssim \left(1 + \frac{\rho B}{2^{K-i}\epsilon}\right)\frac{(GR)^2 + B^2}{\epsilon^2}\log^2\left(1 + \frac{\rho B}{\epsilon^2}\right)$$

gradient computations, where we have used $\lambda^{(i)}/\lambda^{(i+1)} \leq 2$, $\lambda^{(i)} \leq \frac{B}{\rho}$, and $\lambda^{(i+1)} \geq \frac{\epsilon}{2\rho}2^{K-i}$. Summing over $i$ (and applying a union bound) gives the claimed guarantee. Since the minimizer of $f_\rho(x, \lambda)$ over $x \in \mathcal{X}$ and $\lambda \in [\frac{\epsilon}{2\rho}, \frac{B}{\rho}]$ is equivalent is identical to its minimizer in one of the intervals $[\lambda^{(i+1)}, \lambda^{(i)}]$ for $i \leq K$, the result follows from Lemma 5. $\square$

The index $i^\star$ is independent of randomness in our procedure, but we do not know it in advance. Instead, we may estimate the minimized objective for each $i$ and select the index with the lowest estimate. Let $\hat{\lambda}^{(i)}$ be the average of the $\lambda$ iterations of our stochastic gradient method (56) for a particular interval $[\lambda^{(i+1)}, \lambda^{(i)}]$. Our bias and variance bounds on $\mathcal{L}_{\chi^2\text{-pen}}$ (Proposition 1 and Proposition 2' in the appendix) imply the we can estimate[5] $f_\rho(\bar{x}^{(i)}, \hat{\lambda}^{(i)})$ to accuracy $\lesssim \epsilon$ with a sample of size $\asymp B^2/(\lambda^{(i)}\epsilon^2) \asymp 2^{i-K}B^3\rho\epsilon^{-3}$. Taking $i^\star$ to be the index $i$ minimizing this estimate, it is straightforward to argue that $\mathbb{E}\, \mathcal{L}_{\chi^2}(\bar{x}^{(i^\star)}; P_0) - \min_{x \in \mathcal{X}} \mathcal{L}_{\chi^2}(x; P_0) \lesssim \epsilon$. Therefore, the cost of selecting the best $i$ is at most the cost of performing the optimization.

Theorem 4 provides a rigorous guarantee on the complexity of minimizing $\mathcal{L}_{\chi^2}$ with a fixed constraint $\rho$ by optimizing the parameter $\lambda$ of $\mathcal{L}_{\chi^2\text{-pen}}^\lambda$. In practice, we usually have no prior knowledge of $\rho$, so it will often make sense to directly tune $\lambda$ according to validation criteria rather than a target $\rho$. We also note that Duchi and Namkoong [22] prove a lower bound of order $\rho\epsilon^{-2}$, which is smaller than our $\rho\epsilon^{-3}$ rate. Establishing the optimal rate for this problem remains an open question.

# F   Experiments

In this section we give a detailed description of our experiments. We begin with a description of the problems we study (Section F.1) followed by our hyperparameter settings (Section F.2) and brief remarks about our PyTorch implementation (Section F.3). Then, in Sections F.4 and F.5 we present and discuss our results in detail, including speed-up factors over full-batch optimization, a study of the generalization impacts of the DRO objective, and direct empirical evaluation of the bias $\mathcal{L} - \overline{\mathcal{L}}$ which we bound in Proposition 1.

## F.1   Dataset description

**Digits.** We consider the MNIST handwritten digit recognition dataset with the standard train/test split into with $6 \cdot 10^4$ and $10^4$ training and test images, respectively. There are 10 classes corresponding to the ten digits. We augment the training set with $N_{\text{typed}} = 600$ randomly chosen digits from the characters dataset [19], i.e., 1% of the hand-written digits. Our test set includes the MNIST test set as

well as a class-balanced sample of 8K typed digits not included in the training data. Creating an 8K image test set requires that we disregard the original test/train split of [19], but is important in order to make estimates of per-class accuracy reliable. To featurize our data, we train a small convolutional Neural Network (two convolutional layers, two fully-connected layers with ReLU activation function) with a standard ERM objective and 10 epochs of SGM on the MNIST training set (with no typed digits). For both handwritten and typed digits, we use the activations of the last layer as the feature vector.

We perform DRO to learn a linear classifier $x$ on our features, taking the loss $\ell$ to be multi-class logarithmic loss with a quadratic regularization term on $x$ (the weight part only, not the bias), namely, for a data point $s = (z, y)$ with $z \in \mathbb{R}^d, y \in [C]$ (with $C$ the number of classes) and regularization strength $\mu \geq 0$, we use

$$\ell([x, b]; (z, y)) := \log\left(\sum_{c=1}^{C} \exp(\langle x_c - x_y, z\rangle + b_c - b_y)\right) + \frac{\mu}{2}\sum_{c=1}^{C}\|x_c\|_2^2,$$

where $x \in \mathbb{R}^{C \times d}, b \in \mathbb{R}^C$ and $x_c$ denotes the $c$-th row of $x$. As the generalization metric, we report accuracy and log loss on the worst sub-group of the data—where a sub-group corresponds to a tuple (subpopulation, class), e.g., (typed, 9).

**ImageNet.** The ImageNet dataset comprises of $1.2 \cdot 10^6$ training images and $5 \cdot 10^4$ test images with 1000 different classes. We featurize the dataset using a pre-trained ResNet-50 [35] (trained on ImageNet itself with an ERM objective). We use those features as the input to a linear classifier, with regularized multi-class logarithmic loss as in the previous experiment. As the robust generalization metric, we report the average loss and accuracy on the 10 classes with highest test loss.

### F.2 Hyperparameter tuning

We fix the budget of our algorithms to 300 epochs for Digits and 30 epochs for ImageNet, where an epoch corresponds to $N$ computations of $\nabla\ell$, where $N$ is the training set size. For all (mini)-batch methods we use Nesterov acceleration (50) with constant momentum $\omega = 0.9$; we did not carefully tune this parameter but did observe it performs better than no momentum. For MLMC using no momentum ($\omega = 0$) performs slightly better than momentum 0.9, so we use no momentum in this case. We also perform iterate averaging with the scheme of Shamir and Zhang [65] with parameter 3 (roughly averaging over the last third of the iterates). Our experiments with CVaR use $\nabla\mathcal{L}_{\text{CVaR}}$ rather than $\nabla\mathcal{L}_{\text{kl-CVaR}}$, in contrast to our theory; we leave empirical exploration of entropy smoothing for CVaR to future work.

**Stepsizes.** We tune our stepsizes with a coarse-to-fine strategy. More precisely, for each stepsize in $\{10^i\}_{-5 \leq i \leq 0}$, we perform a single run of the experiment, and pick the best two stepsizes in terms of the final training value. For these two stepsizes, we evaluate $\frac{\eta}{2}, 2\eta$ and select the stepsize that gives the best value of the training loss. For this final stepsize, we repeat the experiments with 5 different seeds (affecting weight initialization and mini batch samples but not the dataset structure) and report the minimum and maximum across seeds at each iteration. We select all the stepsizes in our experiments using this strategy, except for batch size $n = 10$ in ImageNet where we extrapolated the stepsize from other batch sizes. Table 3 summarizes our step size choices—for batch sizes up to 5K we see a clear linear relationship between the batch size and optimal step size.

$\ell_2$**-regularization and parameters of the robust loss** We choose the strength of the regularizer in the set $\{0, 10^{-5}, \ldots, 10^{-1}\}$. For each robust loss, we consider an appropriate grid of either the size of the uncertainty set ($\alpha$ and $\rho$) or the strength of the penalty ($\lambda$). We evaluate each configuration ($\ell_2$ regularization and robust loss parameters) with the stepsizes from the coarse grid and pick the configuration that achieves a good trade-off in terms worst-subgroup and average-case generalization. For simplicity, we choose the same regularization strenght for all the robust losses—$\mu = 10^{-3}$ for ImageNet and $\mu = 10^{-2}$ for Digits. For ERM, we choose the two values of $\ell_2$ regularization that optimize either worst subgroup loss or worst subgroup accuracy. That is, for ImageNet we tune the $\ell_2$ regularization for the best result on either the worst 10 classes loss and worst 10 classes accuracy respectively, and for Digits we choose the values that optimize loss/accuracy on the hardest typed class—for both experiments, this results in $\mu \in \{10^{-4}, 10^{-3}\}$ for ERM.

### F.3 PyTorch Integration

Figure 2 illustrates our integration of DRO into PyTorch. Users simply define the robust loss they wish to use (in the example $\mathcal{L}_{\chi^2}$ with $\rho = 1$) and feed the loss for the examples in the batch to the robust

| | | ImageNet | | | Digits | | |
|---|---|---|---|---|---|---|---|
| Algorithm | | $\mathcal{L}_{\text{CVaR}}$ $\alpha = 0.1$ | $\mathcal{L}_{\chi^2}$ $\rho = 1$ | $\mathcal{L}_{\chi^2\text{-pen}}$ $\lambda = 0.4$ | $\mathcal{L}_{\text{CVaR}}$ $\alpha = 0.02$ | $\mathcal{L}_{\chi^2}$ $\rho = 1$ | $\mathcal{L}_{\chi^2\text{-pen}}$ $\lambda = 0.05$ |
| Batch | $n = 10$ | $1 \cdot 10^{-4}$ | $2 \cdot 10^{-4}$ | $2 \cdot 10^{-4}$ | $1 \cdot 10^{-4}$ | $5 \cdot 10^{-5}$ | $5 \cdot 10^{-5}$ |
| | $n = 50$ | $5 \cdot 10^{-4}$ | $1 \cdot 10^{-3}$ | $1 \cdot 10^{-3}$ | $1 \cdot 10^{-4}$ | $2 \cdot 10^{-4}$ | $1 \cdot 10^{-4}$ |
| | $n = 500$ | $5 \cdot 10^{-3}$ | $1 \cdot 10^{-2}$ | $1 \cdot 10^{-2}$ | $1 \cdot 10^{-3}$ | $2 \cdot 10^{-3}$ | $1 \cdot 10^{-3}$ |
| | $n = 5K$ | $5 \cdot 10^{-2}$ | $1 \cdot 10^{-1}$ | $1 \cdot 10^{-1}$ | $5 \cdot 10^{-3}$ | $2 \cdot 10^{-2}$ | $1 \cdot 10^{-2}$ |
| | $n = 50K$ | $2 \cdot 10^{-1}$ | $5 \cdot 10^{-1}$ | $2 \cdot 10^{-1}$ | – | – | – |
| | $n = 150K$ | $5 \cdot 10^{-1}$ | $5 \cdot 10^{-1}$ | $5 \cdot 10^{-1}$ | – | – | – |
| MLMC | $n_0 = 10$ | $1 \cdot 10^{-3}$ | $2 \cdot 10^{-3}$ | $2 \cdot 10^{-3}$ | $5 \cdot 10^{-4}$ | $5 \cdot 10^{-4}$ | $5 \cdot 10^{-4}$ |
| Full-batch | | $5 \cdot 10^{-1}$ | $5 \cdot 10^{-1}$ | $5 \cdot 10^{-1}$ | $1 \cdot 10^{-2}$ | $2 \cdot 10^{-2}$ | $1 \cdot 10^{-2}$ |

**Table 3.** Stepsizes for the experiments we present in this work. We use momentum 0.9 for all configurations except MLMC, where we do not use momentum. We select the stepsizes according to the 'coarse-to-fine' strategy we describe in this section.

```
1  from robust_losses import RobustLoss
2
3  # we define the usual variables but also our robust loss
4  model = ...
5  criterion = ...
6  robust_loss = RobustLoss(geometry='chi-square', size=1.0)
7  # [...]
8  # training loop
9      outputs = model(inputs)
10     if not robust:
11         loss = criterion(outputs, targets).backward()
12     else:
13         loss = robust_loss(
14             criterion(outputs, targets, reduction='none')
15         ).backward()
16  # rest of the training loop
```

**Figure 2.** An example training loop in PyTorch where one can decide to use the robust training objective at the cost of three extra lines of code (lines 1, 6 and 13).

layer. While our current implementation only supports the robust objectives we analyze—namely, CVaR, KL-regularized CVaR, constrained-$\chi^2$ and penalized-$\chi^2$—it is easy to extend to other choices of $\phi$ and $\psi$.

## F.4 Experiment results

We complement the training curves in Figure 1 with comparisons of robust generalization metrics and training efficiency. In Figures 3 and 4 we show the training curves of Figure 1 along with two "robust" generalization metrics and two "average" performance metrics. For Digits, we consider the loss and accuracy on the worst sub-group—typically the typed digit 9—as the robust generalization metrics. For ImageNet, we look at the average loss (resp. accuracy) on the 10 labels with highest loss (resp. lowest accuracy). In each figure we also show the values achieved by ERM with two different regularization strengths chosen to optimize either loss or accuracy on the worst-subgroup. In Tables 4 and 5 we compare the number of epochs the various algorithms require to reach a training loss within 2% of the minimal value found across all runs. To achieve such convergence with the full batch method we run it for much longer: 30K epochs for Digits and 1K epochs for ImageNet.

## F.5 Discussion

### F.5.1 Generalization performance

We now take a closer look at the curves presented in Figures 3 and 4. We first note that, in the context of machine learning, one does not wish to reach the minimum of the training objective but rather find a model that achieves good generalization performance. From that perspective, we observe that mini-batch methods achieve their best generalization performance in a shorter time than necessary to

**Figure 3.** Detailed results from our digit recognition experiment. Shaded areas indicate range of variability across 5 repetitions (minimum to maximum), and the zoomed-in regions highlight the (often very small) "bias floor" of small batch sizes.

converge on the training objective, e.g., less than 50 epochs for CVaR on Digits when the training objective always requires more than 115 epochs.

In the case of Digits, we observe that DRO achieves a better trade-off than ERM in all settings. More precisely, DRO achieves better worst sub-group loss and accuracy than either of the ERM runs with no visible degradation in average accuracy and slightly worse average loss. We observe a similar trend in the case of ImageNet, albeit with a more visible degradation in average loss and accuracy.

We note that in the Digits experiment batch size $n = 10$ has generalization performance more similar to ERM. This is an expected by-product of the bias inherent in small batch size, as in the edge case $n = 1$, the mini-batch method degenerates to ERM.

Hu et al. [38] observe that applying DRO objectives of the form (12) directly on the 0-1 loss amounts to a simple monotonic transformation of the average accuracy, and is therefore equivalent to minimizing average accuracy. Thus, in as far as the logarithmic loss is a surrogate to the 0-1 loss (which is arguably the case in near realizable-settings), DRO might not provide improvements in robust accuracy. This is consistent with the observations in our experiments, where we see only small effects on the accuracy in the Digits experiments (which is close to realizable), and a somewhat more pronounced but still modest effect on ImageNet (which is not quite realizable, as the training accuracy is below 90%). Nevertheless, these observation do not preclude DRO from improvement the subpopulation test loss itself, as we see in our experiments: for Digits DRO provides between between 17.5% and 27% reduction in worst subgroup loss compared to ERM, and for ImageNet the reduction is a more modest 5.6% and 9%. While the common practice in machine learning is to view accuracy as the more important performance metric, logarithmic loss is also operationally

**Figure 4.** Detailed results from our ImageNet classification experiment. Shaded areas indicate range of variability across 5 repetitions (minimum to maximum), and the zoomed-in regions highlight the (often very small) "bias floor" of small batch sizes.

|  | Number of epochs to 2% of opt | | | | Speed-up |
|---|---|---|---|---|---|
|  | $n = 50$ | $n = 500$ | $n = 5K$ | Full-batch | vs. full-batch |
| $\mathcal{L}_{\text{CVaR}}, \alpha = 0.02$ | $189 \pm 3$ | $\mathbf{115 \pm 1}$ | $193 \pm 4$ | $1035$ | $9.0\times$ |
| $\mathcal{L}_{\chi^2}, \rho = 1$ | $\infty$ | $74 \pm 1$ | $\mathbf{60 \pm 3}$ | $570$ | $9.5\times$ |
| $\mathcal{L}_{\chi^2\text{-pen}}, \lambda = 0.05$ | $107 \pm 1$ | $\mathbf{104 \pm 1}$ | $131 \pm 5$ | $1680$ | $16.2\times$ |

**Table 4.** Empirical complexity for the Digits experiment in terms of number of epochs required to reach within 2% of the optimal training objective value, averaged across 5 seeds $\pm$ one standard deviation. (For the full-batch experiments we only ran one seed). The "speed-up" column gives the ratio between the full batch complexity and the best mini-batch complexity.

meaningful, as it measures the calibration of the model predictions. Thus, DRO is potentially helpful in situations where robust precise uncertainty estimates are important.

We remark that approaches that explicitly target the subgroups on which we measure the generalization [e.g., 61] will likely perform better than DRO. However, in contrast to these methods DRO is agnostic to the subgroup definition—except that we use a subgroup validation set in order to tune its uncertainty set size—and therefore requires less data annotation.

### F.5.2 Optimization performance

As Figure 1 and Tables 4 and 5 indicate, mini-batch methods converge significantly faster than full-batch. We also see that, while theoretically optimal, MLMC methods are slower to converge.

| | Number of epochs to 2% of opt | | | | | | | Speed-up |
|---|---|---|---|---|---|---|---|---|
| $n =$ | 10 | 50 | 500 | 5K | 50K | 150K | Full-batch | vs. full-batch |
| $\mathcal{L}_{\text{CVaR}}, \alpha = 0.1$ | 20 | 10 | **9** | **9** | 19 | - | 245 | 27× |
| $\mathcal{L}_{\chi^2}, \rho = 1$ | 6 | **5** | **5** | **5** | 8 ± 1 | 23 | 160 | 32× |
| $\mathcal{L}_{\chi^2\text{-pen}}, \lambda = 0.4$ | 7 | **5** | **5** | **5** | 22 | 26 | 180 | 36× |

**Table 5.** Empirical complexity for the ImageNet experiment in terms of number of epochs required to reach within 2% of the optimal training objective value, averaged across 5 seeds ± one standard deviation, whenever it is not zero. (For the full-batch experiments we only ran one seed). The "speed-up" column gives the ratio between the full batch complexity and the best mini-batch complexity.

| | | ImageNet times [minutes] | | | Digits times [minutes] | | |
|---|---|---|---|---|---|---|---|
| Algorithm | | per epoch | to 2% of opt | # epochs | per epoch | to 2% of opt | # epochs |
| Batch | $n = 10$ | 120 ± 5 | 850 ± 30 | 7 | 0.80 ± 0.1 | ∞ | ∞ |
| | $n = 50$ | 23 ± 0.7 | 116 ± 4 | **5** | 0.23 ± 0.01 | 24 ± 1 | 107 ± 1 |
| | $n = 500$ | 5.9 ± 0.2 | 29 ± 1 | **5** | 0.056 ± 0.004 | 5.8 ± 0.4 | **104 ± 1** |
| | $n = 5K$ | 3.3 ± 0.04 | **16.5 ± 0.2** | **5** | 0.033 ± 0.004 | **4.4 ± 0.7** | 131 ± 6 |
| | $n = 50K$ | 2.2 ± 0.03 | 50 ± 0.9 | 22 | – | – | – |
| | $n = 150K$ | 2.1 ± 0.03 | 55 ± 0.7 | 26 | – | – | – |
| MLMC | $n_0 = 10$ | 16 ± 1 | ∞ | ∞ | 0.34 ± 0.02 | ∞ | ∞ |
| Full-batch | | 2.1 | 380 | 180 | 0.022 | 37.0 | 1680 |

**Table 6.** Comparison wallclock time (in minutes) of the different algorithms, in terms of time per epoch and time to reach within 2% of the best training loss. In the last two columns, we report the number of epochs required to reach within 2% of the best training loss. We report ∞ for configurations that do not reach the sub-optimality goal for the duration of the experiment, and omit standard deviations when then they are 0.

Furthermore, the bias is empirically much smaller than what the theory predicts and setting the batch size as small as 50 guarantees negligible bias; we investigate this further below. As the theory predicts, the MLMC method (for corresponding values of $n_0$) effectively counteracts this bias, and is able to converge to the optimal value even when $n_0$ is 10.

We also note that the effect of batch size on the depth of the algorithm (number of iterations) is remarkably consistent with the theoretical prediction of the variance-based analysis in Section 3: for smaller batch sizes the number of steps is roughly inversely proportional to the batch size, and the total amount of work is constant. The best stepsize also grows linearly with the batch size (see Table 3). As batch sizes grow, the best stepsize plateaus and the number of steps required for convergence also stops decreasing with the batch size, making the total work become larger.

**Runtime comparison.** In Table 6 we report the gradient complexity and wallclock time to reach accuracy within 2% of the optimal value. For brevity, we show it for a single robust objective (penalized-$\chi^2$), but we observe that similar results across robust objectives. We note that for small batch sizes the time per epoch is significantly larger than for larger batch sizes, this due in part to parallelization in evaluating $\ell$ and $\nabla\ell$ and in part to logging and Python interpreter overhead, which increase linearly with the number of iterations. However, these effects diminish as the batch size grows, and for batch size 5K the wallclock time to reach an accurate solution is an order of magnitude smaller than with the full-batch method. We run our experiments with 4 Intel Xeon E5-2699 CPUs and 12–32Gb of memory. Increasing the number of CPUs or using GPUs would allow for greater parallelism and improve the runtime at greater batch sizes. However, increasing the model complexity (e.g., to a deep neural network) would have the opposite effect. Using 4 CPUs for linear classification gives roughly the same range of feasible batch sizes as a ResNet-50 on large GPU arrays.

**Bias analysis.** Figure 1 shows that even for small batch sizes—where the guarantees of Proposition 1 are essentially vacuous—stochastic gradient steps with the mini-batch gradient estimator find solutions very close to optimal. There could be two explanations for this finding: (a) $\mathcal{L}$ and $\overline{\mathcal{L}}$ are actually much closer to each other than the theory predicts, or (b) $\mathcal{L}$ and $\overline{\mathcal{L}}$ are far apart as expected, but still their minimizers are close.

**Figure 5.** Evaluation of the bias $\mathcal{L}(\bar{x}_T; P_0) - \overline{\mathcal{L}}(\bar{x}_T; n)$ at the last iterate $\bar{x}_T$ of the experiments in Figure 1, for different batch sizes $n$. (These batch sizes $n$ are not the same as the mini-batch size used to compute $\bar{x}_T$; we take the latter to be 10). Error bars indicate a 95% confidence interval computed using the bootstrap.

To test hypothesis (a), we examine the loss values at the last iterate $\bar{x}_T$ of our Digits and ImageNet experiments with mini-batch size 10. For each objective, we estimate $\overline{\mathcal{L}}(\bar{x}_T; n)$ for various values of $n$ by averaging 50K evaluations of $\mathcal{L}(\bar{x}_T; S_1^n)$, and use it to compute an estimate of the bias $\mathcal{L}(\bar{x}_T; P_0) - \overline{\mathcal{L}}(\bar{x}_T; n)$.[6] In Figure 5 we plot the bias estimate against the mini-batch size $n$. We see that hypothesis (a) is false: for both ImageNet and Digits, the difference $\mathcal{L}(\bar{x}_T; P_0) - \overline{\mathcal{L}}(\bar{x}_T; n)$ is quite large at small $n$, as our upper bounds and matching lower bounds in the Bernoulli case would suggest. We also see that the bias decays as $1/n$ in all cases except for $\chi^2$ constraint in Digits; this is again consistent with our theory as we expect the inverse-cdf assumption to be relevant in practice and particularly for CVaR where it only needs to hold around the $1 - \alpha$ quantile. We conclude that despite the significant bias at small batch size $n$, approximate minimizers of $\overline{\mathcal{L}}(x; n)$ are also approximate minimizers of $\mathcal{L}(x; P_0)$. This is possibly due to the fact that the bias $\mathcal{L}(x; P_0) - \overline{\mathcal{L}}(x; n)$ is nearly constant as a function of $x$. We leave further study of this hypothesis to future work.

### F.6 Comparison with alternative optimization methods

We complement the worst-case complexity comparison in Table 1 by repeating our experiments with two alternative optimization methods: dual SGM and primal-dual methods.

#### F.6.1 Comparison with dual SGM

**Experiment description.** Recall the dual SGM method we describe and analyze in Section A.3. The complexity guarantees of dual SGM depend quadratically on the size of the uncertainty set—scaling with $\alpha^{-2}$ for CVaR and with $\lambda^{-2}$ for the penalized version of the $\chi^2$ objective. In contrast, our theory predicts that the method we propose have an optimal linear dependence on the size of the uncertainty set. Here we empirically test this prediction on the Digits experiment. To do so, we compare the performance of our proposed mini-batch method with dual SGM for uncertainty sets of increasing size. For CVaR we consider

$$\alpha \in \{0.02, 0.006, 0.002, 0.0006, 0.0002\},$$

and for penalized $\chi^2$ we consider

$$\lambda \in \{0.05, 0.015, 0.005, 0.0015, 0.0005\}.$$

**Parameter tuning.** For each uncertainty set size, we jointly tune the stepsizes $\gamma_x$ and $\gamma_\eta$ over the following grids

$$\gamma_x \in \{1 \cdot 10^{-i}, 3 \cdot 10^{-i}\}_{3 \le i \le 6}, \ \gamma_\eta \in \{1 \cdot 10^{-i}\}_{2 \le i \le 5}.$$

We choose a coarser grid for $\gamma_\eta$ as we noticed that the value of $\gamma_\eta$ had a marginal influence on the final performance. For both the mini-batch algorithm and dual SGM, we pick the batch size $n = 500$ We follow the same averaging scheme and momentum as in our previous experiments.

**Figure 6.** Comparison of batch methods to dual SGM on the digits experiments for increasing sizes of uncertainty set sizes or regularization. We observe that as the size grows, dual SGM performs increasingly worse.

**Discussion of results.** We plot the results of the experiment in Figure 6. As the theory predicts, when the size of the uncertainty set grows, dual SGM performs significantly worse than batch methods. Conversely, as expected, for small uncertainty sets dual SGM performs on par with the mini-batch method. We empirically observe that the performance of dual SGM depends only weakly on the choice of $\gamma_\eta$. As a result, dual SGM is not much more difficult to tune than the mini-batch method.

### F.6.2 Comparison with primal-dual methods

**Experiment description.** We now turn to primal-dual methods, whose complexity guarantees scale as $\epsilon^{-2}$ but are linear in $N$, and are therefore expected to become less efficient as the size of the training set grows. To test this prediction, we repeat our Digits and ImageNet experiments (with $N = 60.6\text{K}$ and $N = 1.2\text{M}$, respectively) using these alternative methods for the contrained-$\chi^2$ and CVaR objectives. We then compare their performance to that of gradient methods with our mini-batch estimator.

**Method description.** Primal-dual methods maintain an iterate sequence $\{x_t, q_t\}_{t \in \mathbb{N}}$, where $q_t \in \mathcal{U}(P_0) \subset \Delta^N$ represent an online estimate of the distribution $q$ attaining the maximum in (3) at $x_1, \ldots, x_t$. To compute $x_{t+1}, q_{t+1}$, we sample a batch of $n$ indices $J_1^n$ drawn independently from $q_t$, and (denoting $S_i = s_{J_i}$) estimate the gradient of $\sum_{i=1}^N q_i \ell(x; s_i)$ with respect to $x$ and $q$ as follows:

$$\tilde{g}_t^x = \frac{1}{n}\sum_{i=1}^n \nabla \ell(x_t; S_i) \text{ and } [\tilde{g}_t^q]_j = \frac{1}{n}\sum_{i=1}^n \frac{1}{q_{J_i}}\ell(x_t; S_i)\mathbf{1}_{\{J_i=j\}}.$$

To compute $x_{t+1}$ from $x_t$ and $\tilde{g}_t^x$ we apply the same stochastic gradient scheme we use in our previous experiments (Nesterov momentum 0.9).[7] We also use the averaging scheme in [65] with parameter 3 as before. To compute $q_{t+1}$ from $q_t$ and $\tilde{g}_t^q$ we apply a mirror descent step. For the constrained $\chi^2$ problem the step is of the form

$$q_{t+1} = \underset{q: D_{\chi^2}(q,\frac{1}{N}\mathbf{1})\leq\rho}{\arg\max}\left\{\langle q, \gamma_q\tilde{g}_t^q\rangle + \frac{1}{2}\|q-q_t\|^2\right\} = \underset{q\in\Delta^N:\|q-\frac{1}{N}\mathbf{1}\|^2\leq 2\rho/n}{\arg\min}\|q-(q_t+\gamma_q\tilde{g}_t^q)\|^2, \quad (58)$$

i.e., a Euclidean projection of the unconstrained gradient step on $q_t$ to the uncertainty set. For the CVaR problem, the step is of the form

$$q_{t+1} = \underset{q\in\Delta^N:\|q\|_\infty\leq\frac{1}{\alpha N}}{\arg\max}\left\{\langle q, \text{clip}(\gamma_q\tilde{g}_t^q)\rangle + D_{\text{kl}}(q,q_t)\right\}, \quad (59)$$

where $\text{clip}(x)$ is the Euclidean projection of $x$ to $[-1,1]^N$.

The $\chi^2$ step is essentially the same as in [47], while the CvaR step is different from the proposal by Curi et al. [17]. Nevertheless, local norms regret analysis [13, 62] readily shows that with appropriate $\gamma_x$ and $\gamma_q$ the step (59) allows us to find $\epsilon$-optimal solutions within $\lesssim \frac{N\log\frac{1}{\alpha}B^2+G^2R^2}{\epsilon^2}$

**Figure 7.** Comparison of batch methods to primal-dual methods. We observe that the primal-dual methods are more efficient on the Digits experiment, but the trend reverses on the large-scale ImageNet experiment.

iterations, similarly to the guarantee that Curi et al. [17] show for a computationally intractable determinantal point process scheme. They also propose a tractable approximation for this scheme, but do not prove that it converges to the solution of the CVaR problem.

**Parameter tuning.** For every training task we jointly tune the parameters $\gamma_x$ and $\gamma_q$. We tune $\gamma_x$ over the values $10^{-i}$, $2 \cdot 10^{-i}$ and $5 \cdot 10^{-i}$ for $i \geq 1$ (similarly to our previous experiments) and we tune $\gamma_q$ over the values $10^{-i}$ and $3 \cdot 10^{-i}$ for $i \geq 1$. The best-performing values of $(\gamma_x, \gamma_q)$ are $(0.02, 0.003)$ for Digits/CVaR; $(0.02, 3 \cdot 10^{-7})$ for Digits/$\chi^2$; $(0.05, 3 \cdot 10^{-5})$ for ImageNet/CVaR; and $(0.02, 3 \cdot 10^{-11})$ for ImageNet/$\chi^2$. We use batch size $n = 500$ throughout.

**Discussion of results.** Figure 7 compares primal-dual and mini-batch primal methods with the best-performing hyperparameters, for two datasets and two objectives. For the Digits experiment, the primal-dual method perform better that the primal-only method (for $\chi^2$ significantly so). This may appear surprising, since the primal-dual complexity guarantees are larger by an additional factor of $N = 60.6K$ for this dataset. However, a closer look at the analysis of primal-dual methods shows that the term $NB^2$ is actually an upper bound on $\sum_{i=1}^{N}[\ell(x; s_i)]^2$ at $x = x_1, x_2, \ldots$. As the method converges, many data points are correctly classified with high confidence and therefore have very low value of $[\ell(x; s_i)]^2$. Hence, a more realistic complexity estimate would replace $N$ by the number of incorrectly classified training points, which for Digits is quite small (less than 100). Moreover, we observe that the optimal value of $\gamma_x$ for primal-dual methods is significantly larger than the corresponding step size for the primal-only method, likely because $\tilde{g}^x$ gives uniform weights to each $s_i$ as opposed to the adversarial weight of the primal-only method. The larger step sizes enable more rapid optimization over $x$.

For the larger-scale ImageNet experiment, the primal-only method significantly outperforms the primal-dual method. This is consistent with the above discussion, since here the number of misclassified training examples is large (more than 100K).

As an additional illustration of the superior scalability of primal-only method, consider a thought experiment where we replicate each element in our dataset $m$ times to form a new dataset of size $mN$. Clearly, this will have no impact on the primal-only method. In contrast, the norm of $\tilde{g}^q$ will grow by a factor of $m$, and we may expect the complexity of the method to increase by that factor as well.

Finally, we remark that tuning the primal-dual method is considerably more difficult than tuning the primal-only method. In addition to having two learning rates to search over, using an overly large value for $\gamma_q$ typically causes the algorithm to converge to a suboptimal point rather than diverge. Therefore, the common procedure of decreasing the learning rate until divergence no longer occurs will fail for the primal-dual method.