[Reviews · NeurIPS 2020]

Review 1

Summary and Contributions: The paper studies the use of batch stochastic gradient methods to solve large scale DRO problems. In these scenarios, we face two problems: (1) Stochastic gradient estimates of DRO problems are biased; (2) Due to the size of large-scale problems, the convergence rate of the methods used to tackle them should not depend on either the number of parameters d or number of training examples N. The authors tackled problem (1) by defining a surrogate objective for which the gradient estimates are unbiased. Then, by carefully bounding the difference between the true and surrogate objectives as a function of the batch size n, the authors are able to give optimality bounds for the true cost by optimizing the surrogate cost, using a large enough batch size. Moreover, for some classes of robust risks, the authors also bound the variance of the gradient estimates. This allows them to use an accelerated version of the stochastic gradient method which achieves tighter convergence bounds. Regarding problem (2), all the bounds given do not depend on either d or N, which makes the results suitable for large-scale applications. The authors also give lower bounds for the classes of DRO problems studied, proving the (sub)optimality of the proposed methods. Finally, these methods are used in experimental setups, which apparently corroborate with the theoretical results presented. %%% post rebuttal comments: %%% I appreciate the authors' effort to prepare the response letter. The authors satisfactorily addressed my comments concerning the bias in gradient estimates and numerical comparison with the existing approaches. I am happy with the response and increase my score to 7. %%%%%%%%%%%%%%%%%%%

Strengths: (1) The paper tackles an interesting problem. DRO, as well as methods for large-scale optimization (e.g. stochastic gradient methods), have increasingly gained the attention of the ML research community for the past few years. Developing methods to solve large-scale DRO problems efficiently, with theoretical guarantees, could be applicable and impact many different areas. (2) For (most of) the DRO problems studied in the paper, the authors present both upper and lower bounds for the convergence rate of the proposed approaches. (3) The experiments presented seem to corroborate with the theoretical results of the paper. The authors also provide the code used in the experiments. Moreover, their approach is shown to be easily integrated with PyTorch, which improves the applicability of these methods in exiting optimization pipelines.

Weaknesses: In my opinion, the paper does not present a major weakness. See below for minor comments on the paper.

Correctness: (1) The proofs in the supplementary material were not checked in detail, thus I cannot certify the soundness of the theoretical claims. (2) The authors seemed transparent with respect to their empirical methodology, that is, how the datasets were preprocessed and how hyperparameters were tuned for the experiments.

Clarity: In my opinion, the paper is mostly clear and well written, modulo some minor comments (see below).

Relation to Prior Work: I think the authors did a good job reporting the related literature, for example, when presenting a comparison with known results in Table 1.

Reproducibility: Yes

Additional Feedback: General comments and typos: (1) In Line 37, it is stated that it is straightforward to obtain unbiased gradient estimates for the surrogate risk. In contrast, in Line 146 it is stated that the gradient of an empirical estimate of L on a mini-batch of size n is clearly a biased estimate. It would be nice to explain a little better why these two facts are so clear/straightforward, since they are central to the discussion presented in the paper. (2) Line 61: it is not clear what is meant by “error floor”. (3) Line 75: at this point in the paper, it is not clear what is meant by “s \in S”. (4) Line 90: shouldn’t it be “\alpha^{-1}\epsilon^{-2}” (5) Line 152: the hyperlinked reference to “Nesterov” looks strange. I think it would be better to use a regular reference, that is, [46]. (6) As explained in the paragraph starting in Line 231, some of the results presented in Table 1 follow when \nu, \lambda and \alpha respect some inequalities. Can these conditions be assumed without loss of generality? If not, I think it needs to be mentioned that the results in Table 1 follow given some extra conditions. (7) One way to improve the empirical evaluation of the proposed methods would be to compare their performance with other methods from the literature (for example, the ones presented in Table 1). Even though those methods have worse theoretical guarantees, it would be interesting to see how they compare in practice. (8) In the last two equations of Line 910, isn’t “>=” supposed to be “<=” in both equations? (9) Line 1221/1222: what is the justification behind using \Delta L_{CVaR} rather than \Delta L_{kl-CVaR}, even though it goes against the theory?


Review 2

Summary and Contributions: ## UPDATE ## I'm grateful to the authors for kindly answering my question. I've read the response and will maintain my score. A minor suggestion: It would be helpful for practitioners if authors could present pseudocode of the algorithm in the appendix. ############ The paper proposes and analyzes distributionally robust optimization (DRO) algorithms. The analysis shows that the number of gradient evaluations the algorithms require is independent of the training set size and number of parameters, which makes the algorithms scalable to large instances. The analysis is based on the bound of bias caused by the use of empirical robust objective functions. A gradient estimation method with multi-level Monte Carlo (MLMC) is also analyzed, which yields optimal convergence results. The theoretical results are validated with experiments.

Strengths: - The paper makes many significant theoretical contributions to DRO, an important subject that is relevant to the NeurIPS community. - The experiments strongly support the theoretical results. - The advantages over previous work are clearly described.

Weaknesses: - In experiments, comparisons with other existing algorithms (e.g., [Curi et al. (2019); Namkoong and Duch (2016)]) are not presented. - For readers who are unfamiliar with this are (like me), it is hard to intuitively understand the key reason why such bounds, which are independent of the training set size and the number of parameters, are possible.

Correctness: It was hard to check the correctness, but no errors were found as far as I can see.

Clarity: The paper is well written and the results are clearly described.

Relation to Prior Work: Relation to previous work is clearly described.

Reproducibility: Yes

Additional Feedback: I would appreciate it if the authors could briefly describe the key point that makes it possible to obtain the bounds shown in Table 1 and why existing studies failed to do so.


Review 3

Summary and Contributions: The paper considers distributionally robust optimization problems with CVaR and \chi^2 divergence ambiguity sets. The authors first show that a biased gradient estimate, based on a mini batch approach, can be used to find an epsilon optimal solution with lower number of gradient evaluations compared to naive gradient based methods. The sample complexity then improves by using a better gradient estimate based on a Monte Carlo approach. =================================================== [Update after reading rebuttal]: I would like to thank the authors for answering my concerns. As a suggestion for the final version, it will be very interesting to include a plot comparing the execution time of the vanilla SGD with your proposed method. In practice, SGD variants with adaptive sampling scheme cannot outperform vanilla SGD with uniform sampling in terms of execution time. This is mainly due to time complexity of the sampling oracle. I was wondering if the authors can add a short remark or discussion about this concern. All in all, my evaluation remains the same.

Strengths: I think the paper certainly is a strong submission. In particular, the use of stochastic gradient descent with a more sophisticated gradient oracle in the context of distributionally robust optimization problems (with \chi^2 divergence and CVaR ambiguity sets) is new and very interesting. I would also replace the accelerated SGD algorithm with the simple averaged SGD algorithm since the averaged SGD converges to the minimizer of the problem with the rate O(1/T) + O(1/sqrt(T)); see for example Theorem 6.3 of [Bubeck 2015]. Note that using the accelerated SGD algorithm only improves the rate to O(1/T^2) + O(1/sqrt(T)). ========================================================= Bubeck, Sébastien. "Convex Optimization: Algorithms and Complexity." Foundations and Trends® in Machine Learning 8.3-4 (2015): 231-357.

Weaknesses: I found the paper a bit hard to read because the authors analyzed several different cases at the same theorem. However, I want to thank them to really push all theoretical results and consider the most general case. I also found one of the assumptions in the paper restrictive. Having bounded loss function simply implies that the feasible set X has to be compact. This is certainly fine because both average and accelerated SGD algorithms heavily relies on the project step on set X. My suggestion is to replace bounded loss assumption with bounded feasible set because together with Lipschitz continuity you will have a bounded loss function.

Correctness: I checked the proof, and to the best of my knowledge, all results are correct.

Clarity: The paper is very well-written.

Relation to Prior Work: The connection to previous studies are clearly made. I particularly really like Table 1 in the paper.

Reproducibility: Yes

Additional Feedback:


Review 4

Summary and Contributions: ##### Update ##### I have read the rebuttal and will maintain my score. ################## The authors offer an analysis of applying SGD to the primal problem of distributional robust optimization. While the mini-batch gradient estimator is an biased estimate for the true gradient, it is an unbiased estimate for a surrogate function which is proven to be within epsilon distance from the original objective as a function of the number of samples in the minibatch. The authors show convergence results for difference uncertainty sets of interest and conclude with experiments exhibiting convergence of standard SGM applied to DRO.

Strengths: While I have not confirmed correctness of the proofs (I assume a much longer paper will come out of this based on the long supplement), the main strength of the work is in the bounds on convergence rates that are claimed as they are independent of the number of points in the training set and the number of parameters. The authors note other works that use minibatch gradient estimates but do not offer any such analysis. DRO is of great relevance to the NeurIPS community so faster algorithms for such problems are important.

Weaknesses: The only weakness is in the experiments. Convergence is shown for the proposed method, but not compared against previous methods such as Dual SGM or Stochastic primal-dual. This could include various methods noted by the authors that do not have a convergence analysis. Furthermore, the reader would greatly benefit from seeing the benefit of DRO that exemplify how it can achieve more robust models on test data.

Correctness: I have not verified correctness.

Clarity: The paper is well-written. I recommend some changes to consider below.

Relation to Prior Work: It is clearly discussed. Various other works that use mini-batch gradient estimators are discussed, but it is claimed they do not have the convergence bounds stated here.

Reproducibility: Yes

Additional Feedback: In my opinion, the proof sketches do not add much to the reader. The space could be better used with other discussions. For example, define the gradients as done in the supplement in A.1.5 and explain explicitly why the gradient is biased but unbiased for the surrogate (rather than saying "Clearly").

[Author Response · NeurIPS 2020]

We thank the reviewers for their valuable time and insightful feedback. We begin by addressing two important points
raised by multiple reviewers, and address the remaining comments of each reviewer in turn.

**Intuition for the SGM bias** Reviewers 1 and 5 ask for clarification and intuition for why mini-batch gradient estimates
are biased for the true objective gradients but unbiased for the surrogate objective gradients. In a nutshell, the bias exists
because expectation and minimization do not commute, and the true and surrogate objectives differ exactly by their
order: see lines 681–682 in the supplementary material. On the other hand, the mini-batch loss estimate is unbiased for
the surrogate loss *by definition*, and since $\nabla$ and $\mathbb{E}$ do commute, we have that the mini-batch gradient is unbiased for the
surrogate gradient. To see why the objective *gradient* is also similarly biased, consider the constructions in Proposition
4 (which lower bounds the bias) except with $\ell(x; s) = x \cdot s$. Then, the objective gradient at $x > 0$ is proportional to the
loss must therefore be biased. We will discuss this in detail in the revised paper.

**Comparison to other methods** Reviewers 1, 2 and 5 ask for empirical comparison of
the proposed mini-batch method with Dual SGM as well as the primal dual approaches
of [15] and [43]. We agree that such comparisons are important, and we will add them to
the revised manuscript. We include here preliminary results comparing our method with
dual SGM for CVaR and $\chi^2$ penalty on the digits experiment. Our theory predicts that the
advantage of mini-batch over dual SGM increases as the uncertainty set grows (i.e., as $\alpha$
and $\lambda$ decrease). Consequently, we vary the uncertainty set size (re-tuning the learning rates
in a grid each time) and see results consistent with our prediction. For the revised paper
we will also perform ImageNet experiments and comparison with [15] and [43]. We note
that, consistently with the bound in Table 1, Namkoong and Duchi [43, Figure 1] observe
that stochastic primal-dual performs on par or worse than the full-batch method; it should
therefore be considerably slower than mini-batch SGM.

**Reviewer 1** Thank you for the detailed comments and particularly the helpful questions
and suggestions. We are glad you found our problem interesting and potentially impactful.
Below we address the additional comments and questions given in point 8 of the review; we
will make sure to include all clarifications in the revised paper as well. **(2)** By error floor,
we refer to the suboptimality of the solution mini-batch SGM with batch size $n$ finds when it has converged. That is,
the error floor is $\mathcal{L}(\bar{x}; P_0) - \inf_{x \in \mathcal{X}} \mathcal{L}(x; P_0)$, where $\bar{x} = \arg\min_{x \in \mathcal{X}} \overline{\mathcal{L}}(x; n)$. **(3)** In this context $\mathbb{S} = \{1, ..., N\}$ and
we refer to evaluating the loss over the entire dataset (see also Appendix A.2). **(4,8)** Thank you for pointing out these
typos. **(6)** The runtimes presented in Table 1 correspond to the multilevel Monte Carlo guarantees in Section 4, and as
such they require no assumptions on $\nu, \beta$ and $\alpha$. **(9)** Using $\mathcal{L}_{\text{CVaR}}$ instead of $\mathcal{L}_{\text{kl-CVaR}}$ saved us tuning a parameter (the
regularization strength), and still performed well compared to the full-batch method. A well-tuned smoothing parameter
might obtain better results, though preliminary experiments did not show a major difference.

**Reviewer 2** Thank you for the kind review and important questions. For intuition why it is possible to solve DRO
problems with complexity independent of the training set size $N$ is that the objective $\mathcal{L}(\cdot; P_0)$ is a *statistic* which one
can estimate and optimize using a sufficiently large sample from $P_0$ ([19] proves this rigorously). This holds true even
when $N = \infty$ (so $P_0$ has infinite support), and we therefore expect to have guarantees independent of $N$. The key
challenge in obtaining our $N$-independent rates is that the standard analysis of SGM does not apply, because of the
bias described above. We propose two ways to circumvent this issue. First, in Section 3 we characterize the surrogate
objective for which an unbiased estimate is easy to write down. There, the key points are to bound the bias (Proposition
1) and the variance (Proposition 2). Second, in Section 4 we use multilevel Monte Carlo to formulate a sophisticated
unbiased estimator of the objective (more precisely, one with arbitrarily low bias). There, the key point is bounding the
second moment (Proposition 3). We will further highlight these points in the revision.

**Reviewer 3** Thank you for the thoughtful suggestions; we are glad you found our paper interesting and well-written.
Replacing AGD with SGD would unfortunately not allow us to obtain the guarantees in Theorem 1. To see this, note
that the $O(1/T)$ in the SGD rate is proportional to the objective smoothness, which for us is $\Theta(\epsilon^{-1})$. Therefore, to
make the error $\epsilon$ we would have to take $T$ of the order $\epsilon^{-2}$, harming our convergence guarantee. (Note also that the
common $O(T^{-1/2})$ in both rates is proportional to the variance, which we make small by choosing a large batch size
and appealing to Proposition 2). Regarding the suggestion to subsume the bounded loss assumption, note that if we
only assume that $\mathcal{X}$ is bounded and $\ell(x; s)$ is Lipschitz in $x$, it does not give us bounded loss (consider $\ell(x; s) = x + s$
when $\mathbb{S} = \mathbb{R}$). We can, however, assume that $\ell(x_0; s) \in [-GR, GR]$ for all $s \in \mathbb{S}$ and some $x_0 \in \mathcal{X}$, which combined
with boundedness and Lipschitz assumptions would imply a bound on the loss; we will comment on this in the revision.

**Reviewer 5** Thank you for the helpful review and for highlighting important content and presentation issues. We
discuss test-time robustness in our experiments in Appendix F.6.1, and note that prior work report significance robustness
gains from the DRO objectives that we study [19, 31, 51, 68]. We hope that by providing efficient methods for DRO at
scale, our paper will enable new demonstrations of the benefits of DRO.

[Meta-Review · NeurIPS 2020]

The paper analyzes the bias and variance of a biased batch gradient estimator for solving DRO with CVaR and chisquare divergence uncertainty sets, and provides the upper/lower complexity bounds for the associated batch gradient methods. All reviewers found the algorithmic results interesting, given the growing relevance of DRO in ML research community. Please take the reviewers' comments into consideration in the revision and include numerical comparisons against previous methods as suggested by R3 and R5.